# Insights into DNA repeat expansions among 900,000 biobank participants

Margaux L. A. Hujoel[1,2,3,4,5 ✉], Robert E. Handsaker[3,6,7], David Tang[1,2,3,8], Nolan Kamitaki[1,2,3,8], Ronen E. Mukamel[1,2,3], Simone Rubinacci[1,2,3,9], Pier Francesco Palamara[10,11], Steven A. McCarroll[3,6,7] & Po-Ru Loh[1,2,3 ✉]

Expansions and contractions of tandem DNA repeats generate genetic variation in human populations and in human tissues. Some expanded repeats cause inherited disorders and some are also somatically unstable[1,2]. Here we analysed DNA sequencing data from over 900,000 participants in the UK Biobank and the All of Us Research Program using computational approaches to recognize, measure and learn from DNA-repeat instability. Repeats at different loci exhibited widely variable tissue-specific propensities to mutate in the germline and blood. Common alleles of repeats in *TCF4* and *ADGRE2* exhibited high rates of length mosaicism in the blood, demonstrating that most human genomes contain repeat elements that expand as we age. Genome-wide association analyses of the extent of somatic expansion of unstable repeat alleles identified 29 loci at which inherited variants increased expansion of one or more DNA repeats in blood ($P = 5 \times 10^{-8}$ to $2.5 \times 10^{-1,438}$). These genetic modifiers exhibited strong collective effects on repeat instability: at one repeat, somatic expansion rates varied fourfold between individuals with the highest and lowest 5% of polygenic scores. Modifier alleles at several DNA-repair genes exhibited opposite effects on the blood instability of the *TCF4* repeat compared with other DNA repeats. Expanded repeats in the 5′ untranslated region of the glutaminase (*GLS*) gene associated with stage 5 chronic kidney disease (odds ratio (OR) = 14.0 (5.7–34.3, 95% confidence interval (CI))) and liver diseases (OR = 3.0 (1.5–5.9, 95% CI)). These results point to complex dynamics of DNA repeats in human populations and across the human lifespan.

Short tandem repeats (STRs) of 1–6 bp of DNA are mutable genomic elements with diverse influences on cellular and organismal phenotypes[2]. Common STR polymorphisms, which have been characterized in human populations using short-read[3] and long-read[4] sequencing, influence gene expression[5] and complex traits[6,7]. Rare STR expansions cause more than 60 genetic disorders[1]. The allelic diversity that underlies these effects is generated by frequent mutation: around 1 million polymorphic STRs in the human genome generate around 50–60 de novo repeat-length mutations per offspring[8–10]. Germline mutation rates of specific STRs vary widely[11] and are influenced by repeat motif sequence, interruptions of pure repeats and number of repeat units[8–12] as well as genetic variation in DNA-repair genes[9].

STRs are also prone to somatic mutation[2], and lifelong somatic expansion in at least one STR locus can lead to disease. Recently, genome-wide association studies (GWASs) have provided insights into the molecular mechanisms underlying somatic repeat instability[13] by finding common genetic modifiers of the timing or progression of Huntington's disease (HD)[14–19], which is caused by inherited alleles in which a CAG repeat in the *HTT* gene is longer than 35 CAGs; these genetic modifiers were found in many DNA-repair genes that affect the stability of DNA repeats[14–19]. Neurodegeneration in HD was subsequently found to be caused by somatic expansion of this repeat beyond a high threshold of about 150 CAG repeats[20]. The genetic-modifier studies, so far of up to 16,640 persons with HD, have provided early clues toward a few potential therapeutic targets for slowing or halting somatic expansion of DNA repeats; however, the number of such potential targets is so far modest.

Whole-genome sequencing (WGS) of biobank cohorts offers opportunities to study repeat instability in much larger sample sizes than previously possible. Here we analysed repeat instability at 356,131 polymorphic repeat loci using short-read WGS data from the blood-derived DNA of 490,416 participants in UK Biobank (UKB)[21] and 414,830 participants in All of Us (AoU)[22]. To do so, we developed several computational techniques, overcoming challenges in estimating the length and

[1]Division of Genetics, Department of Medicine, Brigham and Women's Hospital and Harvard Medical School, Boston, MA, USA. [2]Center for Data Sciences, Brigham and Women's Hospital and Harvard Medical School, Boston, MA, USA. [3]Program in Medical and Population Genetics, Broad Institute of MIT and Harvard, Cambridge, MA, USA. [4]Department of Human Genetics, David Geffen School of Medicine, University of California, Los Angeles, Los Angeles, CA, USA. [5]Department of Computational Medicine, David Geffen School of Medicine, University of California, Los Angeles, Los Angeles, CA, USA. [6]Stanley Center for Psychiatric Research, Broad Institute of MIT and Harvard University, Boston, MA, USA. [7]Department of Genetics, Harvard Medical School, Boston, MA, USA. [8]Department of Biomedical Informatics, Harvard Medical School, Boston, MA, USA. [9]Institute for Molecular Medicine Finland, Helsinki, Finland. [10]Department of Statistics, University of Oxford, Oxford, UK. [11]Centre for Human Genetics, University of Oxford, Oxford, UK. ✉e-mail: mhujoel@ucla.edu; poruloh@broadinstitute.org

instability of DNA repeats from large numbers of short WGS reads[23]. These methods enabled us to characterize allele-specific expansion and contraction rates of common repeats, identify genetic influences on somatic repeat expansion and identify associations of expanded repeats with diseases.

## CAG-repeat expansions in the UKB

We began by analysing CAG trinucleotide repeats, which we could efficiently ascertain from biobank sequencing data and which cause many progressive, neurodegenerative repeat-expansion disorders[1,2]. We identified UKB participants with long CAG-repeat alleles (≥45 repeat units) by analysing WGS data for 151 bp sequencing reads comprised entirely or almost entirely of CAG-repeat units (in-repeat reads (IRRs); Extended Data Fig. 1a). Such reads were easily extractable, as nearly all of them had been aligned to the *TCF4* CAG-repeat sequence by bwa[24] (Supplementary Fig. 1). For each participant with one or more IRRs, we determined the locus or loci from which the IRRs originated by identifying mate sequences that mapped near one of 1,159 commonly polymorphic CAG-repeat loci[3].

The vast majority of CAG-repeat expansions in the UKB occurred at only a few loci: 18 autosomal CAG-repeat sequences in the human genome were expanded to at least 45 repeat units in at least five UKB participants (Extended Data Fig. 1b and Supplementary Table 1). Three repeat loci were expanded in thousands of UKB participants—*CA10* (137,673 participants), *TCF4* (42,004) and *ATXN8OS* (7,736)—together accounting for 97% of all observed expansions beyond 45 repeat units. Most of these repeats (15 out of 18) were in transcribed genomic regions, consistent with the idea that transcription contributes to repeat instability[25] (Supplementary Table 2). For 9 out of the 18 repeats, expanded alleles are known to be pathogenic[1].

To study the mutability of these repeats, we measured the lengths of common, short alleles of each repeat (≤30 repeat units) by analysing sequencing reads that spanned repeat alleles, focusing on 15 repeat loci that passed additional filters (Extended Data Fig. 1 and Supplementary Table 1). These analyses recovered repeat-length distributions consistent with previous analyses[26] (Extended Data Fig. 1b).

## Germline instability of common CAG repeats

We first analysed germline instability of these repeats, using the large UKB cohort to obtain high-resolution estimates of germline mutability (providing context for analyses of somatic mutability). To estimate allele-specific intergenerational expansion and contraction rates of each repeat, we analysed length discordances among alleles belonging to genomic tracts inherited identical-by-descent (IBD) from shared ancestors, building on IBD-based analyses of single-nucleotide mutations[27–29] (Fig. 1a). We validated this approach using two complementary methods (Supplementary Fig. 2).

Across all 15 CAG-repeat loci, intergenerational mutation rates increased with allele length, rising to 0.5–0.9% per generation for single-repeat-unit expansions of the longest common alleles of repeats in *GLS*, *DMPK* and *ATXN8OS* (Extended Data Figs. 1b and 2). The average mutation rate per locus ranged from $8.2 \times 10^{-5}$ to $9.5 \times 10^{-4}$ (Supplementary Table 3). These rates are relatively high for trinucleotide repeats[8] and exceed the genome-wide average for STRs (around $5 \times 10^{-5}$ per haplotype per generation)[8–10]. Repeat loci tended to either expand more often than contract (particularly so for *ATXN8OS* and *GLS*) or to have similar expansion and contraction rates (Extended Data Figs. 1b and 2). Interruptions of repeat sequences (that is, intrarepeat sequence variants) greatly stabilized alleles: a common 18-repeat *TCF4* allele containing an interruption in its ninth repeat unit exhibited a 135-fold (54–336, 95% CI) lower expansion rate compared with the uninterrupted 18-repeat allele, and an interruption in the second-to-last repeat unit of a 19-repeat *GLS* allele decreased the expansion rate 3.7-fold (1.9–7.2,

95% CI) (Fig. 1b). These results corroborate previous observations that repeat interruptions stabilize the expansion of pathogenic alleles[30–32] and quantify the strength of such effects in the germline.

## Somatic expansion of common CAG repeats

These high rates of germline instability led us to wonder whether common alleles of some repeats might be sufficiently unstable in blood cells for somatic length-change mutations to be ascertainable in short-read WGS data. Identifying such mutations is challenging because polymerase slippage during PCR amplification can spuriously alter repeat lengths[33–35]. Such 'PCR stutter' errors are unavoidable during Illumina sequencing by synthesis, which uses PCR for bridge amplification of DNA fragments[36]. However, we realized that this PCR error mode tends to produce predictable patterns of reduced base quality scores within sequencing reads, enabling us to detect and exclude most reads with artefactual CAG length mutations (Fig. 1c and Supplementary Fig. 3). We applied this filtering strategy in the UKB to estimate repeat-specific, allele-specific somatic expansion rates, which we quantified as the average fraction of blood cells in which a given repeat allele has expanded by one repeat unit.

For 4 out of the 15 CAG repeats (in *TCF4*, *GLS*, *DMPK* and *ATN1*), we detected significant increases in somatic single-repeat-unit expansion rates with age (Extended Data Fig. 3). These findings were replicated in AoU, in which the wider age range of participants (aged 18 to 90+ years) revealed clear increases in fractions of blood cells containing somatic expansions with increasing age and with increasing allele length (Fig. 1d and Extended Data Fig. 4). *TCF4* repeats were the most somatically unstable: individuals carrying alleles with 25 or more repeat units typically exhibited somatic expansion in more than 1% of blood cells by the age of 55 years (Fig. 1d). We did not observe age-associated contraction of any of the 15 repeat loci.

Comparing these estimates of somatic one-repeat-unit expansion rates with our estimates of intergenerational mutation rates showed that the relative (blood/germline) rates of CAG-repeat expansion varied severalfold across repeat loci (Fig. 1e). The *TCF4* repeat exhibited the greatest somatic instability in blood but was relatively stable in the germline, whereas the *GLS* repeat displayed the opposite behaviour (Fig. 1e), as did the *DMPK* repeat (Extended Data Figs. 1b, 2 and 4). These results align with observations that somatic instability of pathogenic repeat expansions is highly tissue-specific, perhaps due to differences in transcription or *trans*-acting factors[25,37–41]. Consistent with the former hypothesis, the four repeats for which we detected instability in blood are in genes with significantly higher expression in blood (Wilcoxon rank-sum test, $P = 0.034$; note that all $P$ values reported in this Article were calculated using two-sided statistical tests; Supplementary Fig. 4).

## Somatic expansion of long *TCF4* CAG repeats

The high somatic expansion rates of *TCF4* repeat alleles—even those of shorter lengths—suggested the possibility that long *TCF4* alleles (≥45 repeat units) might be sufficiently unstable in blood to allow individual-level phenotyping of somatic expansion using short-read WGS data. This would provide an opportunity to learn about instability of long repeats from somatic expansions in very many people—potentially enabling the identification of genetic modifiers of repeat instability[13–19,42,43]—as long *TCF4* alleles are common (42,004 carriers in the UKB; Extended Data Fig. 1b).

A barrier to analysing repeat expansions from short-read WGS data is that alleles exceeding the length of a sequencing read (151 bp) cannot be directly sized. However, short-read WGS data does permit rough estimation of the length of a long allele by counting in-repeat reads[44] (Extended Data Fig. 1a). In an individual who is mosaic for somatic expansions that vary across cells, this approach estimates the average length of expanded alleles.

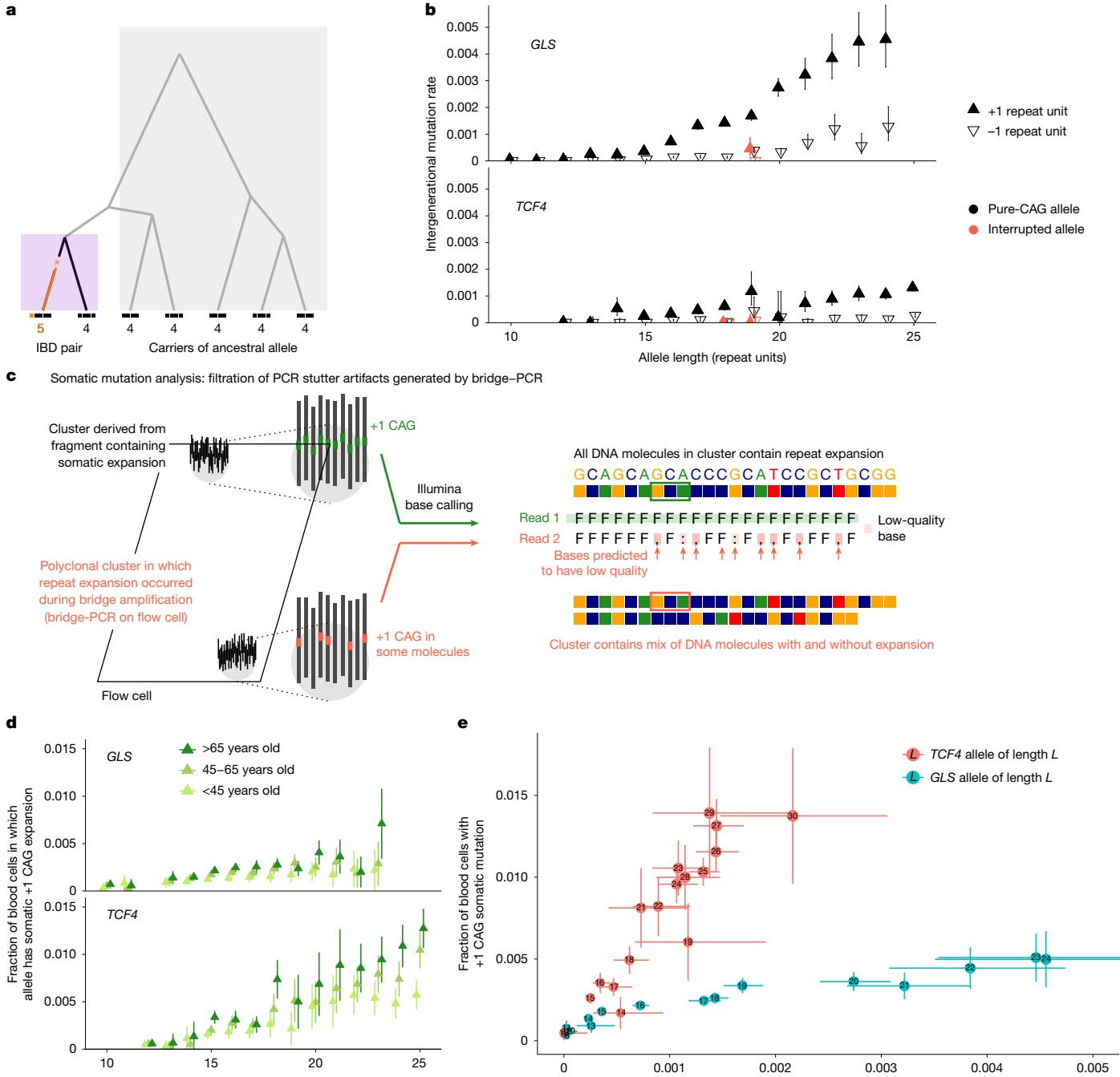

**Fig. 1 | Germline and somatic instability of common CAG-repeat alleles.**
**a**, Germline mutation rates were estimated by analysing discordance rates among alleles inherited within IBD tracts shared by pairs of UKB participants. Ancestral alleles were imputed from more-distantly shared haplotypes. **b**, Per-generation rates of germline expansion (+1 repeat unit) and contraction (−1 repeat unit) of *GLS* and *TCF4* repeat alleles, estimated in the UKB. **c**, The analytical strategy for estimating somatic mutation rates by detecting and filtering out reads that are likely to reflect PCR artifacts introduced during sequencing. During PCR-based bridge amplification on a flow cell, a DNA fragment is clonally amplified into a cluster of colocalized DNA molecules. A PCR stutter error results in a polyclonal cluster containing a mixture of DNA molecules with and without the error. If the molecules containing the error constitute the majority of the cluster, the sequencing read generated from the cluster (reflecting the majority base at each position within the read) will contain the error, but the heterogeneity of the cluster will reduce base qualities at positions within the read that mismatch between molecules with and without the error. **d**, The rates of somatic expansion of *GLS* and *TCF4* repeat alleles (that is, the fractions of blood cells in which an allele has expanded by +1 repeat unit), stratified by age in AoU. **e**, Somatic mutation rates in the UKB plotted against germline mutation rates for *GLS* and *TCF4* repeat alleles. The error bars show the 95% confidence intervals (CIs). Sample sizes are provided in Supplementary Table 3.

We analysed somatic expansion of long *TCF4* alleles in the UKB and AoU by applying this approach with two methodological improvements. First, to control for variation in lengths of inherited *TCF4* alleles, we used imputation to calibrate each individual's allele length against measurements from other individuals sharing the same inherited allele (in lieu of longitudinal measurements). Stratifying individuals by imputed allele length showed that somatic expansion accelerates rapidly with *TCF4* allele size, reaching around 1 repeat unit per year for 100-repeat alleles (Extended Data Fig. 5a). Second, to reduce noise in estimates of long *TCF4* allele lengths, we devised a better-powered

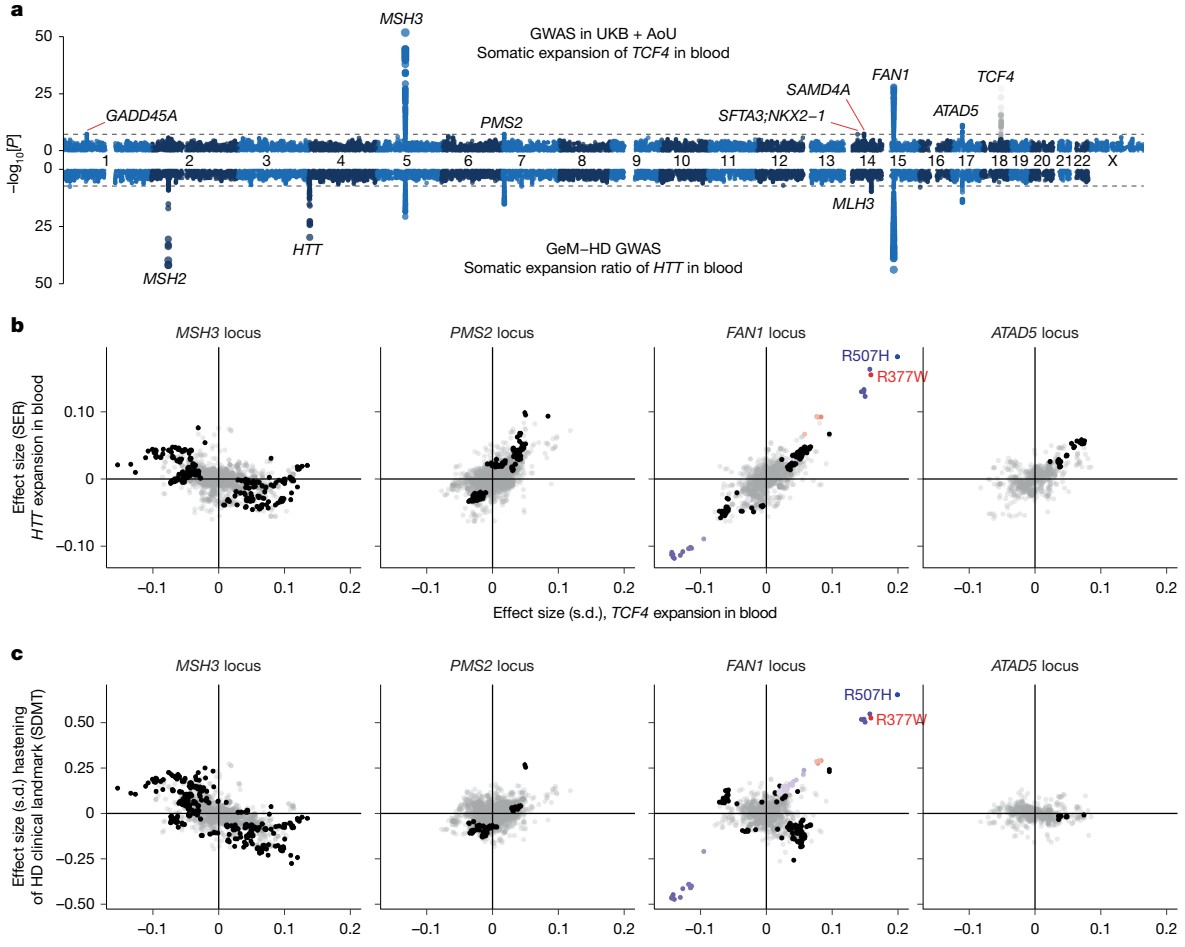

**Fig. 2 | Genetic influences on somatic expansion of *TCF4* repeat alleles in blood. a**, Genome-wide associations with somatic instability of long *TCF4* repeat alleles in the blood (top, meta-analysed across the UKB ($n = 40,231$) and AoU ($n = 8,217$)) compared with genetic associations with somatic instability of pathogenic *HTT* repeat alleles in the blood (bottom; from ref. 18). The *TCF4* locus is shown in grey because these associations could reflect imperfect control for inherited *TCF4* allele length. **b**, Comparison of the effect sizes of variants at *MSH3*, *PMS2*, *FAN1* and *ATAD5* for somatic expansion of *HTT* repeats in the blood (quantified by the somatic expansion ratio; SER[18]) versus *TCF4* repeats in the

blood. **c**, Analogous comparison for variant effect sizes for hastening of an HD clinical landmark of cognitive decline (symbol digit modalities test, SDMT)[18]. In each plot, variants within 1 Mb of the lead variant for *TCF4* somatic expansion are plotted in black if they reached $P < 10^{-5}$ for association with at least one of the two phenotypes; for *FAN1*, a subset of these variants is plotted in red or blue according to linkage disequilibrium with the two low-frequency *FAN1* missense variants ($r^2 > 0.05$). Variants with $P > 10^{-5}$ for both phenotypes are plotted in light grey.

metric based on the number of sequenced DNA fragments derived from a highly expanded repeat (Extended Data Figs. 5b and 6a,b). Long-read sequencing of blood-derived DNA from AoU participants ($n = 1,027$, of whom 28 had long *TCF4* alleles) corroborated *TCF4* allele-length estimates from short-read WGS and demonstrated extensive mosaicism of expanded alleles[41,45] (Extended Data Fig. 5c).

## Genetic modifiers of *TCF4* repeat expansion

Genome-wide association analysis of an optimized *TCF4* somatic-expansion phenotype (Extended Data Fig. 6c) in 48,448 UKB and AoU participants identified seven loci at which common variants modulate *TCF4* repeat expansion in blood ($P < 5 \times 10^{-8}$; Fig. 2a and Supplementary Table 4). Four loci—at *MSH3* ($P = 2.0 \times 10^{-52}$), *FAN1* ($P = 8.5 \times 10^{-29}$), *ATAD5* ($P = 4.9 \times 10^{-12}$) and *PMS2* ($P = 3.0 \times 10^{-8}$)—overlapped DNA-repair and DNA-damage-response genes that were recently implicated in somatic expansion of the *HTT* CAG repeat in blood[18] (Fig. 2a). The three other modifier loci included *GADD45A* ($P = 2.9 \times 10^{-8}$), which encodes a growth arrest and DNA damage protein that binds to R-loops[46].

Comparing genetic modifiers of somatic expansion of the *TCF4* and *HTT* CAG repeats in blood revealed both consistency and heterogeneity

of effects (Fig. 2a,b). Common haplotypes at *PMS2*, *FAN1* and *ATAD5* were associated with broadly concordant effects on *TCF4* and *HTT* repeat expansion in blood, whereas at *MSH3*, common haplotypes that decreased expansion of the *TCF4* repeat tended to increase expansion of the *HTT* repeat in blood (Fig. 2b). Moreover, the strongest modifier of *HTT* expansion in blood—a haplotype containing a missense variant in *MSH2* also implicated in germline STR mutation[9]—appeared not to affect *TCF4* expansion ($P = 0.96$; Fig. 2a and Supplementary Table 5).

Similarly, comparing genetic modifiers of *TCF4* repeat expansion in the blood to genetic modifiers of HD age-at-landmark phenotypes[18] (which are probably regulating *HTT* repeat expansion in neurons) showed that, at both *MSH3* and *FAN1*, common haplotypes that decreased expansion of the *TCF4* repeat in blood appeared to increase the expansion of the *HTT* repeat in the brain (Fig. 2c and Supplementary Table 6). By contrast, two missense variants that reduce FAN1 activity[47] appeared to increase the expansion of both the *TCF4* repeat (in blood) and *HTT* repeat (in brain) (Fig. 2c). These results suggest that the tissue-specific instability of many trinucleotide repeats[37–41] may arise from complex regulation of mismatch repair processes that differs across cell types[18] and even across repeat loci, perhaps interacting with locus-specific differences in chromatin structure or other epigenomic properties.

We also compared modifiers of *TCF4* repeat expansion in blood to loci that influence risk of Fuchs endothelial corneal dystrophy (FECD), a common age-associated eye disorder that is thought to be caused (in most cases) by expansion of the *TCF4* repeat in corneal endothelial cells[48,49]. Notably, no modifiers of *TCF4* repeat expansion in the blood overlapped with FECD risk loci[50], and none of our lead variants for *TCF4* blood instability (Supplementary Table 4) were associated with FECD ($P > 0.15$) in a recent well-powered GWAS[50,51]. Moreover, FECD risk conferred by long *TCF4* repeats appeared to plateau for allele lengths beyond around 75 repeat units (Extended Data Fig. 7). Further work will be required to determine whether the instability-modifying genetic effects that we identified are specific to blood (which is conceivable given the very different (more extreme) dynamics of *TCF4* somatic expansion in corneal endothelium[41]) and whether any modifiers of somatic expansion influence age at FECD onset.

## Varied genetic effects on instability of 17 STRs

A much-larger set of DNA repeats involves other (non-CAG) sequence motifs, and the above results motivated us to investigate their expansion. To this end, we developed a computationally efficient tool for extracting IRRs with any 2–6 bp motif from WGS read alignments and applied it to the UKB WGS data. Mapping these IRRs to 356,131 polymorphic STRs identified 154 STRs for which long repeat alleles (>150 bp) were common (>0.5% carrier frequency; Supplementary Data 1). We constructed somatic-expansion phenotypes from IRR counts for these repeats, controlling for inherited allele lengths (inferred from imputation) as before. To identify STRs with evidence of somatic expansion, we tested these phenotypes for association with age or with the *MSH3* and *MSH2* haplotypes that were most strongly associated with blood instability of *TCF4* and *HTT* repeats. This approach was motivated by initial GWAS analyses on STRs with somatic-expansion phenotypes that associated with age (Supplementary Methods): in these analyses, *MSH2* and *MSH3* haplotypes were consistently the lead variants and sometimes associated more strongly than age, suggesting that, for some repeats, the effects of genetic modifiers—which act across an individual's years of life (mean 56.5 years in the UKB)—might be easier to detect than the effects of age differences (s.d., 8 years) on somatic expansion.

These analyses identified 17 STRs for which one or more of the three tests suggested evidence of somatic instability ($P < 0.0001$; Fig. 3a and Supplementary Data 2). These 17 unstable STRs represented 7 distinct 2–5 bp repeat motifs. Half of these STRs were located in genes that are highly expressed in blood, while six appeared to be in untranscribed regions (Fig. 3a). At some unstable STRs, expanded alleles were very common. Long alleles of an intronic AAAG tetranucleotide repeat in *ADGRE2* (carried by 49% of European-ancestry UKB participants) expanded at an average rate of 0.4 repeat units per decade, demonstrating that human genomes commonly contain repeat elements that expand as we age (Fig. 3b).

GWAS of somatic-expansion phenotypes for the 17 unstable STRs identified 7 loci at which common variants appear to modulate instability of these repeats in blood cells ($P = 1.2 \times 10^{-9}$ to $1.4 \times 10^{-878}$; Extended Data Fig. 8 and Supplementary Data 3). Variants in four mismatch repair genes (*MLH3*, *MSH3*, *MSH2* and *PMS2*) were each associated with somatic expansion of three or more STRs. The relative contributions of these genes to repeat instability varied across STRs, with *MSH2* variation having greater influences on dinucleotide repeats and *MSH3* variation having greater influences on STRs with longer motifs (Fig. 3a and Supplementary Data 4), consistent with MutSβ (a heterodimer of MSH2 and MSH3) having higher affinity for longer insertion–deletion loops in DNA[52,53]. Across a broader set of modifier haplotypes identified by fine-mapping genetic associations with optimized somatic-expansion phenotypes (see below), different modifiers appeared to influence different subsets of STRs (Fig. 3c) but generally with a consistent effect direction, with the exception of several opposite-direction effects on somatic expansion of the *TCF4* repeat (Fig. 3c). Multiple modifiers were associated with opposite-direction effects on STR expansion in the blood compared with *HTT* repeat expansion in the brain (as inferred from the timing of HD phenotypes), consistent with recent findings[18] (Fig. 3c).

## Genetic determinants of AAAG-repeat expansions

Somatic expansion of AAAG repeats at two loci (at chromosome 2: 232.4 Mb and chromosome 19: 14.8 Mb) was particularly strongly shaped by inherited variation, prompting deeper analyses. Mid-length alleles of these repeats (19–26 repeat units) were sufficiently common and unstable for somatic expansions to often be directly observable from spanning reads, enabling us to construct mid-length-allele somatic-expansion phenotypes for these two STRs in the UKB and AoU. We also optimized the common chromosome 19: 14.8 Mb (*ADGRE2*) long-allele somatic-expansion phenotype in the UKB to increase GWAS power.

Common and low-frequency variants (minor allele frequency (MAF) > 0.1%) at 26 loci were associated with these AAAG somatic-expansion phenotypes ($P = 5 \times 10^{-8}$ to $2.5 \times 10^{-1,438}$; Fig. 4a and Supplementary Data 5). Beyond the top associations at DNA-repair genes previously found to be modifiers in HD (*MSH3*, *MSH2*, *MLH3*, *PMS2* and *PMS1*), additional modifier loci included genes involved in recognizing DNA damage (*XPC* and *PARP1*; $P = 1.1 \times 10^{-21}$ and $2.4 \times 10^{-45}$) and in responses to DNA damage (*NEIL2*, which encodes a DNA glycosylase involved in base excision repair[54], and *SMARCAD1*, which encodes a chromatin remodeller that can play critical roles in DNA repair[55]; $P = 1.4 \times 10^{-14}$ and $1.2 \times 10^{-20}$). Other loci with subtler effects on somatic expansion contained genes for transcription factors such as *RUNX1* and the *HOXA* and *HOXB* clusters, suggesting additional, harder-to-interpret complexity in the ways in which genetics shapes repeat instability. The effect sizes of instability-modifying haplotypes were broadly consistent across the two AAAG-repeat loci (chromosome 2: 232.4 Mb versus chromosome 19: 14.8 Mb) and across allele length ranges (mid-length versus long alleles) with a few exceptions (for example, variation at *PARP1* appeared to have a larger influence on mid-length alleles; Fig. 4b and Extended Data Fig. 9). Fine-mapping analyses suggested the presence of multiple causal variants at several modifier loci including an expression quantitative trait locus (eQTL) signal at *MSH6* (Supplementary Data 6).

Burden tests identified strong instability-modifying effects of rare coding variants in nine genes (Fig. 4c and Supplementary Table 7). These genes included seven mismatch repair genes, corroborating recent results from an HD mouse model[56], as well as *XPC* ($P = 6.3 \times 10^{-14}$) and *NEIL2* ($P = 1.7 \times 10^{-6}$), at which we also detected common modifier haplotypes. Variants predicted to cause loss-of-function (pLoF) of *XPC* ($P = 4.8 \times 10^{-10}$) and *NEIL2* ($P = 0.0015$) were associated with considerably increased somatic mutation of chromosome 19: 14.8 Mb AAAG repeats ($\beta = +0.32$ (s.e., 0.05) and +0.22 (s.e., 0.07) standard deviations; Fig. 4c), suggesting that deficiencies in nucleotide-excision repair and base-excision repair increase instability of repeated DNA. *NEIL2* was also recently implicated in de novo STR mutation[9]. A low-frequency missense variant in *PMS1* (rs61756360, *PMS1* T75I; allele frequency = 0.2%) was associated with decreased expansion of AAAG repeats (Supplementary Data 5) and appeared to have a previously unrecognized onset-delaying effect on HD ($\beta = 5.2$ years (s.e., 1.9)); $P = 0.005$ in ref. 17).

The combined effects of these genetic modifiers across an individual's genome strongly influenced mutation rates of these AAAG repeats: somatic expansion rates varied by 4.0-fold (3.7–4.3) between individuals with the highest and lowest 5% of polygenic scores (trained in the UKB and applied in AoU) for expansion of the chromosome 2: 232.4 Mb AAAG repeat (Fig. 4d). These strong genetic effects also provided power to explore gene-by-gene and gene-by-environment interactions. We observed approximately multiplicative effects of genetic modifiers

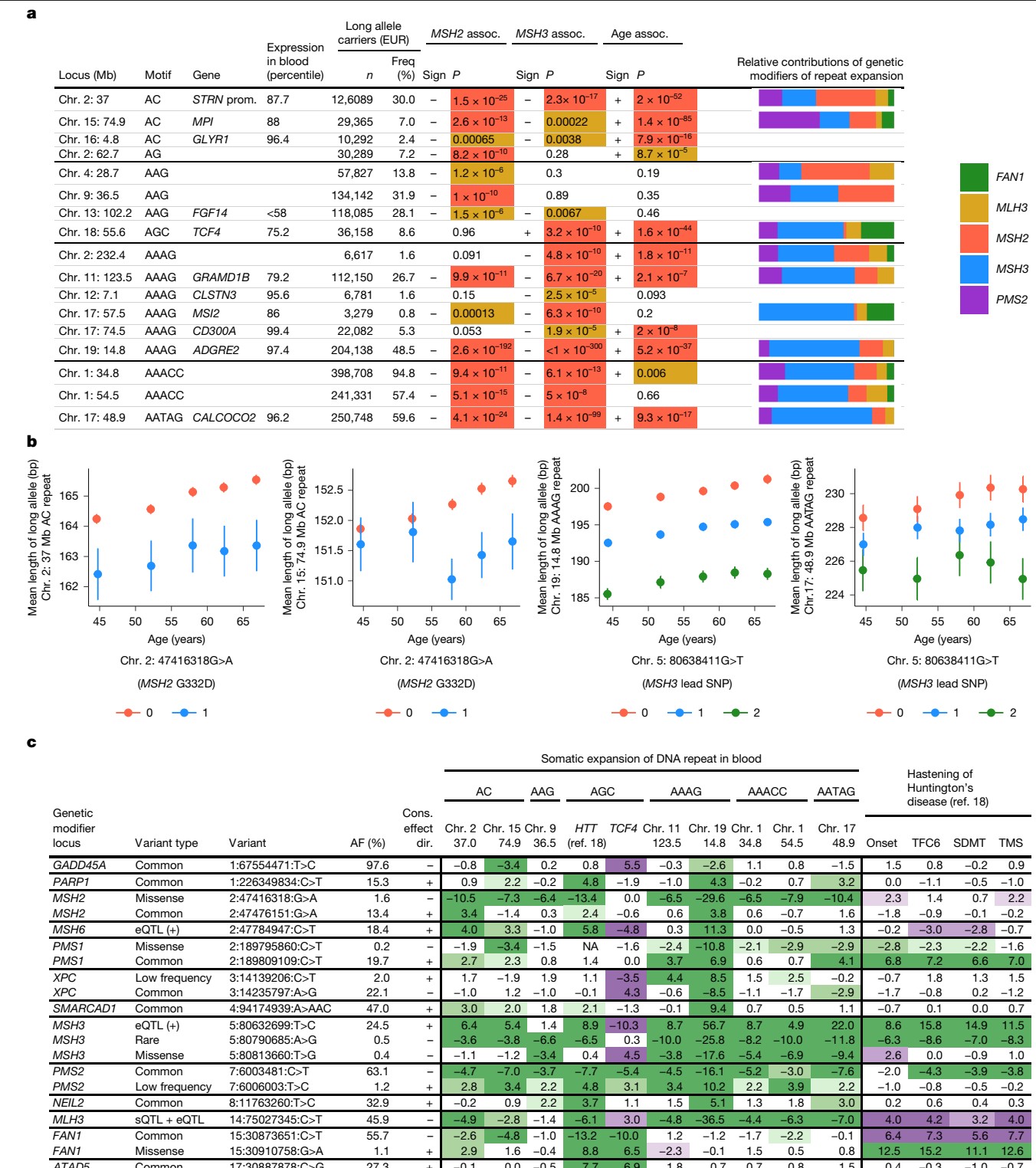

**Fig. 3 | Variation in genetic influences on 17 unstable STRs. a**, Genomic context, population frequencies (freq) among 420,522 unrelated European-ancestry UKB participants, associations (assoc.) with *MSH2* and *MSH3* variants and age, and the relative contributions of genetic modifiers of instability of 17 STRs. Prom., promoter. The relative contributions of five genetic modifier loci were estimated using local heritability analyses for STRs with sufficient signal (specifically, local heritability *z*-score > 2.5 for at least one of the five modifiers). **b**, The mean lengths of long alleles among UKB participants heterozygous for a long allele (based on imputation; from left to right, *n* = 155,291, 81,387, 257,934 and 240,294), stratified by age quintile and by genotype of an instability-modifying haplotype. The two STRs with the strongest age association and

the two with the strongest genetic associations are shown. Error bars show the 95% CIs. **c**, Associations of modifier haplotypes of DNA-repair and DNA-damage-response genes with blood instability of 10 STRs (including *HTT*[18]) and with hastening of four HD clinical phenotypes[18]. The table cells contain *z* statistics from association analyses; associations with *P* < 0.05, 0.01 and 0.001 are shaded in green or purple depending on whether the effect size agrees or disagrees with the consensus effect direction (cons. effect dir.) for blood instability. The effect sign in each table cell corresponds to the direction of effect on repeat instability: a positive effect indicates that the alternate allele associated with increased somatic expansion or hastening of an HD clinical landmark. TFC6, a score of 6 on the 13-point total functional capacity scale; TMS, total motor score.

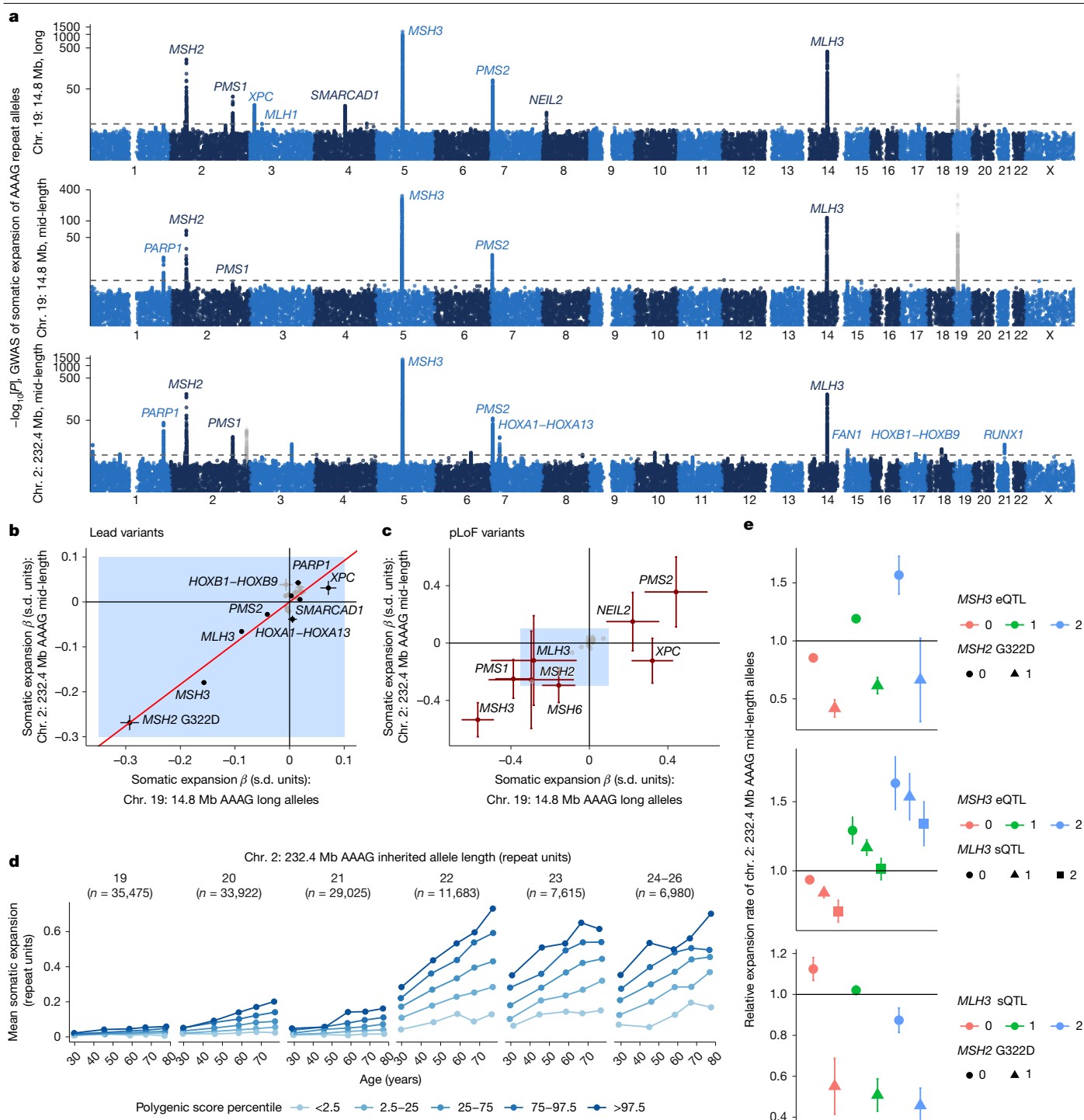

**Fig. 4 | Genetic determinants of somatic expansion of common AAAG repeats in blood. a**, Genome-wide associations with blood-instability of long chromosome 19:14.8 Mb AAAG repeat alleles (top; UKB), mid-length chromosome 19:14.8 Mb AAAG repeats (middle; UKB + AoU meta-analysis) and mid-length chromosome 2:232.4 Mb AAAG repeats (bottom; UKB + AoU). Associations of variants proximal to the repeat loci are shown in grey as they could reflect imperfect control for lengths of inherited alleles. **b**, Comparison of effect sizes for blood instability of mid-length chromosome 2:232.4 Mb AAAG repeats versus long chromosome 19:14.8 Mb AAAG repeats for index variants from the common-variant GWAS (that is, lead associations with MAF > 1%). Black dots correspond to labelled loci. **c**, Analogous comparison for rare pLoF variants in the UKB. The blue shaded rectangle and grey dots within it correspond to effect size range and common-variant associations shown in **b**.

**d**, The mean level of somatic expansion of mid-length chromosome 2:232.4 Mb AAAG repeat alleles (that is, the average increase in repeat length across spanning reads) observed in AoU participants who inherited one allele of the indicated length (19, 20, 21, 22, 23, 24, 25 or 26 repeat units). Plots were stratified by age and by polygenic score for somatic expansion (using a model that was fit on UKB data). **e**, The relative expansion rates of mid-length chromosome 2:232.4 Mb AAAG repeats in subsets of AoU participants with different pairwise combinations of three modifier genotypes: the lead *MSH3* eQTL (chromosome 5:80632699T>C), the *MSH2* G322D missense SNP (chromosome 2:47416318G>A) and a top *MLH3* splicing QTL (chromosome 14:75002247G>C). The error bars show the 95% CIs. Sample sizes are provided in Supplementary Data 5.

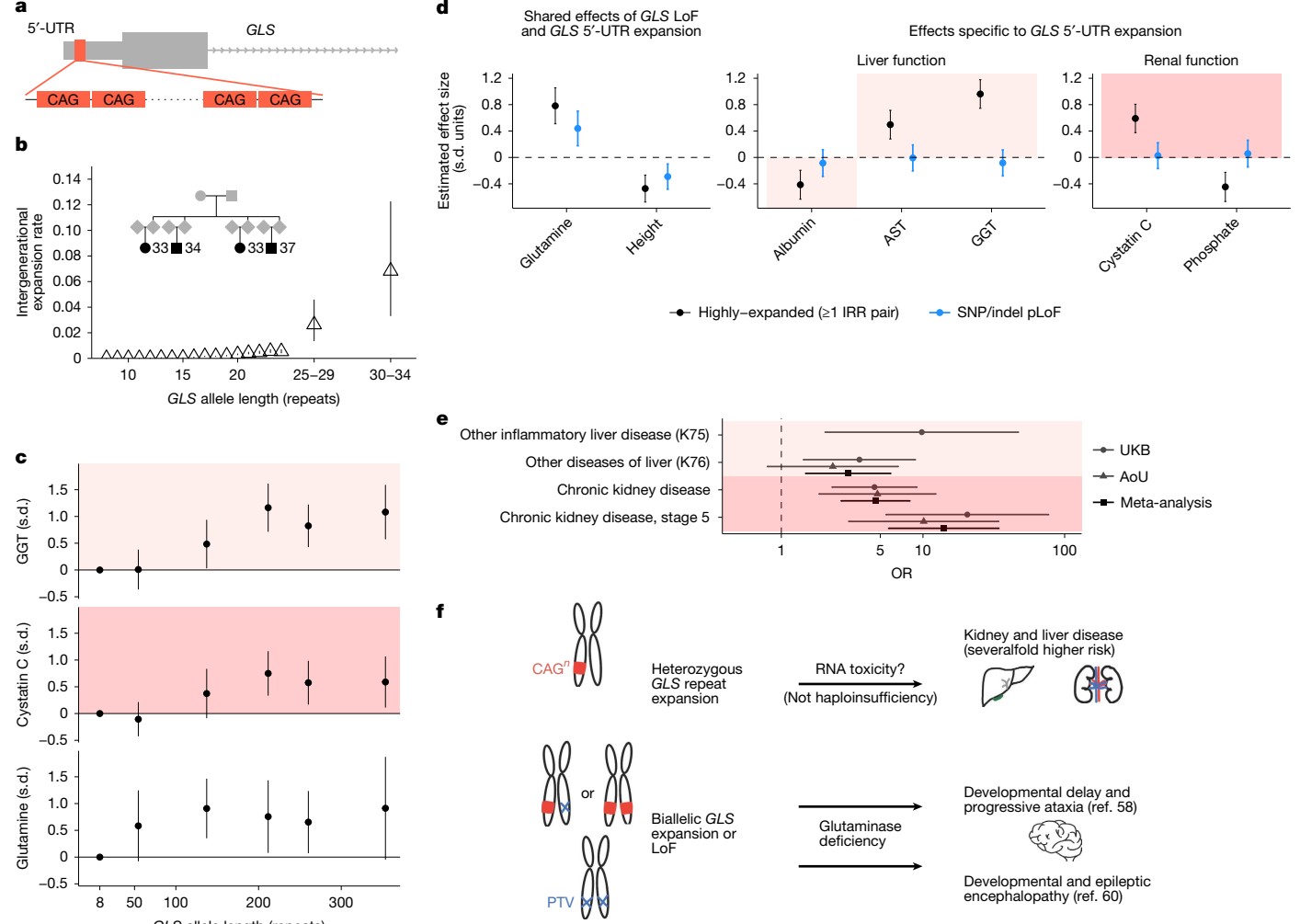

**Fig. 5 | Instability and pathogenicity of CAG-repeat expansions in the 5′ UTR of *GLS*. a**, The location of a polymorphic CAG repeat within the 5′ UTR of *GLS*. **b**, The intergenerational expansion rates of short- and mid-length *GLS* alleles estimated from IBD (for common alleles containing ≤24 repeat units) and from pairs of related UKB participants carrying rarer, mid-length alleles (25–34 repeat units; *n* = 390). Inset: multiple mutations observed among four related UKB participants (likely to be mutual first cousins) carrying mid-length *GLS* repeat alleles. **c**, The mean GGT, cystatin C and glutamine levels (adjusted for age, age squared and sex) among individuals with long *GLS* alleles (stratified into quintiles; *n* = 131) and individuals without a long allele (plotted at the modal length of 8 repeats). **d**, Effect sizes of *GLS* alleles for liver and renal biomarkers and other quantitative traits measured in the UKB, adjusted for age, age squared and sex. *GLS* repeat expansions and SNP and indel variants predicted to cause loss of function (*n* = 109) were all associated with increased glutamine levels

(inverse-normal transformed) and decreased height; by contrast, only highly expanded *GLS* repeat alleles (≥1 IRR pair, that is, around 100+ repeat units; *n* = 94) were associated with altered serum biomarker levels. The shaded rectangles indicate biomarker effect directions typically associated with disease. **e**, Increased odds of liver and renal diseases among UKB and AoU participants with highly expanded *GLS* repeat alleles compared with participants without a long allele; points plotted are odds ratios. The stage 5 chronic kidney disease phenotype included ICD-10 codes for end-stage renal disease. Analyses in the UKB (*n* = 421,377) were adjusted for age and sex; analyses in AoU (*n* = 229,043) were adjusted for age, age squared, sex and genetic ancestry. **f**, Contrasting pathogenic effects and hypothesized mechanism for heterozygous *GLS* repeat expansions compared with biallelic *GLS* expansion or LoF. The error bars show the 95% CIs.

with one another (Fig. 4e) and with smoking, which was associated with substantially increased somatic mutability of chromosome 2: 232.4 Mb AAAG repeats (1.33-fold (1.30–1.36) for current versus never-smokers) but not chromosome 19: 14.8 Mb AAAG repeats (Extended Data Fig. 10).

## *GLS* repeat expansion is associated with disease

The deep phenotyping of the UKB cohort offered an opportunity to search for effects of repeat expansions on a wide variety of diseases and other clinical phenotypes. Phenome-wide analyses of 67,405 repeat loci expanded in at least 5 UKB participants (for association with 6,483 disease phenotypes and 57 heritable quantitative traits) identified 7 repeat loci involved in 23 likely causal associations with quantitative phenotypes and 46 associations with disease phenotypes (Supplementary

Data 7 and 8). Associations of expansions in *C9orf72*, *TCF4* and *DMPK* with disease phenotypes reflected the known roles of these repeats in amyotrophic lateral sclerosis and frontotemporal dementia, FECD and myotonic dystrophy[1], and associations involving repeats in *AFF3* and *DIP2B* were recently reported[57]. However, the association of repeat expansions in *GLS* with a biomarker of liver function was surprising, as such expansions in *GLS* (which encodes kidney-type glutaminase) have been observed to be pathogenic in only extremely rare cases of severe, childhood-onset recessive glutaminase deficiency[58,59]. The large UKB population sample made it possible to see that heterozygous carriers of long *GLS* alleles exhibited anomalous phenotypes.

In the UKB, 139 individuals (0.03%) carried a long CAG repeat (at least 45 repeat units) within the 5′ untranslated region (5′-UTR) of *GLS* (Fig. 5a). Most of these individuals (98 out of 139) exhibited evidence

of a highly expanded allele (around 100+ repeats based on observing IRR pairs; Extended Data Fig. 1b), consistent with long *GLS* repeats being highly unstable somatically[58] and in the germline (Fig. 1b,d,e). Germline instability of the repeat increased rapidly with allele length: mid-length alleles (25–40 repeats) were already sufficiently unstable for mutations to be observable between close relatives, including a quartet of genetically inferred first cousins among whom multiple intergenerational mutations had occurred (Fig. 5b).

Repeat expansions in *GLS* were associated strongly with elevated biomarkers of liver and kidney disease ($P = 3.4 \times 10^{-15}$ for gamma-glutamyl transferase (GGT); $P = 7.3 \times 10^{-7}$ for cystatin C). These associations appeared to be driven by highly expanded alleles (beyond a threshold of around 100–200 repeat units; Fig. 5c); single-nucleotide polymorphism (SNP) and insertion–deletion (indel) pLoF variants in *GLS* did not associate with these biomarkers (Fig. 5d and Supplementary Table 8). Carriers of highly expanded alleles exhibited severalfold increased risk of liver diseases and chronic kidney disease, results that were replicated in AoU, with particularly elevated risk of stage 5 chronic kidney disease (OR = 14.0 (5.7–34.3), $P = 7.2 \times 10^{-9}$; in meta-analysis across UKB and AoU; Fig. 5e and Supplementary Table 9). These results suggest that highly expanded *GLS* repeats cause a dominant (but low penetrance) DNA-repeat disorder that is distinct from recessive glutaminase deficiency (Fig. 5f). Glutaminase deficiency is caused by biallelic impairment of *GLS* function—either by SNP/indel pLoF variants[60] or long *GLS* repeats that suppress *GLS* expression[58], both of which were associated with elevated serum glutamine in the UKB (Fig. 5c,d and Supplementary Table 8). By contrast, the effects on kidney and liver biomarkers were specific to highly expanded repeats, indicating a different pathological mechanism unrelated to GLS function, such as RNA toxicity[61] (Fig. 5f).

## Discussion

These results show how biobank WGS datasets contain abundant information about unstable DNA repeats. We observed that somatic expansion (in blood) of some repeats is under strong genetic control. Different repeats appear to be affected by a largely shared set of common alleles at DNA-repair genes, but the relative influence of these alleles varied across repeat loci, and their effect directions even varied: common modifier haplotypes at several loci appeared to have opposite effects on blood instability of the *TCF4* CAG repeat compared with other repeats, including the *HTT* CAG repeat. These results, along with the repeat locus specificity that we observed in the relative mutation rates of repeats in blood versus the germline, reinforce recent evidence of tissue specificity of genetic modifiers of *HTT* expansion in blood versus brain[18], pointing to highly complex regulation of somatic repeat expansion that varies across repeats and cell types. The modulation of genetic effects by locus-specific effects may suggest roles for locus-specific chromatinization or transcriptional dynamics and will be an interesting area for mechanistic studies.

The clear and strong differences in genetic effects on repeat expansion of different repeats and in different tissues suggest a need for care and caution in efforts to use DNA repeats in clinically accessible tissues (such as blood) to inform on the status of somatic expansion in disease-relevant tissues (such as brain). However, our results also suggest the potential for repeats that are unstable in blood to be used as biomarkers of target engagement for future expansion-slowing therapies. We identified several repeat loci at which common alleles expand in blood as humans age, at rates that are strongly influenced by genetic modifiers (for example, at *MSH3*). Future analyses using long read sequencing to identify hypermutable loci with longer alleles[10] may detect even better candidates.

The deep phenotype data available in biobank datasets also enabled us to observe evidence suggestive of a dominant DNA-repeat disorder involving highly expanded 5′-UTR repeat alleles in *GLS*, which was associated with a severalfold higher risk of kidney and liver diseases.

Large WGS cohorts provide an opportunity to identify pathogenic rare alleles that, despite their strong effects on disease risk, have not been identified to date owing to their low penetrance in families. Analyses of the phenotypic effects of common repeat variation, which we did not undertake here, may reveal subclinical phenotypes and may also resolve the question of whether intermediate-length alleles of pathogenic repeats have any beneficial effects (that could in principle cause them to persist in human populations); association analyses conducted to date[6,7] have not detected evidence of such effects.

Analysis of repeat instability in population biobanks does have several limitations. Although we could study germline mutation rates by analysing IBD among unrelated individuals, we could not assess effects of genetic variation, parental age or parent-of-origin on germline mutability, as this requires ascertaining de novo mutations[8–10]. Moreover, the short-read WGS data that we analysed provided only glimpses of somatic mutation, through observations of one or a few reads spanning shorter mutated alleles and through read-count-based evidence of expanded alleles of unknown lengths. Nonetheless, the analytical tools that we have developed here for biobank-scale WGS analysis provide a useful complement to studying repeat instability in families[8–10] and in patient cohorts using targeted sequencing techniques[18,62], and combining these approaches should provide opportunities for further discovery.

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

## Methods

Methods are provided in the Supplementary Information.

### Reporting summary

Further information on research design is available in the Nature Portfolio Reporting Summary linked to this article.

### Data availability

Summary association statistics for GWAS of somatic-expansion phenotypes are available at the NHGRI-EBI GWAS Catalog (GCST90704615 to GCST90704642). Summary statistics for association tests of repeat expansions with quantitative traits and diseases are available at https://data.broadinstitute.org/lohlab/UKB_STR_expansion_sumstats/ and Zenodo[63] (https://doi.org/10.5281/zenodo.17419996). Access to the following data resources used in this study is obtained by application: UK Biobank (http://www.ukbiobank.ac.uk/) and All of Us (https://allofus.nih.gov/). We also used the following data resources: the 1000 Genomes+H3Africa STR reference panel generated previously[3] (https://github.com/gymrek-lab/EnsembleTR), the STRipy database (https://stripy.org) and the GTEx eQTL/sQTL browser (https://gtexportal.org).

### Code availability

Code for efficiently extracting sequencing reads with high repeat content (that is, potential IRRs) from sequencing read alignment files is available at GitHub (https://github.com/poruloh/extractLongSTRs). Code implementing other analytical pipelines used in the study is also provided at GitHub (https://github.com/mhujoel/STRs). Snapshots of both repositories have been deposited at Zenodo[63] (https://doi.org/10.5281/zenodo.17419996).

63. Hujoel M. L. A. et al. Code and pheWAS data from 'Insights into DNA repeat expansions among 900,000 biobank participants'. *Zenodo* https://doi.org/10.5281/zenodo.17419996 (2025).

**Acknowledgements** We thank R. Cai and S. Browning for providing estimates of effective population size in the UKB; and B. Gorman and S. Iyengar for discussions. This research was conducted using the UKB resource under application no. 40709. M.L.A.H. was supported by a US National Institutes of Health (NIH) fellowship F32 HL160061; R.E.H. and S.A.M. by US NIH grant R01 HG006855; D.T. by US NIH training grant T32 HG002295; N.K. by US NIH training grant T32 HG002295 and fellowship F31 DE034283; R.E.M. by US NIH grant K25 HL150334; S.R. by a Swiss National Science Foundation Postdoc.Mobility fellowship (P500PB_211106); P.F.P. by ERC Starting Grant no. 850869; and P.-R.L. by US NIH grants R56 HG012698, R01 HG013110 and UM1 DA058230 and a Burroughs Wellcome Fund Career Award at the Scientific Interfaces. The funders had no role in study design, data collection and analysis, decision to publish or preparation of the manuscript. The content is solely the responsibility of the authors and does not necessarily represent the official views of the National Institutes of Health. The All of Us Research Program is supported by the NIH, Office of the Director: Regional Medical Centers: 1 OT2 OD026549; 1 OT2 OD026554; 1 OT2 OD026557; 1 OT2 OD026556; 1 OT2 OD026550; 1 OT2 OD026552; 1 OT2 OD026553; 1 OT2 OD026548; 1 OT2 OD026551; 1 OT2 OD026555; IAA no. AOD 16037; Federally Qualified Health Centers: HHSN 263201600085U; Data and Research Center: 5 U2C OD023196; Biobank: 1 U24 OD023121; The Participant Center: U24 OD023176; Participant Technology Systems Center: 1 U24 OD023163; Communications and Engagement: 3 OT2 OD023205; 3 OT2 OD023206; and Community Partners: 1 OT2 OD025277; 3 OT2 OD025315; 1 OT2 OD025337; and 1 OT2 OD025276. The All of Us Research Program would not be possible without the partnership of its participants.

**Author contributions** M.L.A.H., R.E.H., S.A.M. and P.-R.L. designed the study. M.L.A.H. and P.-R.L. wrote analysis scripts, conducted and interpreted analyses, and wrote the manuscript. D.T. constructed burden masks for analyses of protein-coding variants. R.E.H., D.T., N.K., R.E.M., S.R., P.F.P. and S.A.M. helped with the interpretation of the analyses. R.E.H. and S.A.M. reviewed and edited the manuscript.

**Competing interests** The authors declare no competing interests.

**Additional information**
**Correspondence and requests for materials** should be addressed to Margaux L. A. Hujoel or Po-Ru Loh.

**a**

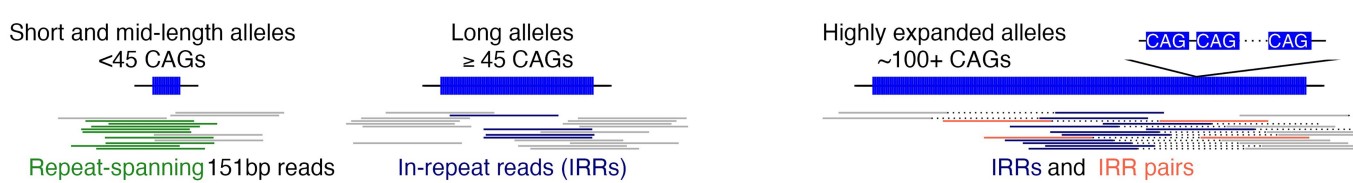

| Short and mid-length alleles <45 CAGs | Long alleles ≥ 45 CAGs | Highly expanded alleles ~100+ CAGs |
|---|---|---|

Repeat-spanning 151bp reads | In-repeat reads (IRRs) | IRRs and IRR pairs

**b**

| Locus (Mb) | Gene(s) | Genic context | Short and mid-length alleles | | | Long alleles | | |
|---|---|---|---|---|---|---|---|---|
| | | | Frequencies of common short allele lengths | Per-generation mutation rate as a function of allele length | | UKB participants with in-repeat reads | | % with IRR pair given IRR |
| | | | | Expansion +1 CAG | Contraction −1 CAG | ≥ 1 IRR (45+ CAGs) | ≥ 1 IRR pair | |
| chr2:190.9 | *GLS* | 5' UTR | 8 — 24 | 0.005 (10–24) | 0.001 (10–24) | 139 | 98 | 70.5% |
| chr3:171.5 | | intergenic | 8 — 26 | <0.001 (10–26) | <0.001 (10–26) | 306 | 19 | 6.2% |
| chr4:3.1 | *HTT* | CDS | 9 — 30 | 0.003 (10–30) | 0.006 (10–30) | 16 | 0 | 0% |
| chr4:30.7 | | intergenic | 10 — 22 | <0.001 (10–22) | <0.001 (10–22) | 56 | 0 | 0% |
| chr5:146.9 | *PPP2R2B* | 5' UTR | 9 — 18 | <0.001 (10–18) | <0.001 (10–18) | 7 | 1 | 14.3% |
| chr10:76.0 | *LRMDA* | intron | 10 — 27 | 0.001 (10–27) | 0.003 (10–27) | 44 | 9 | 20.4% |
| chr12:6.9 | *ATN1* | CDS | 8 — 22 | 0.003 (10–22) | 0.001 (10–22) | 5 | 0 | 0% |
| chr13:35.5 | *MAB21L1;NBEA* | 5' UTR;intron | 6 — 27 | 0.005 (10–27) | 0.002 (10–27) | 194 | 72 | 37.1% |
| chr13:70.1 | *ATXN8OS* | lncRNA | 6 — 25 | 0.009 (10–25) | <0.001 (10–25) | 7736 | 2022 | 26.1% |
| chr13:112.9 | | intergenic | 8 — 23 | <0.001 (13–23) | 0.001 (13–23) | 117 | 10 | 8.6% |
| chr14:92.1 | *ATXN3* | CDS | 8 — 22 | <0.001 (12–22) | <0.001 (12–22) | 33 | 1 | 3% |
| chr16:73.5 | *ENSG00000260848* | lncRNA | 10 — 25 | 0.003 (10–25) | <0.001 (10–25) | 22 | 3 | 13.6% |
| chr17:51.8 | *CA10* | intron | 7 — 28 | <0.001 (10–24) | <0.001 (10–24) | 137673 | 1069 | 0.8% |
| chr18:55.6 | *TCF4* | intron | 12 — 30 | 0.002 (12–30) | 0.002 (12–30) | 42004 | 7992 | 19% |
| chr19:45.8 | *DMPK* | 3' UTR | 5 — 29 | 0.006 (10–29) | 0.005 (10–29) | 438 | 44 | 10.1% |

**Extended Data Fig. 1 | CAG trinucleotide repeat expansions in UK Biobank (n = 490,416). a,** Types of short-read evidence of repeat alleles of different lengths. **b,** Fifteen CAG repeat loci at which at least five UKB participants carried long alleles (≥ 45 repeat units). Repeat expansions in genes highlighted in red are known to be pathogenic. For each locus, the length distribution of common short alleles (≤ 30 repeat units) is shown; the length range is indicated below each histogram, red bars denote interrupted repeat alleles, and blue bars denote alleles with alternate flank sequences. For each common allele between 10–30 repeat units, estimated rates of intergenerational expansion and contraction (by ±1 unit) are plotted as a function of allele length; the mutation rate of the longest plotted allele is indicated at the end of each curve. For long alleles, counts of UKB participants with at least one in-repeat read (IRR) or IRR pair are shown.

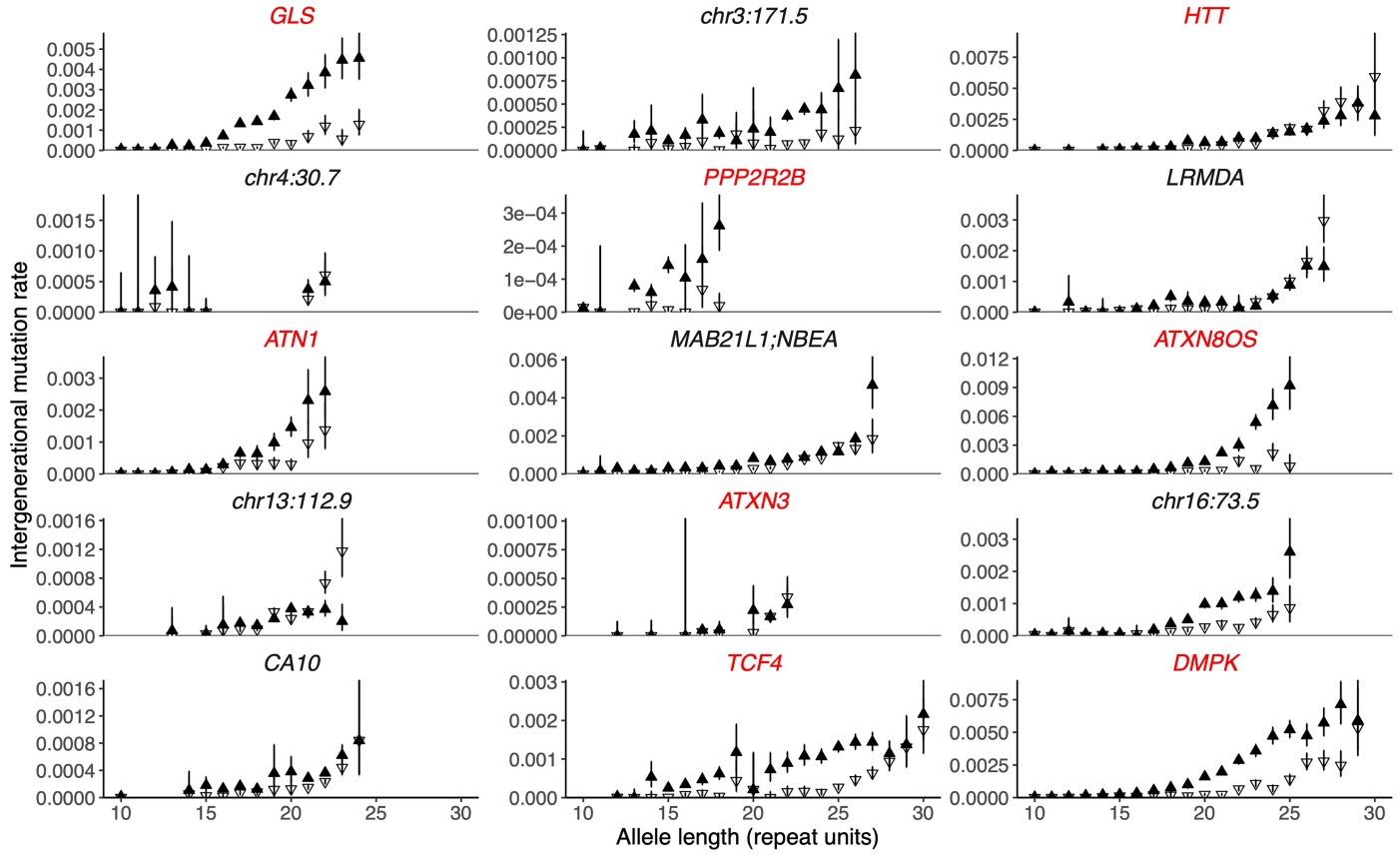

**Extended Data Fig. 2 | Germline mutation rates of CAG repeats.** Estimated per-generation mutation rates of germline expansion (+1 repeat unit) and contraction (−1 repeat unit) of 15 repeat loci in UKB. Repeat expansions in genes highlighted in red are known to be pathogenic. Sample sizes are provided in Supplementary Table 3. Error bars, 95% CIs.

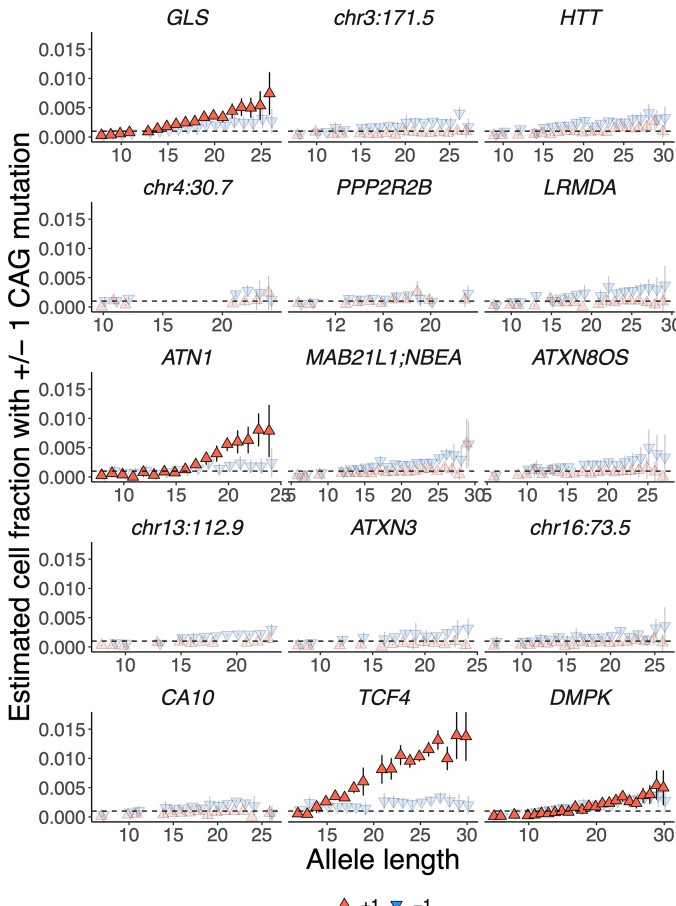

**Extended Data Fig. 3 | Estimated fraction of blood cells with a ±1 repeat unit mutation as a function of allele length.** For each repeat locus, frequencies of somatic +1 unit expansions and −1 unit contractions were estimated among all UKB participants heterozygous for a repeat allele of a given length. For most repeat loci, estimated contraction frequencies exceeded estimated expansion frequencies and these estimates did not significantly increase with age, indicating that even after applying our filtering approach, the estimates still largely reflected residual PCR stutter error rather than somatic mutation. For four repeat loci (*GLS*, *ATN1*, *TCF4*, and *DMPK*), estimated frequencies of +1 unit expansions associated significantly with age; these estimates are indicated with darker markers. These somatic repeat expansions did not associate with mosaic chromosomal alterations or telomere length, indicating that their associations with age were unlikely to be confounded by clonal expansions in blood. We did not observe a significant age association for *HTT* repeat expansions, probably because the shorter *HTT* alleles studied here (≤30 repeats) expand in blood at rates below our power to detect an age effect in UKB. The total number of haplotypes analysed varied by locus (minimum n = 29,976; maximum n = 506,831). Error bars, 95% CIs.

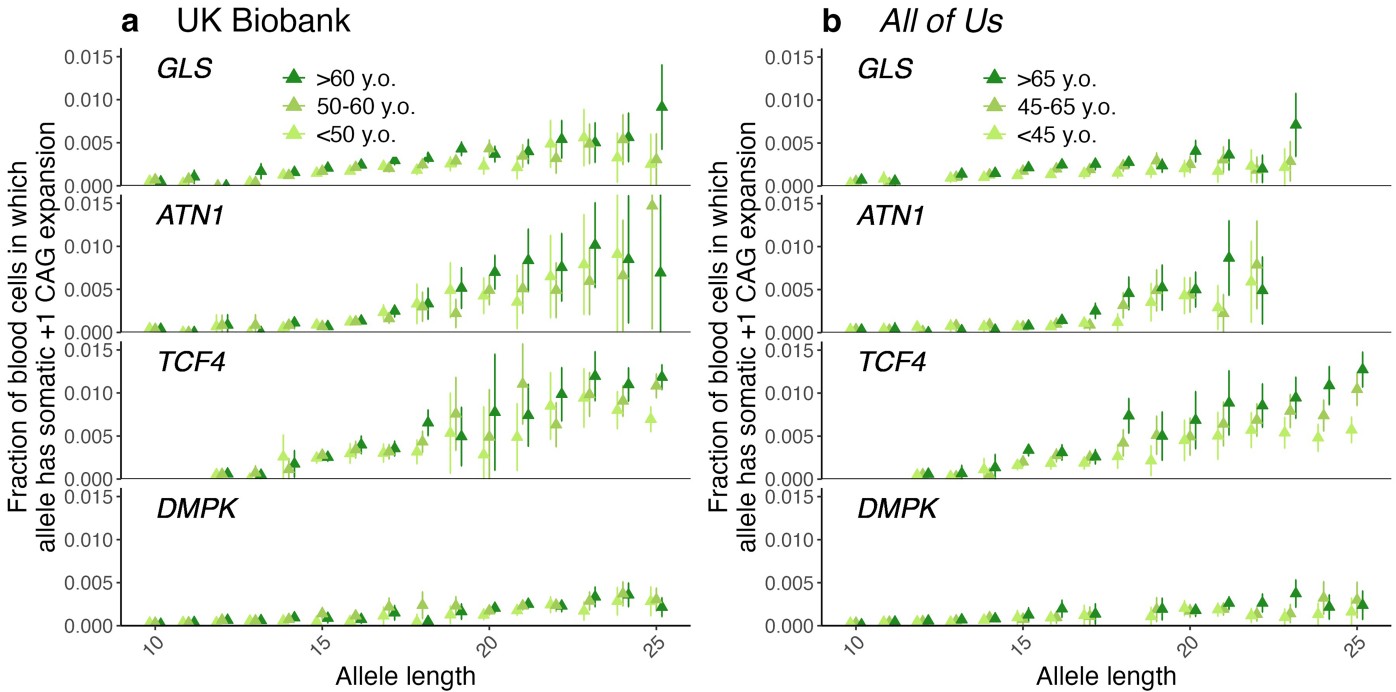

**Extended Data Fig. 4 | Estimated fractions of blood cells with +1 repeat unit expansions increase with age for four CAG repeat loci.** Estimated frequencies of somatic +1 unit expansions among individuals heterozygous for a repeat allele of a given length are shown for three age strata of UKB (**a**) and AoU v7 (**b**). The total number of haplotypes analysed varied by locus (minimum n = 206,893, maximum n = 289,119 in UKB; minimum n = 104,933, maximum n = 139,191 in AoU v7). Error bars, 95% CIs.

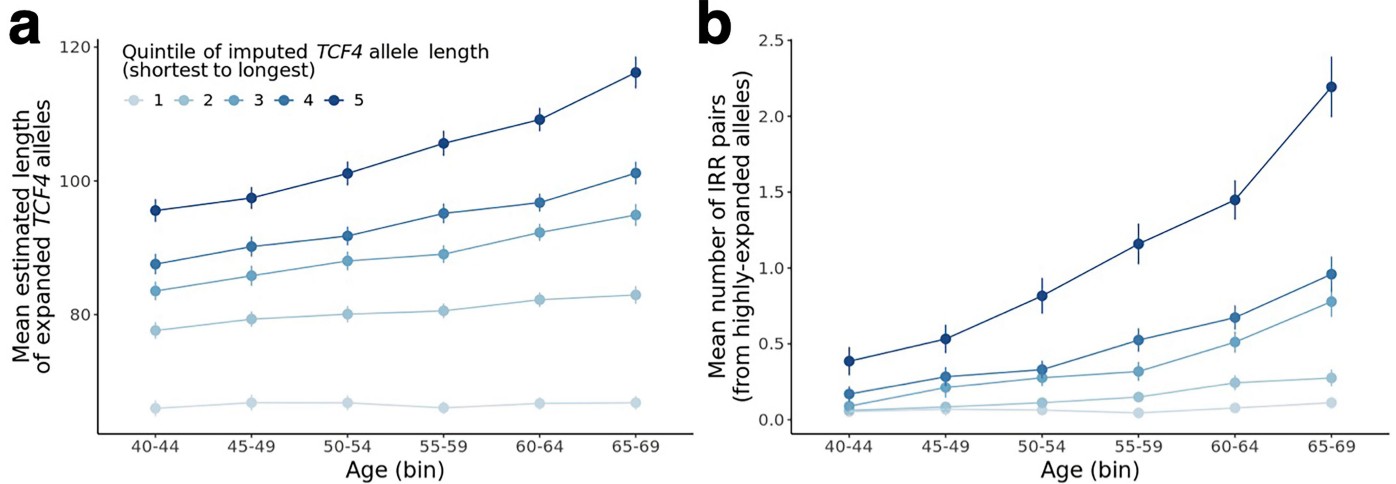

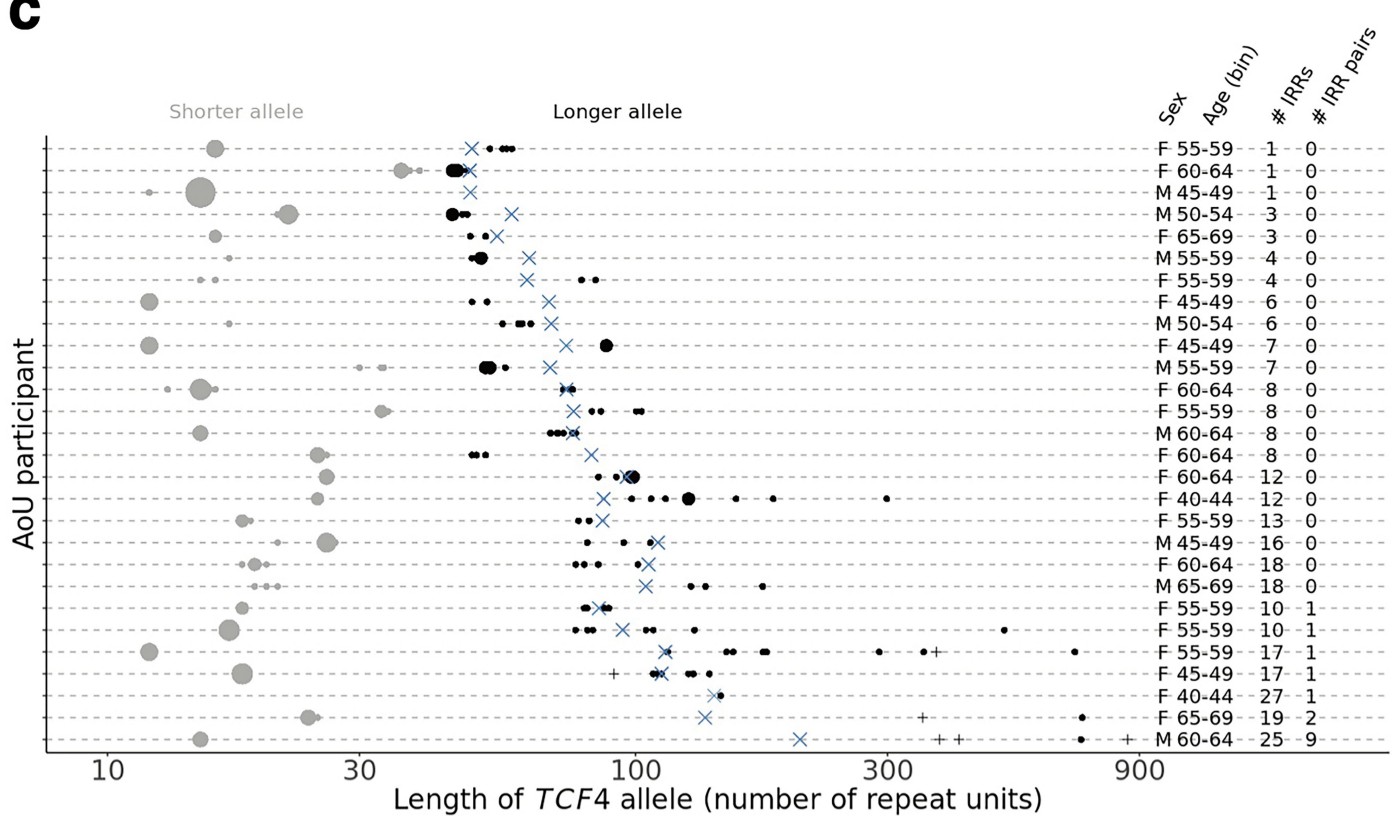

Long read type ● spanning + non-spanning    Number of long reads · 1 ● 5 ● 10

× Estimated length from short reads

**Extended Data Fig. 5 | Somatic instability of long *TCF4* repeat alleles.**
**a**, Mean estimated length (in repeat units) of long *TCF4* alleles (≥ 45 repeat units) in UKB participants of different ages. Heterozygous carriers of long *TCF4* alleles were first stratified into quintiles of imputed *TCF4* allele length, a proxy for inherited allele length. **b**, Mean number of IRR pairs observed per UKB participant heterozygous for a long *TCF4* allele, again stratified by imputed *TCF4* allele length and by age. In both **a** and **b**, analyses were restricted to 38,558 individuals carrying no other long CAG repeat except possibly in *CA10* (such that IRR pairs could be assumed to have originated from *TCF4*). Error bars, 95% CIs. **c**, *TCF4* allele lengths directly measured from long-read sequencing of carriers of long alleles

in AoU. Each horizontal line corresponds to a single AoU participant; black markers indicate repeat lengths observed in long reads that span the *TCF4* repeat (dots) or partially overlap the repeat (pluses, which lower-bound allele lengths), while blue crosses indicate allele lengths estimated from short-read WGS. Long *TCF4* alleles exhibit somatic mosaicism, with alleles sometimes varying in length by hundreds of repeat units within blood cells from the same individual, indicating high somatic instability. We have received an exception from the *All of Us* Resource Access Board to disseminate participants counts less than 20.

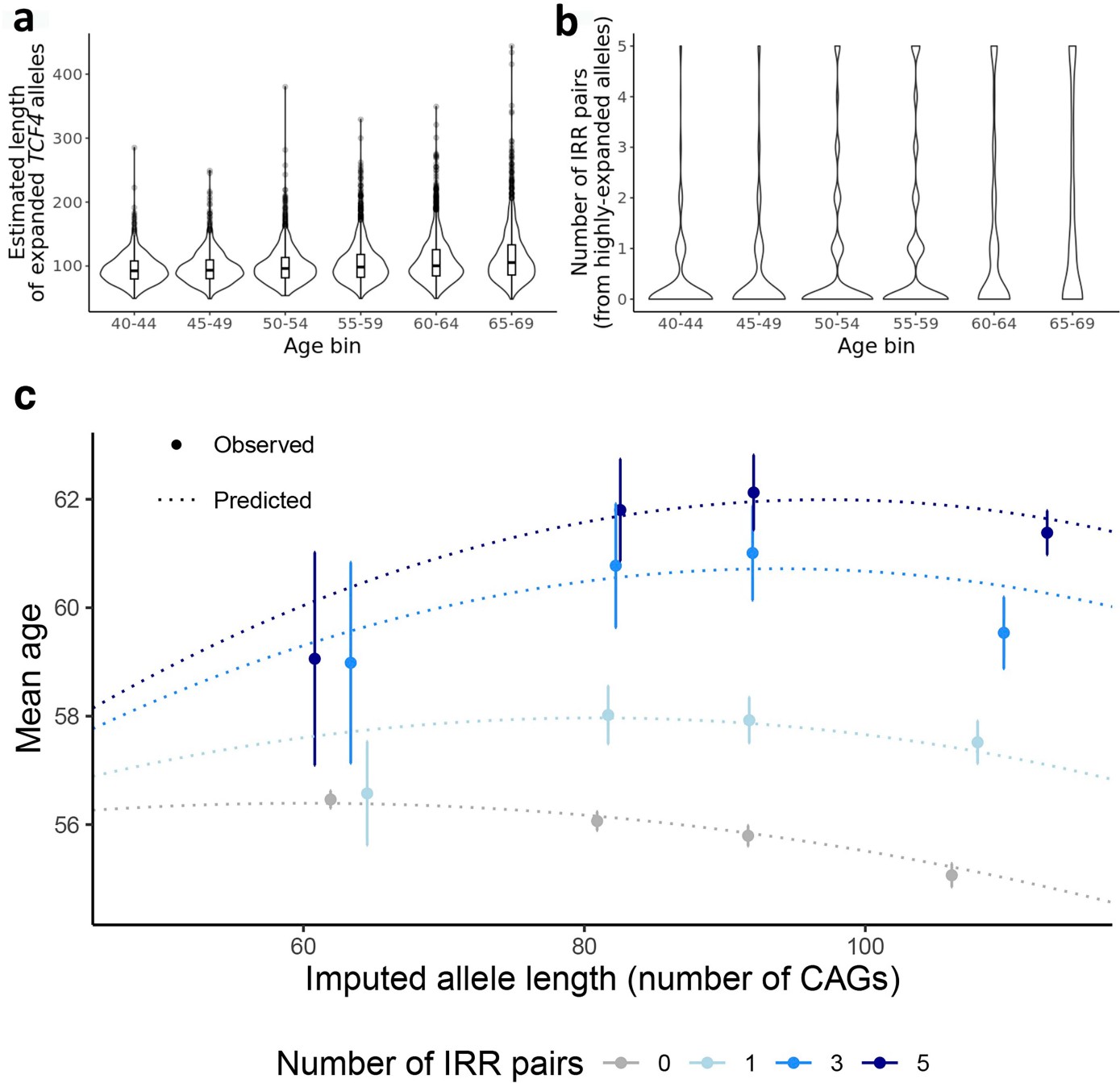

**Extended Data Fig. 6 | Distributions of *TCF4* allele length metrics and relationship between age, imputed *TCF4* allele length, and count of IRR pairs. a**,**b**, Distributions of individual-level measurements of estimated lengths of long (≥45-repeat) *TCF4* alleles (panel **a**) and counts of IRR pairs observed in UKB WGS data (panel **b**, capped at 5) in 5-year age tranches of individuals in the highest quintile of imputed allele length (a metric that captures relative lengths of inherited alleles). While there is a strong statistical signal that these length metrics increase with age in UKB (Extended Data Fig. 5a, b), the individual-level data are noisy and only weakly informative of somatic expansion. Sample sizes are available in the legend of Extended Data Fig. 5. Median and interquartile ranges are shown within violin plots. **c**, Mean age (error bars, 95% CIs) among individuals stratified by IRR pair count and by imputed allele length (*x*-values of dots are means per quartile). For each stratum of IRR pair count shown in the figure (count = 0, 1, 3, 5), mean age predicted by our model is plotted as a function of imputed allele length (dotted lines). This prediction function served as a way to transform IRR pair count and imputed allele length into a somatic-expansion phenotype that improved GWAS power.

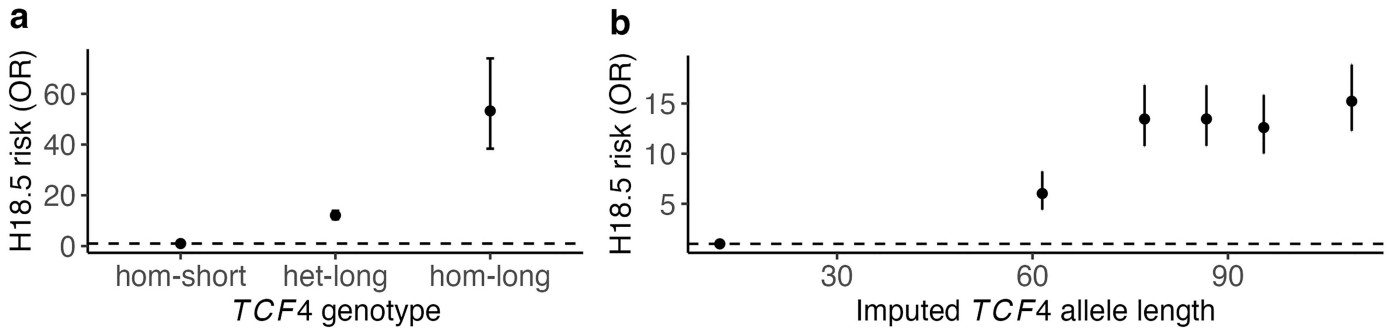

**Extended Data Fig. 7 | Risk of hereditary corneal dystrophies (H18.5, under which FECD is classified) conferred by long *TCF4* repeats. a**, Risk of H18.5 among individuals heterozygous (respectively, homozygous) for a long *TCF4* allele relative to individuals who do not carry a long *TCF4* allele (n = 458,085). **b**, Risk of H18.5 among individuals heterozygous for a long *TCF4* allele, stratified into quintiles of imputed *TCF4* allele length (a proxy for inherited allele length; n = 454,495 including reference individuals without a long allele). Odds ratios were estimated in UKB participants of EUR genetic ancestry using logistic regression, controlling for age and sex. Error bars, 95% CIs.

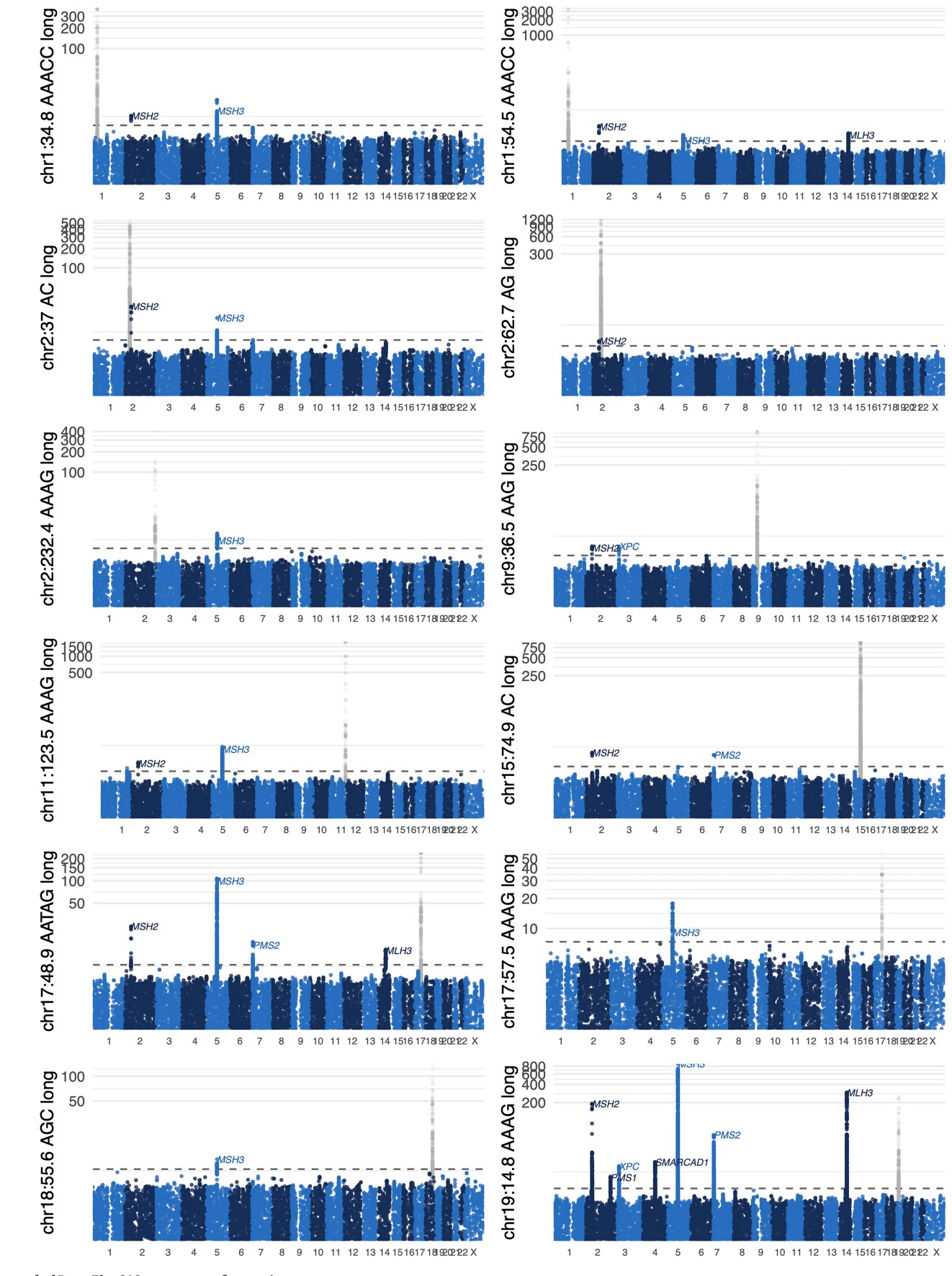

**Extended Data Fig. 8** | See next page for caption.

**Extended Data Fig. 8 | Manhattan plots for GWAS of unstable repeats for the 12 of 17 repeat loci with at least one GWAS hit that is not the repeat locus itself (n = 419,013).** Associations of common variants (MAF > 1%) are plotted, and the y-axis shows $-\log_{10}$ p-values, plotted on a log scale. Associations of variants proximal to the repeat loci are shown in grey as they could reflect imperfect control for lengths of inherited alleles. Effect sizes and association statistics of index variants are available in Supplementary Data 3.

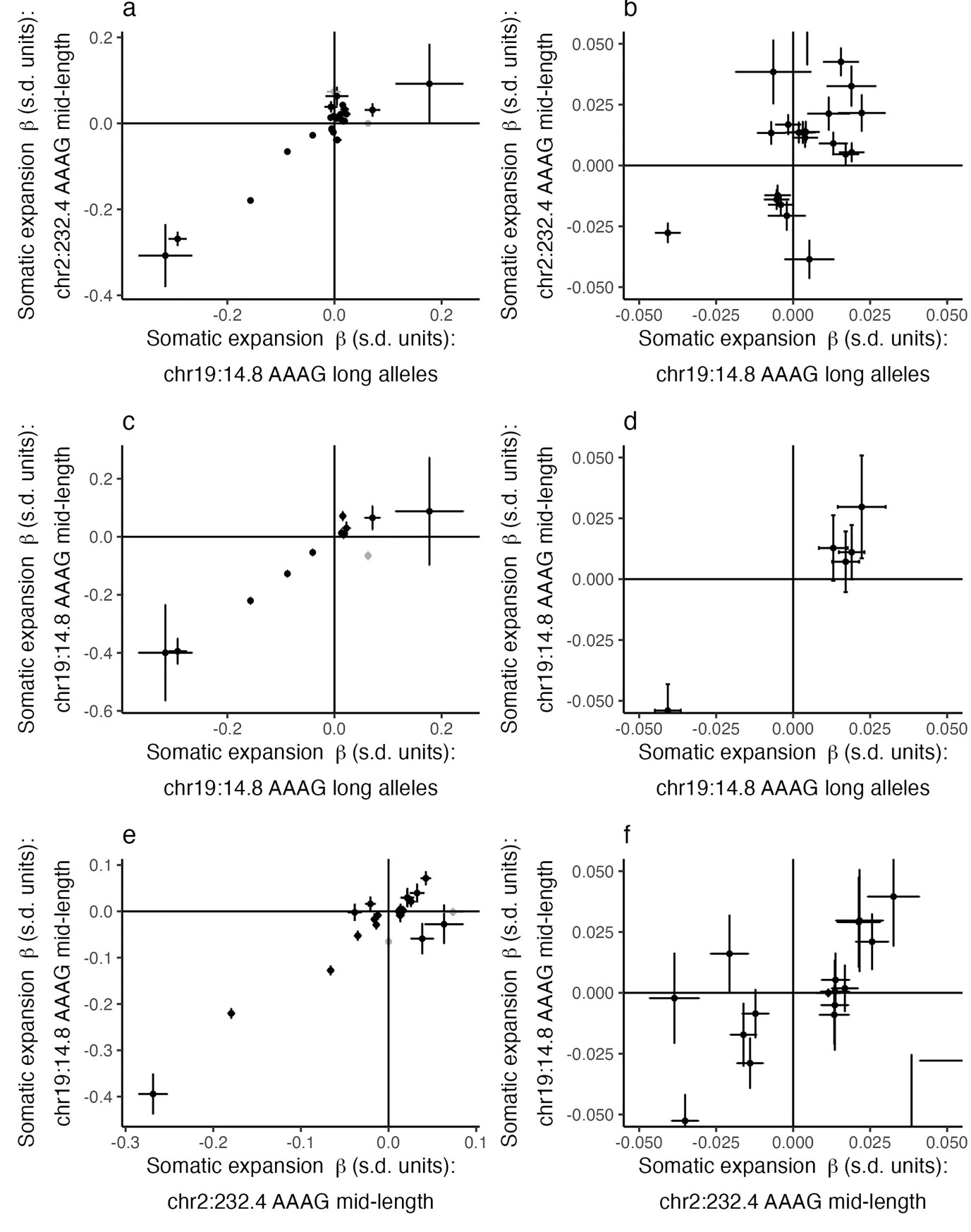

**Extended Data Fig. 9** | See next page for caption.

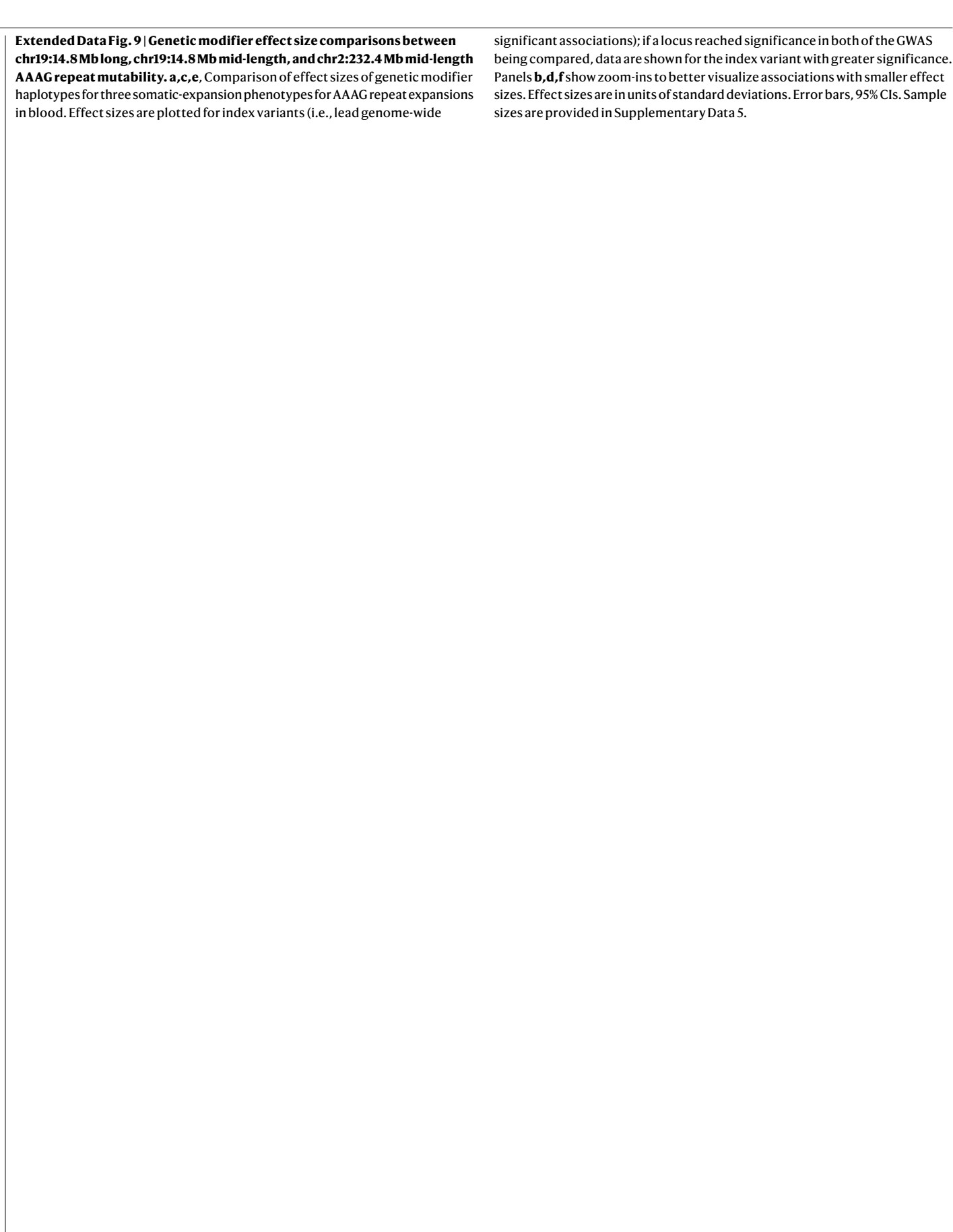

**Extended Data Fig. 9 | Genetic modifier effect size comparisons between chr19:14.8 Mb long, chr19:14.8 Mb mid-length, and chr2:232.4 Mb mid-length AAAG repeat mutability. a,c,e**, Comparison of effect sizes of genetic modifier haplotypes for three somatic-expansion phenotypes for AAAG repeat expansions in blood. Effect sizes are plotted for index variants (i.e., lead genome-wide significant associations); if a locus reached significance in both of the GWAS being compared, data are shown for the index variant with greater significance. Panels **b,d,f** show zoom-ins to better visualize associations with smaller effect sizes. Effect sizes are in units of standard deviations. Error bars, 95% CIs. Sample sizes are provided in Supplementary Data 5.

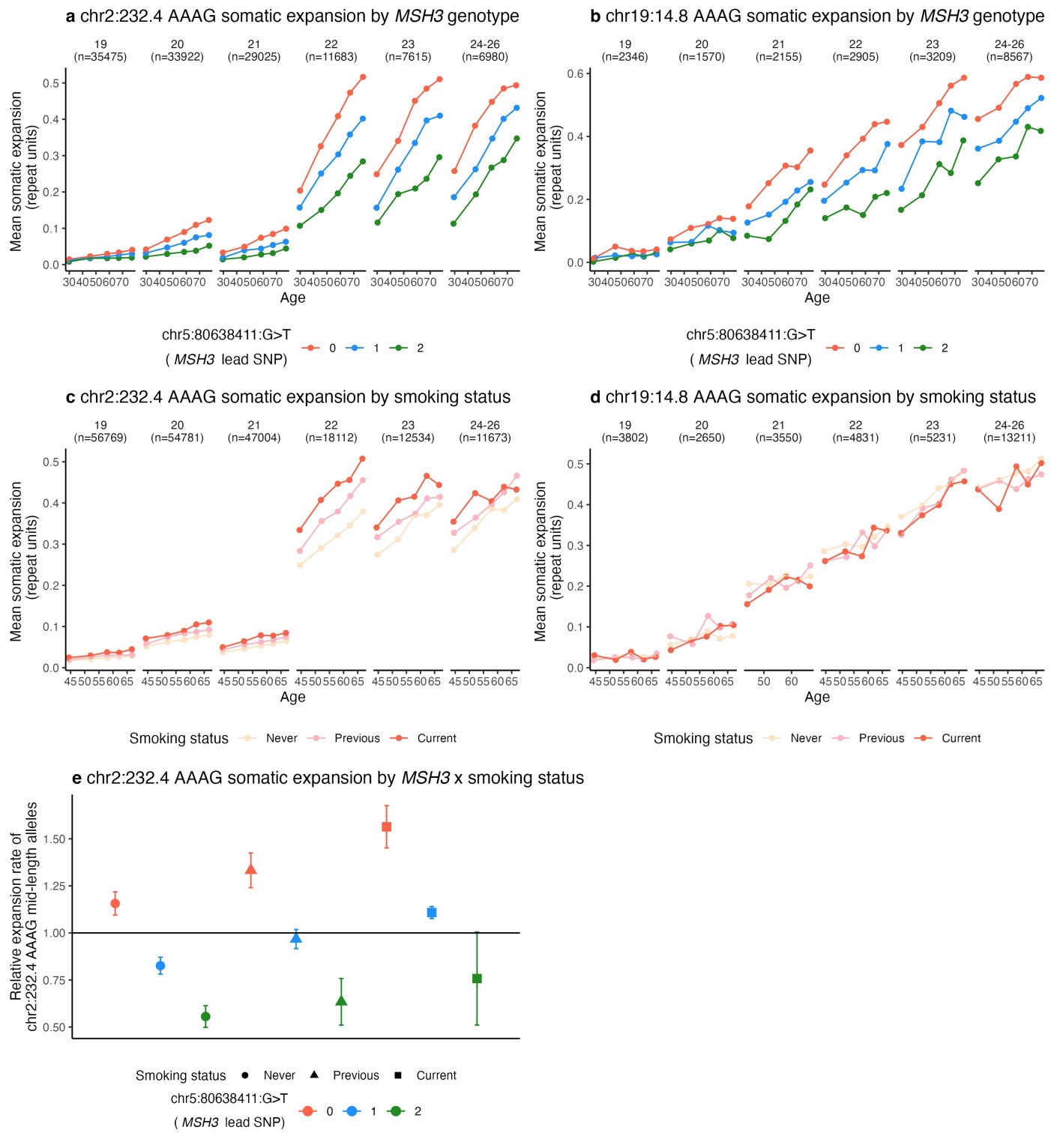

**Extended Data Fig. 10 | Somatic mutability of mid-length chr2:232.4 Mb and chr19:14.8 Mb AAAG repeat alleles. a,b** Mean amount of somatic expansion of mid-length chr2:232.2 Mb and chr19:14.8 Mb AAAG repeat alleles (i.e., average increase in repeat length across spanning reads) observed in AoU participants who inherited one allele of the indicated length (19, 20, 21, 22, 23, 24, 25, or 26 repeat units). Plots are stratified by *MSH3* genotype (chr5:80638411 G > T). **c,d** Mean amount of somatic expansion of mid-length chr2:232.2 Mb and chr19:14.8 Mb AAAG repeat alleles observed in UKB participants who inherited one allele of the indicated length; plots are stratified by smoking status. **e**, Relative expansion rates of mid-length chr2:232.2 Mb AAAG repeats in subsets of UKB participants with different pairwise combinations of smoking status and *MSH3* genotype (chr5:80638411 G > T). Sample sizes are shown within the figure panels. Error bars, 95% CIs.

# Reporting Summary

## Statistics

For all statistical analyses, confirm that the following items are present in the figure legend, table legend, main text, or Methods section.

| n/a | Confirmed | |
|---|---|---|
| ☐ | ☒ | The exact sample size (*n*) for each experimental group/condition, given as a discrete number and unit of measurement |
| ☐ | ☒ | A statement on whether measurements were taken from distinct samples or whether the same sample was measured repeatedly |
| ☐ | ☒ | The statistical test(s) used AND whether they are one- or two-sided<br>*Only common tests should be described solely by name; describe more complex techniques in the Methods section.* |
| ☐ | ☒ | A description of all covariates tested |
| ☐ | ☒ | A description of any assumptions or corrections, such as tests of normality and adjustment for multiple comparisons |
| ☐ | ☒ | A full description of the statistical parameters including central tendency (e.g. means) or other basic estimates (e.g. regression coefficient) AND variation (e.g. standard deviation) or associated estimates of uncertainty (e.g. confidence intervals) |
| ☐ | ☒ | For null hypothesis testing, the test statistic (e.g. *F*, *t*, *r*) with confidence intervals, effect sizes, degrees of freedom and *P* value noted<br>*Give P values as exact values whenever suitable.* |
| ☒ | ☐ | For Bayesian analysis, information on the choice of priors and Markov chain Monte Carlo settings |
| ☒ | ☐ | For hierarchical and complex designs, identification of the appropriate level for tests and full reporting of outcomes |
| ☐ | ☒ | Estimates of effect sizes (e.g. Cohen's *d*, Pearson's *r*), indicating how they were calculated |

*Our web collection on statistics for biologists contains articles on many of the points above.*

## Software and code

Policy information about availability of computer code

| Data collection | None. |
|---|---|
| Data analysis | Code has been provided on GitHub (https://github.com/mhujoel/STRs and https://github.com/poruloh/extractLongSTRs) and has been deposited on Zenodo (10.5281/zenodo.17419996). We additionally used the following open-source software packages: samtools 1.15.1; BOLT-LMM 2.4.2; R 3.6.3; susieR 0.12.35; SHAPEIT5 v5.1.1; METAL 2020-05-05. |

For manuscripts utilizing custom algorithms or software that are central to the research but not yet described in published literature, software must be made available to editors and reviewers. We strongly encourage code deposition in a community repository (e.g. GitHub). See the Nature Portfolio guidelines for submitting code & software for further information.

## Data

Policy information about availability of data

All manuscripts must include a data availability statement. This statement should provide the following information, where applicable:
- Accession codes, unique identifiers, or web links for publicly available datasets
- A description of any restrictions on data availability
- For clinical datasets or third party data, please ensure that the statement adheres to our policy

Summary association statistics for GWAS of somatic-expansion phenotypes are available at the NHGRI-EBI GWAS Catalog (GCST90704615 to GCST90704642). Summary statistics for association tests of repeat expansions with quantitative traits and diseases are available at https://data.broadinstitute.org/lohlab/

# Research involving human participants, their data, or biological material

Policy information about studies with human participants or human data. See also policy information about sex, gender (identity/presentation), and sexual orientation and race, ethnicity and racism.

| | |
|---|---|
| Reporting on sex and gender | We included sex as a covariate in analyses. |
| Reporting on race, ethnicity, or other socially relevant groupings | We used the inferred genetic ancestry of each participant (uses their coordinates along the top 20 genetic principal components; see Methods) to define the primary analysis set. |
| Population characteristics | Prospective cohort study (~500,000 individuals from across the United Kingdom); individuals were between 40 and 69 years old at recruitment (Sudlow et al. 2015 PLOS Medicine). |
| Recruitment | Recruitment into UK Biobank has been described previously (Sudlow et al. 2015 PLOS Medicine). |
| Ethics oversight | This research complies with all relevant ethical regulations. The study protocol was determined to be not human subjects research by the Broad Institute Office of Research Subject Protection and the Partners HealthCare Human Research Committee (as all data analyzed were previously collected and de-identified). |

Note that full information on the approval of the study protocol must also be provided in the manuscript.

# Field-specific reporting

Please select the one below that is the best fit for your research. If you are not sure, read the appropriate sections before making your selection.

☒ Life sciences  ☐ Behavioural & social sciences  ☐ Ecological, evolutionary & environmental sciences

For a reference copy of the document with all sections, see nature.com/documents/nr-reporting-summary-flat.pdf

# Life sciences study design

All studies must disclose on these points even when the disclosure is negative.

| | |
|---|---|
| Sample size | We used genome sequencing data from UK Biobank and All of Us, which were the largest WGS cohorts available to us and provided ample statistical power to identify genetic modifiers of somatic repeat expansion. Our GWAS on somatic expansion of TCF4 included 40,231 UK Biobank participants and 8,217 All of Us participants. Sample sizes for instability analyses of other STRs are reported in Supplementary Data 5. Phenotypic associations of long repeat expansions were computed in a set of 421,364 unrelated individuals of EUR genetic ancestry in UK Biobank. |
| Data exclusions | We excluded individuals who had withdrawn at the time of our study. We restricted our GWAS on somatic expansion of TCF4 to long-allele carriers and those with EUR genetic ancestry; within All of Us, we additionally restricted to an unrelated set of individuals and required individuals have age >= 40. |
| Replication | We analyzed the All of Us data set to replicate the key (K76 and CKD) GLS associations identified in UK Biobank; both the K76 and CKD associations replicated. |
| Randomization | Not applicable to our study; participants were analyzed together and not allocated into groups. |
| Blinding | Not applicable to our study; all data were previously collected, and participants were not allocated into groups. |

# Reporting for specific materials, systems and methods

We require information from authors about some types of materials, experimental systems and methods used in many studies. Here, indicate whether each material, system or method listed is relevant to your study. If you are not sure if a list item applies to your research, read the appropriate section before selecting a response.

## Materials & experimental systems

| n/a | Involved in the study |
|-----|----------------------|
| ☒ ☐ | Antibodies |
| ☒ ☐ | Eukaryotic cell lines |
| ☒ ☐ | Palaeontology and archaeology |
| ☒ ☐ | Animals and other organisms |
| ☒ ☐ | Clinical data |
| ☒ ☐ | Dual use research of concern |
| ☒ ☐ | Plants |

## Methods

| n/a | Involved in the study |
|-----|----------------------|
| ☒ ☐ | ChIP-seq |
| ☒ ☐ | Flow cytometry |
| ☒ ☐ | MRI-based neuroimaging |

## Plants

| | |
|---|---|
| Seed stocks | *Report on the source of all seed stocks or other plant material used. If applicable, state the seed stock centre and catalogue number. If plant specimens were collected from the field, describe the collection location, date and sampling procedures.* |
| Novel plant genotypes | *Describe the methods by which all novel plant genotypes were produced. This includes those generated by transgenic approaches, gene editing, chemical/radiation-based mutagenesis and hybridization. For transgenic lines, describe the transformation method, the number of independent lines analyzed and the generation upon which experiments were performed. For gene-edited lines, describe the editor used, the endogenous sequence targeted for editing, the targeting guide RNA sequence (if applicable) and how the editor was applied.* |
| Authentication | *Describe any authentication procedures for each seed stock used or novel genotype generated. Describe any experiments used to assess the effect of a mutation and, where applicable, how potential secondary effects (e.g. second site T-DNA insertions, mosiacism, off-target gene editing) were examined.* |

