## [Peer Review file · Nature]

Insights into DNA repeat expansions among 900,000 biobank participants

Corresponding Author: Dr Margaux Hujuel

Version 0:

Reviewer comments:

Referee #1

(Remarks to the Author)

This study investigated the germline and somatic instability of 15 expansion-prone CAG repeat loci in >700,000 individuals from the UK Biobank and the All of Us consortium. This is, to my knowledge, the largest cohort used to study this question. The authors used sophisticated novel methods to estimate germline mutation rates from population data by inferring the genotypes of common ancestors based on shared IBD segments between two individuals. The estimated germline mutation rates for the 15 loci follow the known rules of mutation rate dependence on allele lengths and repeat interruptions. For refining the search for somatic mutations, the authors present a novel method for filtering out potential PCR error reads using base quality signatures. They profile somatic instability in 15 CAG loci, with four showing an association with the individual's age and allele lengths. They performed a GWAS identifying genetic modifiers of somatic instability in the highly unstable TCF4 locus. Variations in DNA repair-associated genes associated with the somatic instability at the HTT locus are also shown to be associated with somatic instability of the TCF4 locus. Interestingly, the genetic variants in MSH3 and FAN1 positively associated with somatic instability at the HTT locus in the brain are observed to be negatively associated with TCF4 instability in blood. Missense variation in the MSH2 locus strongly associated with somatic instability of the HTT locus in blood tissue is not associated with TCF4 somatic instability. These intriguing findings shed light on the potential tissue-specific and locus-specific roles of DNA repair mechanisms, particularly at STR loci. With available clinical data, the authors further probe the association of repeat instability in the GLS locus, a gene known to be associated with glutaminase deficiency and growth defects, to be associated with increased risk of kidney and liver diseases.

This study presented innovative methods and intriguing results, however, the study is limited to a small set of 15 CAG loci located in likely conserved genic regions. We wondered if the methods could be adapted for STR loci with different sequences and locus characteristics. Are the methods adequately reusable and scalable for a larger set of loci, more specifically for a genome-wide STR analysis with ~1 million loci? Also, we would have appreciated a more refined comparison of the intergenerational mutation rates against the conventional germline mutation rates based on family studies.

The study is a one-of-a-kind analysis using the largest collection of biobank datasets ever used to investigate STR mutation dynamics and associated genetic modifiers with the use of innovative methods. The authors also present interesting novel insights into the somatic mutation dynamics with tissue-specific and location-specific involvement of different DNA repair pathway genes. The limitation of the study was the selection of a small set of STR loci with limited sequence and location characteristics but this limitation does not undermine the novelty of the study.

Major comments

- 1 Given how specific the analysis is to only 15 CAG loci, the title is too broad. Please adjust it to reflect the scope.
- 2 The authors considered only 15 STRs with the motif CAG for the analysis, which represents Only a subset of the known CAG Mendelian disease loci. The manuscript does not sufficiently address the justification for this narrow selection criterion. We would recommend including other STR motifs, particularly those associated with known expansion disorders, which could provide insights into the polymorphisms of additional STR loci. We would additionally suggest including some loci

located in non-transcribed regions to serve as controls since they would presumably not show tissue-specific patterns. While the inclusion of these regions would have been interesting, we acknowledge it would be a substantial computational undertaking and note that this limitation does not critically undermine the study.

3 “requiring mate sequences to map to loci that contain commonly-polymorphic CAG repeats” Can you elaborate on how this data from Ziaei Jam, H. et al. 2023 was used? Did it help to define the set of initial loci? Please clarify the total number of CAG loci in the human genome and the number of them considered polymorphic for this study. How many loci/bins contained IRRs that were discarded because they didn’t meet this criteria?

4 Reads with >45 occurrences of the CAG motif are designated as IRRs and serve as a proxy for repeat expansions. The threshold length for considering an expansion reasonably depends on the reference allele length or the major allele length. For instance, HTT CAG repeat becomes pathogenic beyond the threshold length of 36 units. Have the authors explored lowering the threshold for defining repeat expansions?

5 Also, the authors identified IRRs based on only the reads that mapped to TCF4. Though it is shown that more than 95% of CAG IRRs align to this region, there are existing efficient tools that search for IRRs without making this assumption (e.g. ExpansionHunter Denovo, STRling, superSTR). Please justify the development of a new tool for this purpose and estimate the potential risks of missing these ~5% of reads.

6 Please justify the choice not to use an established STR genotyper such as HipSTR, GangSTR, ExpansionHunter etc. for genotyping repeat alleles <30 units with good spanning read coverage. In section 2.4 of the supplementary note (page 10 line 6) authors write “analyzing allele length discordances among close relatives in UKB based on DRAGEN genotyping”. This would presumably mean ExpansionHunter was used for some analyses, yet there was a description of a novel genotyping method. Please clarify.

7 Please include benchmarking of the method for detecting repeat interruptions.

8 The authors present a novel and sophisticated method for estimating germline mutation rates of STR loci using population genomes. IBD segments from individual pairs are used to estimate the TMRCA and the ancestral alleles are predicted based on an outgroup analysis. Could authors include statistics regarding the number of loci/samples with matching IBD segments within the selected criteria of >5cM and >0.5cM on either flank of the repeat? What is the average and range of generations represented at each of these thresholds? Do these vary by locus or even allele? The authors mentioned “if this consensus allele also matches one or both of the alleles in the IBD pair” in page 7 line 16 of the supplementary note indicating that the predicted allele of the ancestor should match one or both the alleles in the IBD pair. Please elaborate on the importance of this filter.

9 In the mutation rate calculation using TMRCA (formula mentioned in page 8 line 10), a single mutation is assumed to have occurred. Could you please discuss the possibility of multiple and back mutations and if this may lead to a conservative mutation rate estimate?

10 How did your mutation rates compare with prior studies that directly measured de novo mutations in trios/families? In general, and specifically triplet-repeat specific rates.

11 The HTT repeat locus was not observed to have somatic mutation rates associated with age. Can you please elaborate on why?

12 We were unable to fully assess the reusability and efficiency of the novel computational methods. Please provide a more detailed readme file for the software including usage information and considerations. Please provide sample input files for testing the submitted R-scripts.

13 Considering the potential broader applicability of the methods to similar studies, have the authors considered packaging the script into reusable tools?

Minor comments

1 The introduction uses an older citation for tandem repeat mutation rates (ref 13). Some of the more recent studies are cited, but not until the results section. I suggest updating the introduction to match the latest literature. Specifically, I would suggest briefly summarizing the collective findings of refs 14, 17, 33, and 34 in the introduction, and potentially adding PMID: 36510265.

2 Could the authors please elaborate on how the IBD analysis-derived germline mutation rates were validated based on sibling analysis?

3 “Single-nucleotide variants that interrupted repeat sequences greatly stabilized alleles”. Were the authors able to infer the introduction or loss of such interruptions? Were interruptions more likely to be observed in certain alleles? For example, longer vs shorter alleles or in certain loci?

4 Samples with heterozygous genotype and length differences of >3 units between the two alleles were used for studying germline instability. Please justify this threshold choice and state the number of loci before and after applying filter thresholds.

5 The first point in section 2.2 of the supplementary note needs further clarification. As I understand it the allele lengths are phased based on SNP haplotypes, and then the ancestral allele length is calculated from the top 5 IBD matching segments. Could you please clarify the description of this?

6 “As above, we estimated confidence intervals by rounding the total number of generations to the nearest integer and computing a binomial confidence interval, which was a reasonable approximation given that only a small fraction of individuals appeared in multiple pairs of closely related haplotypes.” Please add the fraction.

7 The authors mention that germline instability analyses are restricted to IBD pairs for which the ancestral allele was confidently determined. Please add summary statistics for the number of confidently predicted ancestral allele lengths.

8 The germline mutation rates for midsize alleles of the GLS repeats are calculated using an IBD pair from close relative (third degree and closer). Please include the number of samples from the cohort with mid-sized allele lengths and third-degree relationships.

9 Samples with heterozygous genotype and allele length difference of at least 5 units between the two alleles for investigating somatic mutations. Please include the number of samples for each locus with this difference between the alleles. Can you elaborate on why this threshold differs from the 3-unit threshold for germline instability, and if using different

thresholds could result in differences between these analyses?

(Remarks on code availability)

We were unable to fully assess the reusability and efficiency of the novel computational methods. Please provide a more detailed readme file for the software including usage information and considerations. Please provide sample input files for testing the submitted R-scripts.

Referee #2

(Remarks to the Author)

This study investigated the germline and somatic instability of 15 expansion-prone CAG repeat loci in >700,000 individuals from the UK Biobank and the All of Us consortium. This is, to my knowledge, the largest cohort used to study this question. The authors used sophisticated novel methods to estimate germline mutation rates from population data by inferring the genotypes of common ancestors based on shared IBD segments between two individuals. The estimated germline mutation rates for the 15 loci follow the known rules of mutation rate dependence on allele lengths and repeat interruptions. For refining the search for somatic mutations, the authors present a novel method for filtering out potential PCR error reads using base quality signatures. They profile somatic instability in 15 CAG loci, with four showing an association with the individual's age and allele lengths. They performed a GWAS identifying genetic modifiers of somatic instability in the highly unstable TCF4 locus. Variations in DNA repair-associated genes associated with the somatic instability at the HTT locus are also shown to be associated with somatic instability of the TCF4 locus. Interestingly, the genetic variants in MSH3 and FAN1 positively associated with somatic instability at the HTT locus in the brain are observed to be negatively associated with TCF4 instability in blood. Missense variation in the MSH2 locus strongly associated with somatic instability of the HTT locus in blood tissue is not associated with TCF4 somatic instability. These intriguing findings shed light on the potential tissue-specific and locus-specific roles of DNA repair mechanisms, particularly at STR loci. With available clinical data, the authors further probe the association of repeat instability in the GLS locus, a gene known to be associated with glutaminase deficiency and growth defects, to be associated with increased risk of kidney and liver diseases.

This study presented innovative methods and intriguing results, however, the study is limited to a small set of 15 CAG loci located in likely conserved genic regions. We wondered if the methods could be adapted for STR loci with different sequences and locus characteristics. Are the methods adequately reusable and scalable for a larger set of loci, more specifically for a genome-wide STR analysis with ~1 million loci? Also, we would have appreciated a more refined comparison of the intergenerational mutation rates against the conventional germline mutation rates based on family studies.

The study is a one-of-a-kind analysis using the largest collection of biobank datasets ever used to investigate STR mutation dynamics and associated genetic modifiers with the use of innovative methods. The authors also present interesting novel insights into the somatic mutation dynamics with tissue-specific and location-specific involvement of different DNA repair pathway genes. The limitation of the study was the selection of a small set of STR loci with limited sequence and location characteristics but this limitation does not undermine the novelty of the study.

Major comments

- 1 Given how specific the analysis is to only 15 CAG loci, the title is too broad. Please adjust it to reflect the scope.
- 2 The authors considered only 15 STRs with the motif CAG for the analysis, which represents Only a subset of the known CAG Mendelian disease loci. The manuscript does not sufficiently address the justification for this narrow selection criterion. We would recommend including other STR motifs, particularly those associated with known expansion disorders, which could provide insights into the polymorphisms of additional STR loci. We would additionally suggest including some loci located in non-transcribed regions to serve as controls since they would presumably not show tissue-specific patterns. While the inclusion of these regions would have been interesting, we acknowledge it would be a substantial computational undertaking and note that this limitation does not critically undermine the study.
- 3 "requiring mate sequences to map to loci that contain commonly-polymorphic CAG repeats" Can you elaborate on how this data from Ziaei Jam, H. et al. 2023 was used? Did it help to define the set of initial loci? Please clarify the total number of CAG loci in the human genome and the number of them considered polymorphic for this study. How many loci/bins contained IRRs that were discarded because they didn't meet this criteria?
- 4 Reads with >45 occurrences of the CAG motif are designated as IRRs and serve as a proxy for repeat expansions. The threshold length for considering an expansion reasonably depends on the reference allele length or the major allele length. For instance, HTT CAG repeat becomes pathogenic beyond the threshold length of 36 units. Have the authors explored lowering the threshold for defining repeat expansions?
- 5 Also, the authors identified IRRs based on only the reads that mapped to TCF4. Though it is shown that more than 95% of CAG IRRs align to this region, there are existing efficient tools that search for IRRs without making this assumption (e.g. ExpansionHunter Denovo, STRling, superSTR). Please justify the development of a new tool for this purpose and estimate the potential risks of missing these ~5% of reads.
- 6 Please justify the choice not to use an established STR genotyper such as HipSTR, GangSTR, ExpansionHunter etc. for genotyping repeat alleles <30 units with good spanning read coverage. In section 2.4 of the supplementary note (page 10 line 6) authors write "analyzing allele length discordances among close relatives in UKB based on DRAGEN genotyping". This would presumably mean ExpansionHunter was used for some analyses, yet there was a description of a novel genotyping method. Please clarify.
- 7 Please include benchmarking of the method for detecting repeat interruptions.
- 8 The authors present a novel and sophisticated method for estimating germline mutation rates of STR loci using population genomes. IBD segments from individual pairs are used to estimate the TMRCA and the ancestral alleles are predicted

based on an outgroup analysis. Could authors include statistics regarding the number of loci/samples with matching IBD segments within the selected criteria of $>5cM$ and $>0.5cM$ on either flank of the repeat? What is the average and range of generations represented at each of these thresholds? Do these vary by locus or even allele? The authors mentioned "if this consensus allele also matches one or both of the alleles in the IBD pair" in page 7 line 16 of the supplementary note indicating that the predicted allele of the ancestor should match one or both the alleles in the IBD pair. Please elaborate on the importance of this filter.

9 In the mutation rate calculation using TMRCA (formula mentioned in page 8 line 10), a single mutation is assumed to have occurred. Could you please discuss the possibility of multiple and back mutations and if this may lead to a conservative mutation rate estimate?

10 How did your mutation rates compare with prior studies that directly measured de novo mutations in trios/families? In general, and specifically triplet-repeat specific rates.

11 The HTT repeat locus was not observed to have somatic mutation rates associated with age. Can you please elaborate on why?

12 We were unable to fully assess the reusability and efficiency of the novel computational methods. Please provide a more detailed readme file for the software including usage information and considerations. Please provide sample input files for testing the submitted R-scripts.

13 Considering the potential broader applicability of the methods to similar studies, have the authors considered packaging the script into reusable tools?

Minor comments

1 The introduction uses an older citation for tandem repeat mutation rates (ref 13). Some of the more recent studies are cited, but not until the results section. I suggest updating the introduction to match the latest literature. Specifically, I would suggest briefly summarizing the collective findings of refs 14, 17, 33, and 34 in the introduction, and potentially adding PMID: 36510265.

2 Could the authors please elaborate on how the IBD analysis-derived germline mutation rates were validated based on sibling analysis?

3 "Single-nucleotide variants that interrupted repeat sequences greatly stabilized alleles". Were the authors able to infer the introduction or loss of such interruptions? Were interruptions more likely to be observed in certain alleles? For example, longer vs shorter alleles or in certain loci?

4 Samples with heterozygous genotype and length differences of >3 units between the two alleles were used for studying germline instability. Please justify this threshold choice and state the number of loci before and after applying filter thresholds.

5 The first point in section 2.2 of the supplementary note needs further clarification. As I understand it the allele lengths are phased based on SNP haplotypes, and then the ancestral allele length is calculated from the top 5 IBD matching segments. Could you please clarify the description of this?

6 "As above, we estimated confidence intervals by rounding the total number of generations to the nearest integer and computing a binomial confidence interval, which was a reasonable approximation given that only a small fraction of individuals appeared in multiple pairs of closely related haplotypes." Please add the fraction.

7 The authors mention that germline instability analyses are restricted to IBD pairs for which the ancestral allele was confidently determined. Please add summary statistics for the number of confidently predicted ancestral allele lengths.

8 The germline mutation rates for midsize alleles of the GLS repeats are calculated using an IBD pair from close relative (third degree and closer). Please include the number of samples from the cohort with mid-sized allele lengths and third-degree relationships.

9 Samples with heterozygous genotype and allele length difference of at least 5 units between the two alleles for investigating somatic mutations. Please include the number of samples for each locus with this difference between the alleles. Can you elaborate on why this threshold differs from the 3-unit threshold for germline instability, and if using different thresholds could result in differences between these analyses?

(Remarks on code availability)

We were unable to fully assess the reusability and efficiency of the novel computational methods. Please provide a more detailed readme file for the software including usage information and considerations. Please provide sample input files for testing the submitted R-scripts.

Referee #3

(Remarks to the Author)

In this manuscript, Hujuel et al. analyze germline and somatic mutations for 15 highly polymorphic CAG-repeat loci from 700,000 biobank participants. By using somatic expansion of TCF4 as a phenotype, the authors conduct a GWAS and identify seven loci where SNPs appear to modulate TCF4 repeat instability in blood cells. The authors also perform a TR PheWAS analysis on four CAG-repeat loci, reporting that a GLS TR germline expansion is associated with chronic kidney disease and liver diseases. Overall, although each part of the study offers some insights, none are especially novel or exciting, and the manuscript in its current form feels somewhat disconnected. Furthermore, the scope is very limited by analyzing only a small subset of potential TR loci. I have the following specific concerns regarding the manuscript:

Major Concerns

Scope vs. Title: The title, "Insights into the causes and consequences of DNA repeat expansions from 700,000 biobank participants," suggests a much broader scope than what is actually presented. In reality, only 15 CAG-repeat loci are analyzed; most tandem repeats (>3 million) are not addressed. And CAG repeats may not represent other motifs (e.g., GGC,

GCC, TA).

Disconnected Content: The manuscript seems to comprise three relatively disconnected parts:

Part 1 (Figure 1): Germline InstabilityThe authors estimate allele-specific intergenerational expansion and contraction rates for 15 CAG-repeat loci. Although potentially interesting, this portion is primarily descriptive.

Part 2 (Figures 2–4): Somatic InstabilityA GWAS on somatic expansion of TCF4 repeats identifies seven significant loci, most related to DNA damage pathways—an unsurprising result given the known association between DNA damage and repeat instability. Additionally, other recent research (e.g., Nature Genetics, PMID: 39843659) has explored repeat instability modifiers through in vivo CRISPR screening for HTT CAG repeats.

Part 3 (Figure 5): Germline TR-PheWASThe authors test only four TRs (GLS, HTT, TCF4, DMPK) in the UK Biobank and All of Us, uncovering a single novel association between GLS TR germline expansion and chronic kidney/liver diseases. This part is less interesting since other studies (e.g., Nature Communications, 2024, PMID: 39627187; Cell Genomics, 2023, PMID: 38116119) have identified hundreds of new TR-disease/trait associations in these biobanks.

It is challenging to replicate or verify the authors' workflow using the provided short, less-detailed scripts. A more comprehensive, step-by-step pipeline would be needed. This should include:

WGS data mapping

Extraction of WGS reads covering long CAG repeats

Quantification of somatic instability in short CAG-repeat alleles

PCR stutter filtering

GWAS on somatic expansion of TCF4 repeat alleles

Phenotypic associations of long CAG repeats

A practical example, such as using 1000 Genomes Project data, might help illustrate the approach.

Minor Concerns

The manuscript does not include a fine-mapping step to confirm whether the GLS expansion is causal rather than being in linkage disequilibrium (LD) with SNPs, indels, or structural variants.

In Figure 5D, the authors use “ ≥ 1 IRR pair” to categorize samples as “Highly-expanded.” It would be informative to examine higher cutoffs (e.g., ≥ 3 , ≥ 5 , ≥ 10 IRR pairs) to determine whether there is a dose-dependent effect on expansion.

Figure 5E appears to include only stage 5 CKD. It would be helpful to address stages 1–4 or clarify why the authors focused solely on stage 5.

(Remarks on code availability)

Please see Comments for Author

Referee #4

(Remarks to the Author)

This manuscript describes a study that characterizes somatic DNA repeat expansions in the blood, identifies germline genetic predictors of somatic expansions in TCF4, and links carrier status of somatic expansions to clinical phenotypes and traits. The paper is generally well written and clear. The methodology of measuring somatic repeat expansions is innovative and tested thoroughly through multiple different analyses. There are novel findings in the results, including genetic loci associated with somatic repeat expansion in TCF4 and links between somatic expansion in GLS and kidney and liver disease.

Introduction

The study focuses on somatic expansions and the introduction should clearly provide the setting for why the authors chose to address this specific topic.

The first paragraph is an extensive list of findings from the literature on short tandem repeats. I would suggest that the authors consider restructuring this into two paragraphs, the first devoted to STRs and germline changes to STRs and the second focused on somatic changes, which is the topic of the current manuscript. The section on somatic changes can be expanded a bit to set up the study at hand. The current second paragraph can then be made the third paragraph explaining how sequence data from large biobanks enable an investigation into somatic expansions that would not otherwise be possible on this scale.

Results

CAG trinucleotide repeat expansions in UK Biobank:

The text here is clear as to how the authors narrowed their focus to a specific set of CAG repeats in the UK Biobank

samples. Either here or in the introduction, the authors should clarify why they chose to focus on trinucleotide repeats, as opposed to other (or additional) repeat lengths. They should also clarify why they focused on CAG trinucleotide repeats versus other types of repeats (CTG repeats are also mentioned in the first paragraph).

Germline instability of common CAG-repeat alleles:

This is a very interesting section and well written on its own, but it needs a bit more of a transition from the first section. As it is, it reads almost like it was plucked from a different paper. One thing that might help here is to note that goal of characterizing changes in germline repeat length is to provide context for the core question of the paper, which is what leads to somatic expansion of CAG repeats in blood?

Somatic expansion of common CAG-repeat alleles in blood:

There is an important question about the somatic expansions identified in this study, and that is whether these are causal changes or a product of some blood specific process (clonal expansion). While it is not possible to compare these findings to other tissues (at this scale), it would be valuable to show that the somatic CAG expansions are not driven by other somatic alterations of the blood. The authors should test whether any of the somatic expansions are associated with telomere length, CHIP, mCA, mLOY, and mLOX. Prior studies have characterized these sequence derived phenotypes in the UKB and there are available GWAS summary statistics for them as well.

Inherited genetic modifiers of somatic TCF4 DNA-repeat expansion in blood:

The authors perform a GWAS of TCF4 somatic expansion, but not ExWAS. The data is available for ExWAS. The strength of the GWAS associations, in particularly MSH3, suggests that there may be sufficient power to conduct an ExWAS.

Repeat expansions in GLS associate with kidney and liver diseases:

Given the presentation of GLS somatic expansion associations with clinical outcomes, it is worth noting in the main text why the GWAS was restricted to TCF4.

Discussion

While the authors are right to emphasize the importance of large datasets and new computational approaches, the novel associations of both germline predictors of somatic expansions and the clinical consequences of those somatic expansions should be included in this first paragraph.

The use of the term “tissue-specific” in the second paragraph seems a bit presumptive given the phenotypes listed—two age of onset phenotypes and one in blood, which is the same tissue being studied here. If the comparison was to TCF4 somatic expansions in brain that would fit tissue specificity better. The issue here is the extent to which somatic changes observed in blood reflect similar effects in target tissues or whether they exert an influence indirectly via changes to the blood itself (immune system effects for example). This seems like an open question with some evidence from the current study for the same loci/genes but potentially different alleles being important in different phenotypes.

The discussion could use a final paragraph highlighting that there are novel associations between somatic expansions in the blood and clinical outcomes for multiple genes/loci. A second point for this final paragraph is the evidence identified within this study that disruption of expanded repeats appears to limit the impact of somatic expansions on disease outcomes. Both of these points are relevant to developing new treatments as the former identified potential novel causes of disease and the latter could point to an approach to treatment of somatic expansion associated disorders.

Supplement:

There are a number of analyses that would benefit from a finemapping and/or credible set analysis. If it is not possible to do this in the All of Us setting, doing so in UKB would greatly benefit the interpretation of the results.

Page 21. Mean PC value. For consideration—median might make more sense when the PC loadings are not normally distributed. (minor point)

Page 25 4.1 typo: resiudalized (minor point)

Supplementary Figure 9: I find this figure a bit hard to interpret. The patterns are sort of the same but there are some differences. It's not clear how much these patterns are driven by LD with lead SNPs. It would be helpful to finemap this locus and show where there is similarity and where there are differences.

Second point—I am not sure that I agree with the last two parenthetical statements. Germline allele length and somatic expansion are correlated phenotypes, as they authors show earlier in the paper. If they are certain that the germline and somatic effects they are reporting in this figure are independent, they should expand on that point to support those claims.

Supplementary Figure 10A: Same as for Figure S9, it would be easier to interpret with finemapping and plotting LD with lead signals. It's not clear how rare rarer variants are.

Supplementary Table 3 should also provide effect estimates and direction of effect for both UKB and All of Us.

Supplementary Tables 4 and 5: Provide more detail on the haplotype nomenclature in the Modifier columns.

Supplementary Tables 6-8: Fine to keep these with the highlighted findings, but the full set of tested phenotypes and their association results should be provided in a supplemental file so that readers can also see whether their phenotype was tested and not associated.

(Remarks on code availability)

"Code will be uploaded to GitHub upon publication (<https://github.com/mhujuel>)"

Version 1:

Reviewer comments:

Referee #1

(Remarks to the Author)

We would like to commend the authors for thoroughly answering and responding to all the initial review questions and comments.

In this revised version of the manuscript, the cohort size has increased substantially by ~200,000 more samples. Addressing the initial concern of limiting the study to only CAG motif repeat loci, the authors expanded the somatic instability analysis to include non-CAG repeat motifs with a total of ~350,000 short tandem repeats. Of these, they identified 154 STRs with common long alleles for further analysis and identified 17 of these to be somatically unstable. The results from the expanded analysis showed some interesting results with insights into the contributing factors of tandem repeat somatic instability.

We appreciate the authors' efforts in sharing their code with such thorough documentation. The inclusion of all post-processing steps along with example input files is especially helpful.

We have a few comments and questions remaining, mostly regarding the new methods implemented for the revision and the discrepancy of analysis between CAG and non-CAG loci.

Major:

The authors introduced a new, more extensive analysis of repeat instability in the revision. We have some concerns about the new approach to identifying unstable repeat loci. However, it is unclear if this is an issue of clarity or of the method itself. To summarise our understanding of the procedure used to identify somatically unstable non-CAG STR loci: the loci were classified as unstable if their instability was associated with variants in MSH2 and MSH3, which have been linked to instability at the TCF4 locus (by both previous studies and the current study). The authors then used these loci to perform a GWAS, which in turn revealed associations between repeat instability and variants in DNA repair genes, including MSH2 and MSH3, among others. This appears to create a circular logic, where the analysis is biased towards identifying the same GWAS loci that were used to select the STR in the first place. If this understanding of the method is incorrect, could the authors please edit the methods to improve clarity? If our understanding is correct, the authors should justify/defend the use of the method prominently in the manuscript. For example, by clearly stating this limitation in the relevant part of the results section, in addition to where it is currently described in the supplementary methods.

To our understanding, the instability measure calculated for CAG loci has not been applied to mid-length alleles of non-CAG loci. If this is the case, could the authors clarify why the same method was not used for non-CAG loci?

Specifically for the two AAAG loci, the authors generated a genetic liability for somatic expansion based on age, inherited allele length, and mean somatic expansion. Could the authors clarify why a similar analysis was not carried out for all 17 non-CAG unstable loci?

Minor:

Figure 3C. You may wish to confirm that this representation of sample-level data is sufficiently anonymized e.g. see <https://www.researchallofus.org/faq/data-user-code-of-conduct/>. If you haven't already, you can ask the Resource Access Board (RAB) by emailing aouresourceaccess@od.nih.gov.

Figure 5C. Could you please define these acronyms in the legend: TFC6, SDMT, TMS

(Remarks on code availability)

We appreciate the authors' efforts in sharing their code with such thorough documentation. The inclusion of all post-processing steps along with example input files is especially helpful.

Referee #2

(Remarks to the Author)

I co-reviewed this manuscript with one of the reviewers who provided the listed reports.

(Remarks on code availability)

We appreciate the authors' efforts in sharing their code with such thorough documentation. The inclusion of all post-processing steps along with example input files is especially helpful.

Referee #3

(Remarks to the Author)

The authors have addressed all my concerns with detailed response and changes to the manuscript

(Remarks on code availability)

NA

Referee #4

(Remarks to the Author)

The authors have provided a manuscript with greatly expanded scope and impact. Changes include an expansion of the number of repeats analyzed and additional association analyses. There remain a few additional issues to address.

1. "The vast majority of CAG repeat expansions in UKB occurred at only a few loci: 18 autosomal CAG repeat sequences in the human genome were expanded to ≥ 45 repeat units in at least five UKB participants (Fig. 1B and Supplementary Table 1)."

There is a brief reference to autosomal repeat sequences in the main text, as well as several notes about autosomal repeats in the supplement, but the rationale for focusing on only autosomal loci is not provided. Given that there is a well-studied, disease-associated GGC repeat in the first exon of the androgen receptor (see, for example, <https://academic.oup.com/molehr/article/9/7/375/1088087> and <https://www.sciencedirect.com/science/article/pii/S0002929707609100>), which is located on the X chromosome, this seems like a lost opportunity to include repeats with germline variation in length that are correlated with clinical phenotypes. It would be worth calling repeats on the X chromosome, sex-stratified, and including them in the PheWAS, also sex-stratified, if there are any that reach criteria for inclusion.

2. "Comparing genetic modifiers of somatic expansion of the TCF4 and HTT CAG repeats in blood revealed surprising heterogeneity (Fig. 4A,B). Common haplotypes at PMS2, FAN1, and ATAD5 associated with broadly concordant effects on TCF4 and HTT repeat expansion in blood, whereas at MSH3, common haplotypes that decreased expansion of the TCF4 repeat tended to increase expansion of the HTT repeat in blood (Fig. 4B). Moreover, the strongest modifier of HTT expansion in blood—a haplotype containing a missense variant in MSH2 also implicated in germline STR mutation14—appeared not to affect TCF4 expansion ($p=0.96$; Fig. 4A and Supplementary Table 5)."

It's unclear to me why the heterogeneity would be considered surprising—it also seems there is more consistency of effects than there is heterogeneity.

3. Figure 4

The X chromosome results are missing for the somatic expansion of HTT Manhattan plot.

4. Figure 5

There is a lot of information here and I think the presentation can be improved with some minor modifications.

Panel A: The middle columns present p-values and color code them according to significance. This presents the same (or similar) information twice. I think it would be more informative to present the effect sizes and directions and use the color coding to indicate significance level (nominal, genome-wide significance, or neither).

It's also unclear why certain loci do not have relative contribution plots. A note in the legend can address this.

Panel B: How were these four expansions selected for presentation when there were many more in the table in panel A? A note in the legend can address this.

Panel C: What do the signs (+/-) and scale of the effects mean here? Negative means shorter allele? Z score represents what measured units (a link to a supplemental file or methods description that gives these conversions or explains the scaling would be useful in the legend). What do the effect sizes shown mean for Huntington's disease (scale)? Why not make the colors consistent for positive and negative across the table. In general, the presentation of this data could be improved.

5. "Phenome-wide association analyses of 67,405 repeat loci expanded in at least five UKB participants identified seven repeat loci involved in 23 likely-causal associations with quantitative phenotypes and 46 associations with disease phenotypes (Supplementary Data 7 and 8)."

How many binary and quantitative phenotypes were tested?

(Remarks on code availability)

Detailed readme, code, and example output provided.

Version 2:

Reviewer comments:

Referee #1

(Remarks to the Author)

Thank you for addressing all our comments!

(Remarks on code availability)

Referee #2

(Remarks to the Author)

I co-reviewed this manuscript with one of the reviewers who provided the listed reports.

(Remarks on code availability)

Referee #4

(Remarks to the Author)

The authors have addressed the remaining comments.

(Remarks on code availability)

No additional review in this round.

Response to reviews of Nature 2024-12-26855 (Hujoel et al.)

We appreciate the reviewers' careful reading of our manuscript and thoughtful suggestions of ways to improve it. We have now completed a thorough revision that incorporates all of these suggestions. In particular:

- We have broadened the scope of the work to consider expansions of many thousands of STRs with 2-6bp repeat motifs of any type (rather than only CAG trinucleotide repeats). This was a great suggestion that turned up several exciting new results, including unstable AAAG repeats for which somatic expansion in blood cells associated with common inherited variants at 26 loci ($P = 5 \times 10^{-8}$ to 2.5×10^{-1438}). These genetic modifiers exhibited strong, first-order effects on repeat instability: somatic expansion rates varied by up to 4-fold between individuals with the highest and lowest 5% of polygenic scores.
- We have packaged the newly-developed analytical methods into more broadly usable software tools, including a tool for rapid extraction of sequencing reads containing expanded repeats and a tool that implements IBD-based estimation of germline STR mutation rates using haplotype data available on the UKB-RAP platform. We have also provided thorough documentation of our analytical workflow including code and example inputs and outputs.

Please see below for point-by-point responses to individual reviewer suggestions.

Referee #1/2:

This study investigated the germline and somatic instability of 15 expansion-prone CAG repeat loci in >700,000 individuals from the UK Biobank and the All of Us consortium. This is, to my knowledge, the largest cohort used to study this question. The authors used sophisticated novel methods to estimate germline mutation rates from population data by inferring the genotypes of common ancestors based on shared IBD segments between two individuals. The estimated germline mutation rates for the 15 loci follow the known rules of mutation rate dependence on allele lengths and repeat interruptions. For refining the search for somatic mutations, the authors present a novel method for filtering out potential PCR error reads using base quality signatures. They profile somatic instability in 15 CAG loci, with four showing an association with the individual's age and allele lengths. They performed a GWAS identifying genetic modifiers of somatic instability in the highly unstable TCF4 locus. Variations in DNA repair-associated genes associated with the somatic instability at the HTT locus are also shown to be associated with somatic instability of the TCF4 locus. Interestingly, the genetic variants in MSH3 and FAN1 positively associated with somatic instability at the HTT locus in the brain are observed to be negatively associated with TCF4 instability in blood. Missense variation in the MSH2 locus strongly associated with somatic instability of the HTT locus in blood tissue is not associated with TCF4 somatic instability. These intriguing findings shed light on the potential tissue-specific and locus-specific roles of DNA repair mechanisms, particularly at STR loci. With available clinical data, the authors further probe the association of repeat instability in the GLS locus, a gene known to be associated with glutaminase deficiency and growth defects, to be

associated with increased risk of kidney and liver diseases.

This study presented innovative methods and intriguing results, however, the study is limited to a small set of 15 CAG loci located in likely conserved genic regions. We wondered if the methods could be adapted for STR loci with different sequences and locus characteristics. Are the methods adequately reusable and scalable for a larger set of loci, more specifically for a genome-wide STR analysis with ~1 million loci? Also, we would have appreciated a more refined comparison of the intergenerational mutation rates against the conventional germline mutation rates based on family studies.

The study is a one-of-a-kind analysis using the largest collection of biobank datasets ever used to investigate STR mutation dynamics and associated genetic modifiers with the use of innovative methods. The authors also present interesting novel insights into the somatic mutation dynamics with tissue-specific and location-specific involvement of different DNA repair pathway genes. The limitation of the study was the selection of a small set of STR loci with limited sequence and location characteristics but this limitation does not undermine the novelty of the study.

We appreciate these kind comments about our work, and we have now implemented these helpful suggestions. Specifically, we have broadened the scope of the study to include other repeat motifs (see response to Comment 2 below), released reusable and scalable tools implementing the new methods (Comments 12 and 13), and compared our estimates of intergenerational mutation rates to previous literature (Comment 10).

Major comments

1 Given how specific the analysis is to only 15 CAG loci, the title is too broad. Please adjust it to reflect the scope.

We agree that the title was too broad for the scope of our initial submission. We have now extended the scope of the study to include all non-homopolymer STRs in the EnsembleTR catalog (Zaiei Jam et al. 2023) that were polymorphic in the 1000 Genomes EUR population, together with all disease-associated STRs in the STRipy catalog (Halman et al. 2022). Among 356,131 such repeat loci, we identified 67,405 STRs for which long alleles (~150bp or longer) were observed in at least five UKB participants, and we tested these repeats for association with 6,483 disease phenotypes. We further analyzed 154 of these STRs (those with 2,500 or more carriers of long alleles) for somatic instability, leading to several additional insights into genetic effects on instability of repeats with other motif types (Figures 5 and 6; Results, pp. 6-8). In light of the increased scope of the revised manuscript, we believe the title is now appropriate.

2 The authors considered only 15 STRs with the motif CAG for the analysis, which represents only a subset of the known CAG Mendelian disease loci. The manuscript does not sufficiently address the justification for this narrow selection criterion. We would recommend including other STR motifs, particularly those associated with known expansion disorders, which could provide insights into the polymorphisms of additional STR loci. We would additionally suggest including some loci located in non-transcribed regions to serve as controls since they would presumably not show tissue-specific patterns. While the inclusion of these regions would have

been interesting, we acknowledge it would be a substantial computational undertaking and note that this limitation does not critically undermine the study.

This was a great suggestion, and as explained above, we have now extended our key analyses to all 2-6bp repeat motifs and included all STRs associated with known repeat expansion disorders. These analyses did indeed provide additional insights into somatic expansion of STRs with different repeat motifs, showing that common genetic modifiers of repeat instability (e.g., at *MSH2*, *MSH3*, *PMS2*, and *MLH3*) contribute differently to expansion of repeats with different motifs (described in the new Fig. 5 and a new section of Results, pp. 6-7). Moreover, non-CAG repeats provided greatly increased power to detect genome-wide genetic influences on somatic repeat expansion in blood (26 loci; $P = 5 \times 10^{-8}$ to 2.5×10^{-1438}) and revealed striking examples of repeats at which somatic expansion rate is strongly shaped by genetics (up to 4-fold variation between individuals with the highest and lowest 5% of polygenic scores). These results are described in the new Fig. 6 and another new section of Results (pp. 7-8).

These new analyses were indeed a substantial computational undertaking, so to perform them efficiently on the UKB 500K WGS data set, we developed a new tool to rapidly extract in-repeat reads (of all motifs) from cram files. We have now released this software tool (see responses to Comments 5, 12, and 13 below).

3 “requiring mate sequences to map to loci that contain commonly-polymorphic CAG repeats” Can you elaborate on how this data from Ziaei Jam, H. et al. 2023 was used? Did it help to define the set of initial loci? Please clarify the total number of CAG loci in the human genome and the number of them considered polymorphic for this study. How many loci/bins contained IRRs that were discarded because they didn’t meet this criteria?

Yes, we used the Ziaei Jam et al. STR catalog to define the set of CAG loci considered. Specifically, from among 8,644 CAG repeat loci in the catalog, we identified 1,159 CAG loci that were polymorphic in 1000 Genomes EUR, and we used this list to filter the set of 100kb bins that contained an anchored IRR in at least five UKB participants. Among 25 such bins, 18 contained CAG loci on the list with coordinates that matched the coordinates of the anchored IRRs in the bin. We discarded the remaining seven bins. We have now clarified the description in Results (p. 3) and provided all of this information in Methods (section 1.2, p. 4-5). We have also added corresponding information for the new analyses in which we implemented a complementary pipeline (see response to Comment 5 below) to consider all non-homopolymer STRs (Methods, section 4.2.1; pp. 27-28).

4 Reads with >45 occurrences of the CAG motif are designated as IRRs and serve as a proxy for repeat expansions. The threshold length for considering an expansion reasonably depends on the reference allele length or the major allele length. For instance, HTT CAG repeat becomes pathogenic beyond the threshold length of 36 units. Have the authors explored lowering the threshold for defining repeat expansions?

This is a good point. We explored trying to lower the threshold for defining repeat expansions but found that focusing our analyses on IRRs was necessary to enable the computational optimizations that allowed us to efficiently extract reads derived from long repeats (at 356,131

STR loci in our new analyses) from ~500,000 WGS cram files. As such, we concluded that studying less-expanded alleles (which we agree is interesting) is better pursued by complementary efforts using other techniques; for example, Andrew Sharp's group recently performed PheWAS on 36,085 tandem repeats that they genotyped in the UKB 200K WGS data using ExpansionHunter (Manigbas et al. 2024, *Nat Commun*; ref. 10).

We did perform follow-up analyses of somatic instability of less-expanded alleles at repeat loci of particular interest based on our genome-wide scan for IRRs. Most notably, mid-length alleles of two AAAG repeats were sufficiently unstable to generate powerful GWAS phenotypes, providing new insights into genetic modifiers of repeat instability (Fig. 6 and Results, pp. 7-8). These analyses would be exciting to extend genome-wide in a future study (as doing so will require considerable additional method development).

5 Also, the authors identified IRRs based on only the reads that mapped to *TCF4*. Though it is shown that more than 95% of CAG IRRs align to this region, there are existing efficient tools that search for IRRs without making this assumption (e.g. ExpansionHunter Denovo, STRling, superSTR). Please justify the development of a new tool for this purpose and estimate the potential risks of missing these ~5% of reads.

We agree that the approach of ascertaining CAG IRRs from reads mapping to *TCF4* warranted evaluation given the potential applicability of existing tools. In our new analyses of STRs with all 2-6bp repeat motifs, we implemented a different approach that no longer relies on this assumption. We first tried ExpansionHunter Denovo (EHdn), STRling, and superSTR as suggested, but in pilot analyses of UKB WGS data, we found that EHdn and STRling each required ~40 mins to analyze a single WGS cram file (using both vCPUs of a UKB-RAP mem1_ssd1_v2_x2 compute instance), and superSTR required hours. Analyzing the full UKB 500K WGS data set would therefore cost ~\$10,000-\$30,000 and take weeks to months using EHdn or STRling (depending on load levels on the UKB-RAP platform).

To reduce the computational burden, we implemented a new IRR extraction tool similar to EHdn but ~5x faster (~8 minutes per WGS cram file). This lightweight utility uses the HTSLib API to rapidly stream a cram file from disk or from a remote URL, decoding only the minimal set of cram fields necessary to identify IRRs, and implementing an early-exit strategy to quickly filter non-IRR reads. We verified on a randomly selected set of 50 cram files that all IRRs reported by EHdn that satisfied our requirements for downstream analysis were also found by this new tool. The new tool is described in Methods (section 4.1, pp. 25-26), and we have now packaged and released it with documentation (see response to Comments 12 and 13).

For CAG-focused IRR analysis, the approach that we had used to extract IRRs from reads aligning to *TCF4* is still the most efficient, requiring only a few seconds of computation per file, so we retained this approach for the initial analyses of CAG repeats in our manuscript, reasoning that this complementary approach could still be valuable to others. We verified that PheWAS based on CAG IRRs detected from our genome-wide scan (without relying on reads aligning to *TCF4*) produced broadly similar results to the PheWAS we previously performed using CAG IRRs ascertained from *TCF4* alignments, suggesting a minimal impact of the small percentage of missed reads.

6 Please justify the choice not to use an established STR genotyper such as HipSTR, GangSTR, ExpansionHunter etc. for genotyping repeat alleles <30 units with good spanning read coverage. In section 2.4 of the supplementary note (page 10 line 6) authors write “analyzing allele length discordances among close relatives in UKB based on DRAGEN genotyping”. This would presumably mean ExpansionHunter was used for some analyses, yet there was a description of a novel genotyping method. Please clarify.

We agree that this choice also needed more explanation. The reason we did not use an established STR genotyper for genotyping short CAG repeat alleles was that for our analyses of somatic instability of these alleles, we needed to gather information about not only repeat length genotypes but also the sequence composition of repeat alleles (i.e., knowledge of any repeat interruptions) and their flanking sequences (which we used to evaluate reads for evidence of PCR stutter error). Additionally, we needed to record information about reads containing evidence of potential somatic expansion or contraction relative to the consensus alleles. The genotyping approach that we implemented was more conservative than established STR genotypers; for instance, it only generated genotypes for individuals that could confidently be determined to be heterozygous for two distinct short alleles that were each spanned by at least three reads. This choice was appropriate for most of our analyses, but to analyze germline mutability of mid-length *GLS* repeat alleles (25–40 repeat units), we needed genotypes from a more sensitive genotyper. The *GLS* repeat happened to be one of the 39 STRs genotyped in the DRAGEN STR call set released by UKB, so we used these genotypes for this specific purpose. We have added this information to Methods (now section 2.5, p. 11).

7 Please include benchmarking of the method for detecting repeat interruptions.

We appreciate this suggestion and have now included a benchmark validating the method for detecting repeat interruptions. In lieu of a direct benchmark against a gold-standard data set containing information about repeat interruptions for hundreds of publicly available WGS samples (which to our knowledge does not yet exist), we instead evaluated the robustness of our repeat interruption calls by examining the concordance of these calls among pairs of IBD2 siblings (who should have the same genotypes at a locus). We focused on the two common repeat interruptions in alleles of *TCF4* and *CA10* that offered the largest sample sizes for this comparison. Among the 922 IBD2 sibling pairs at *TCF4* for whom at least one of the siblings was genotyped as having an interruption within an 18-repeat allele of *TCF4*, both siblings were genotyped as having an interruption 100% of the time. Similarly, among the 336 IBD2 sibling pairs at *CA10* for whom at least one of the siblings was genotyped with an interruption in a *CA10* 17-repeat allele, both siblings were genotyped as having an interruption 100% of the time. We have now added this information to Methods (section 2.2, pp. 6-7).

8 The authors present a novel and sophisticated method for estimating germline mutation rates of STR loci using population genomes. IBD segments from individual pairs are used to estimate the TMRCA and the ancestral alleles are predicted based on an outgroup analysis. Could authors include statistics regarding the number of loci/samples with matching IBD segments within the selected criteria of >5cM and >0.5cM on either flank of the repeat?

We have now provided this information in the new Supplementary Table 3 (p. 57). The number of IBD pairs used in analysis was typically 100,000–200,000, varying mostly because of variation in the heterozygosity rates of different alleles, which determined the number of individuals considered in the analysis.

What is the average and range of generations represented at each of these thresholds? Do these vary by locus or even allele?

The range of generations represented among IBD segments >5cM in length that spanned >0.5cM on each flank of the repeat was typically 10–60 generations (i.e., TMRCA of 5–30 generations) and varied only slightly across loci (Supplementary Fig. 2A, p. 44). We have now further verified that IBD lengths did not exhibit any notable variation by repeat allele:

Average cM length by common repeat allele at TCF4 (across all haplotype pairs included in analysis)		
Repeat allele length	Average cM length	Standard Error of Mean
AGC12	16.677294	0.063507
AGC15	17.218123	0.086754
AGC16	16.614147	0.183060
AGC17	16.174131	0.286798
AGC18	17.577469	0.310757
AGC22	17.395673	0.378523
AGC23	16.858307	0.331274
AGC24	17.662201	0.247879
AGC25	16.688988	0.164563
AGC26	16.462437	0.213057
AGC27	16.417629	0.264880
AGC28	16.697784	0.357476
AGC8 ACC1 AGC9	16.866510	0.083871

The authors mentioned “if this consensus allele also matches one or both of the alleles in the IBD pair” in page 7 line 16 of the supplementary note indicating that the predicted allele of the ancestor should match one or both the alleles in the IBD pair. Please elaborate on the importance of this filter.

We have now clarified that the primary purpose of this filter is to eliminate IBD pairs in which the two alleles in the IBD pair match one another but they do not match the predicted allele of the ancestor (imputed from outgroup haplotypes). The most likely reason for this scenario is that a single mutation occurred on a branch of the tree between the ancestor and the outgroup haplotypes (rather than two mutations having occurred, one on each of the two branches connecting the ancestor to the IBD pair). Since we only wish to count mutations along the branches connecting the ancestor to the IBD pair, this scenario needs to be filtered. We have added this explanation to Methods (section 2.3, p. 8).

9 In the mutation rate calculation using TMRCA (formula mentioned in page 8 line 10), a single

mutation is assumed to have occurred. Could you please discuss the possibility of multiple and back mutations and if this may lead to a conservative mutation rate estimate?

Yes, a single mutation was assumed to have occurred, and this does in theory lead to mutation rate estimates that are slightly conservative because of the possibility of multiple mutations that would either be counted as a single mutation or filtered from analysis. However, the underestimation is expected to be slight given that the mutation rates that we estimated did not exceed 0.005 for most repeat loci and were all <0.01 (Fig. 1B), such that the likelihood of multiple mutations with the typical TMRCA of $\sim 10\text{--}30$ generations is low (Supplementary Fig. 2A). We have now explained this in Methods (section 2.3, p. 9).

10 How did your mutation rates compare with prior studies that directly measured de novo mutations in trios/families? In general, and specifically triplet-repeat specific rates.

The mutation rates that we estimated were broadly consistent with the range of mutation rates previously observed in studies of de novo mutations in families, falling in the more-mutable half of the spectrum of trinucleotide repeat rates (as expected given that we had ascertained CAG loci that reached lengths of 45+ CAG repeat units in UK Biobank). We have now provided estimates of the average mutation rate of each repeat locus (ranging from 8.2×10^{-5} to 9.5×10^{-4}) in the new Supplementary Table 3 (p. 57); these estimates fall within the higher end of the range previously estimated for trinucleotide repeats (3×10^{-6} to 9×10^{-4} in Extended Data Fig. 2b of Mitra et al. 2021, *Nature*) and exceed the genome-wide average of $\sim 5 \times 10^{-5}$ per haplotype per generation from previous de novo studies (Mitra et al. 2021, Kristmundsdottir et al. 2023, Porubsky et al. 2025). We have revised the main text to include these comparisons (Results, p. 3).

11 The HTT repeat locus was not observed to have somatic mutation rates associated with age. Can you please elaborate on why?

Our interpretation is that shorter *HTT* alleles (<30 repeats) expand in blood at rates below our power to detect an age effect in UKB. In order to be able to observe an association with age, we needed the rate of observing true somatic mutations to exceed the rate of residual PCR stutter artifacts (that were not caught by our filter). We were therefore only able to observe an age association for four particularly unstable repeats for which true somatic mutations of 30-CAG alleles were observed in $>0.5\%$ of blood cells (Supplementary Fig. 5). We have added this explanation to the legend of Supplementary Fig. 5.

12 We were unable to fully assess the reusability and efficiency of the novel computational methods. Please provide a more detailed readme file for the software including usage information and considerations. Please provide sample input files for testing the submitted R-scripts.

We apologize for the lack of documentation of the code that we previously submitted. As suggested, we have now provided a much more comprehensive set of scripts, readme files (with usage information and computational considerations), and example inputs and outputs (see response to next comment for further details).

13 Considering the potential broader applicability of the methods to similar studies, have the authors considered packaging the script into reusable tools?

This is a good point, and to support reusability of the methods that we developed, we have now created two GitHub repositories (<https://github.com/poruloh/extractLongSTRs> and <https://github.com/mhujoel/STRs>) containing the following components:

- 1) Standalone tool (extractLongSTRs) for efficiently extracting all reads with high repeat content (i.e., potential IRRs) from a cram file. The GitHub repository for this tool includes C++ code, precompiled binary executables, documentation, and an example.
- 2) Pipeline of bash and R scripts that post-processes extracted reads to identify IRRs that match known STR loci (from Zaiei Jam et al. 2023). We have included an example using publicly available 1000 Genomes Project data.
- 3) Pipeline for further processing IRR measurements into somatic expansion phenotypes and running GWAS on UKB-RAP. This pipeline includes code that imputes inherited allele lengths (which are used to generate somatic expansion phenotypes) using phased SNP-haplotypes available on UKB-RAP.
- 4) Pipeline for assessing somatic mutability of short alleles at a given STR locus (using base quality information to filter potential PCR stutter errors) including an example using 1000 Genomes Project data.
- 5) Pipeline for estimating germline mutation rates of alleles at a given STR locus (using analysis of IBD segments) using phased SNP-haplotypes available on UKB-RAP. We have provided an example using DRAGEN STR genotypes available on UKB-RAP.
- 6) Pipeline for quantifying mean somatic expansion of unstable mid-length alleles of AAAG repeats at chr2:232.4Mb and chr19:14.8Mb, including an example using 1000 Genomes Project data.

Regarding computational efficiency, all of these methods are scalable to hundreds of thousands of WGS files, and the IRR analysis pipeline (items 1-3) is scalable to genome-wide STR loci.

Minor comments

1 The introduction uses an older citation for tandem repeat mutation rates (ref 13). Some of the more recent studies are cited, but not until the results section. I suggest updating the introduction to match the latest literature. Specifically, I would suggest briefly summarizing the collective findings of refs 14, 17, 33, and 34 in the introduction, and potentially adding PMID: 36510265.

We appreciate the suggestion and have updated the introduction to summarize the findings of Mitra et al. 2021, Kristmundsdottir et al. 2023, Porubsky et al. 2025, Willems et al. 2016, and Steely et al. 2022 (PMID: 36510265, now cited as ref. 18).

2 Could the authors please elaborate on how the IBD analysis-derived germline mutation rates were validated based on sibling analysis?

We have now clarified in Methods (section 2.1, p. 6) that sibling pairs were used to validate germline mutation rates by comparing genotype discordance rates among IBD2 sibling pairs to the population-average mutation rate of each CAG repeat locus (i.e., the probability that a randomly-sampled allele mutates in one generation). Full details are provided in section 2.4 (pp. 10-11).

3 “Single-nucleotide variants that interrupted repeat sequences greatly stabilized alleles”. Were the authors able to infer the introduction or loss of such interruptions?

To investigate this question, we re-examined the IBD pairs and the ancestral alleles that we had imputed to estimate germline expansion and contraction rates of STRs. We found very limited evidence of introduction or loss of interruptions: across the 15 CAG loci, only an average of 3 IBD pairs per locus (out of ~100,000–200,000 IBD pairs) showed genotype discrepancies consistent with such a mutation. Given the difficulty of determining how many of these potential mutations represented real introductions or losses of interruptions (as opposed to rare error modes of genotyping, phasing, or IBD calling) and the limited sample size (too small to allow statistical analysis), we did not pursue this question further.

Were interruptions more likely to be observed in certain alleles? For example, longer vs shorter alleles or in certain loci?

The frequency distribution of interruptions seems to be random and most likely the result of genetic drift on rarely-occurring interruption mutations that has caused a few interrupted alleles to become common, as shown in Fig. 1B. A higher-resolution version of the allele frequency distributions shown in Fig. 1B is below; red indicates interrupted alleles, grey indicates non-interrupted alleles, and blue indicates alleles with alternate flank sequences (which could be considered to be interruptions or not depending on the definition that one chooses).

4 Samples with heterozygous genotype and length differences of >3 units between the two alleles were used for studying germline instability. Please justify this threshold choice and state the number of loci before and after applying filter thresholds.

We required an allele length difference of ≥ 3 repeat units in heterozygous individuals used for studying germline instability as this allowed us to confidently phase the two alleles (based on which allele had SNP-haplotype matches with longer repeats). We have provided the number of heterozygous samples for each locus before and after applying this filter in the new Supplementary Table 3 (p. 57) and shown that for most loci, the number of available heterozygous individuals did not greatly change whether the threshold was set at ≥ 1 unit (no filter), ≥ 3 units, or ≥ 5 units; we therefore chose ≥ 3 units to guard against phasing errors while retaining good statistical power. We have added this explanation to Methods (section 2.3, p. 7).

5 The first point in section 2.2 of the supplementary note needs further clarification. As I understand it the allele lengths are phased based on SNP haplotypes, and then the ancestral allele length is calculated from the top 5 IBD matching segments. Could you please clarify the description of this?

Yes, this understanding is correct and nicely summarizes the approach. We have clarified our description in what is now section 2.3 (p. 7) accordingly.

6 “As above, we estimated confidence intervals by rounding the total number of generations to the nearest integer and computing a binomial confidence interval, which was a reasonable approximation given that only a small fraction of individuals appeared in multiple pairs of closely related haplotypes.” Please add the fraction.

We have now included the number of unique and total individuals used in this analysis: the list of closely related haplotypes contained 428 individuals (from 214 haplotype pairs), representing 390 unique individuals (Methods, section 2.5; pp. 11-12). We have also provided the corresponding information for all analyses of germline mutation rates in Supplementary Table 3 (p. 57).

7 The authors mention that germline instability analyses are restricted to IBD pairs for which the ancestral allele was confidently determined. Please add summary statistics for the number of confidently predicted ancestral allele lengths.

We have also added this information in Supplementary Table 3 (specifically, the number of IBD pairs that satisfy our cM length requirements and the number of those IBD pairs with confidently predicted ancestral allele lengths).

8 The germline mutation rates for midsize alleles of the GLS repeats are calculated using an IBD pair from close relative (third degree and closer). Please include the number of samples from the cohort with mid-sized allele lengths and third-degree relationships.

This analysis used 390 unique individuals with mid-sized *GLS* alleles that were involved in 214 pairs of third-degree or closer relationships. We have included this information in Methods (section 2.5, pp. 11-12).

9 Samples with heterozygous genotype and allele length difference of at least 5 units between the two alleles for investigating somatic mutations. Please include the number of samples for each locus with this difference between the alleles. Can you elaborate on why this threshold differs from the 3-unit threshold for germline instability, and if using different thresholds could result in differences between these analyses?

This threshold differs from the ≥ 3 -unit threshold because we initially wished to evaluate somatic mutations of up to 2 repeat units in magnitude. We ultimately found that such 2-repeat-unit somatic expansions and contractions occurred too rarely to obtain useful estimates of their rates, but given that the 5-unit threshold did not greatly reduce the number of heterozygous individuals available for analysis for most loci (Supplementary Table 3, p. 57), we kept the 5-unit threshold to be cautious (as it ensured that an observed somatic mutation that differed from an inherited allele by 1 repeat unit was very likely to have arisen from that allele and not the other inherited allele).

Referee #1 (Remarks on code availability):

We were unable to fully assess the reusability and efficiency of the novel computational methods.

Please provide a more detailed readme file for the software including usage information and considerations. Please provide sample input files for testing the submitted R-scripts.

We have now included all of this information; please see responses to Major Comments 12 and 13 above for details.

Referee #3:

In this manuscript, Hujoel et al. analyze germline and somatic mutations for 15 highly polymorphic CAG-repeat loci from 700,000 biobank participants. By using somatic expansion of TCF4 as a phenotype, the authors conduct a GWAS and identify seven loci where SNPs appear to modulate TCF4 repeat instability in blood cells. The authors also perform a TR PheWAS analysis on four CAG-repeat loci, reporting that a GLS TR germline expansion is associated with chronic kidney disease and liver diseases. Overall, although each part of the study offers some insights, none are especially novel or exciting, and the manuscript in its current form feels somewhat disconnected. Furthermore, the scope is very limited by analyzing only a small subset of potential TR loci.

We appreciate this feedback and have greatly expanded the scope of the work to now consider all STR loci that can be robustly genotyped from short-read sequencing data (based on the EnsembleTR catalog; Zaiei Jam et al. 2023). These analyses have uncovered much richer insights into genetic effects on somatic repeat expansion that we have now reported in two new sections of Results and two new figures (Fig. 5 and Fig. 6). We have also revised the exposition to clarify the relationship between the components of the manuscript, the novelty of the results relative to previous work, and the relevance of the results to basic and translational research on repeat expansion disorders. These new results and revisions are detailed in the responses to specific points below.

I have the following specific concerns regarding the manuscript:

Major Concerns

Scope vs. Title: The title, “Insights into the causes and consequences of DNA repeat expansions from 700,000 biobank participants,” suggests a much broader scope than what is actually presented. In reality, only 15 CAG-repeat loci are analyzed; most tandem repeats (>3 million) are not addressed. And CAG repeats may not represent other motifs (e.g., GGC, GCC, TA).

We agree that the title was too broad for the scope of our initial submission. We have now extended the scope of the study to include all non-homopolymer STRs in the EnsembleTR catalog (containing 1.7 million TRs curated by Zaiei Jam et al. 2023) that were polymorphic in the 1000 Genomes EUR population, as well as all disease-associated STRs in the STRipy catalog (Halman et al. 2022). Among 356,131 such repeat loci, we identified 67,405 STRs for which long alleles (~150bp or longer) were observed in at least five UKB participants, and we tested these repeats for association with 6,483 disease phenotypes. We further analyzed 154 of these STRs (those with 2,500 or more carriers of long alleles) for somatic instability, leading to several additional insights into genetic effects on instability of repeats with other motif types (Figures 5 and 6; Results, pp. 6-8). In light of the increased scope of the revised manuscript, we believe the title is now appropriate, but we are open to modifying it if still desired.

Disconnected Content: The manuscript seems to comprise three relatively disconnected parts:

Part 1 (Figure 1): Germline Instability

The authors estimate allele-specific intergenerational expansion and contraction rates for 15 CAG-repeat loci. Although potentially interesting, this portion is primarily descriptive.

We appreciate the feedback and have now shortened this section to two paragraphs (Results, pp. 3-4). We have also clarified (on p. 3) that the estimates of germline mutation rates provide context for the subsequent analyses of somatic instability in blood (e.g., comparison of CAG repeat expansion rates in blood cells vs. intergenerationally; Fig. 2E). We retained the abbreviated content describing the key results of this section given that Referee #4 described it as a “very interesting section” and Referee #1/2 expressed enthusiasm for the “novel and sophisticated method” that enabled estimating allele-specific intergenerational mutation rates.

Part 2 (Figures 2–4): Somatic Instability

A GWAS on somatic expansion of TCF4 repeats identifies seven significant loci, most related to DNA damage pathways—an unsurprising result given the known association between DNA damage and repeat instability. Additionally, other recent research (e.g., Nature Genetics, PMID: 39843659) has explored repeat instability modifiers through in vivo CRISPR screening for HTT CAG repeats.

After taking the very helpful suggestion to expand the scope of these analyses to many more STRs (with other repeat motifs), we have obtained several new insights into how inherited genetic variants modulate somatic expansion of DNA repeats in blood. We have highlighted five new key results below:

(i) Strong, first-order effects of common inherited variants on somatic expansion of repeats

Non-CAG repeats provided greatly increased power to detect genetic influences on somatic repeat expansion in blood (26 loci; $P = 5 \times 10^{-8}$ to 2.5×10^{-1438}) and revealed striking examples of repeats at which somatic expansion rate is strongly shaped by genetics. These results are described in the new Fig. 6 and a new section of Results (pp. 7-8). At one AAAG repeat, we observed 4-fold variation between individuals with the highest and lowest 5% of polygenic scores for repeat expansion (Fig. 6D, copied below).

(ii) Strong effects of rare coding variants in genes not previously linked to repeat expansion

Burden tests identified strong AAAG repeat expansion-accelerating effects of pLoF variants in two genes (*XPC*, involved in nucleotide excision repair, and *NEIL2*, involved in base excision repair; Supplementary Table 7) that were not previously known to have a role in somatic repeat expansion and were not included in the recent complementary study of *Htt* expansion using in vivo CRISPR screening in an HD mouse model (Mouro Pinto et al. 2025, *Nat Genet*; now cited as ref. 65). Our genome-wide approach also identified common-variant associations at two other DNA damage response genes not previously linked to repeat expansion, *SMARCAD1* and *PARP1*. These new results are highlighted in Fig. 6B,C (copied below).

(iii) Subtler polygenic effects at loci not containing genes in DNA damage pathways

Beyond the strong effects of genetic variation in DNA repair genes, our highest-powered GWAS analyses also revealed subtler effects on somatic repeat instability at loci unrelated to DNA damage response. These loci included several transcription factors such as *RUNX1* and the *HOXA* and *HOXB* clusters, suggesting additional complexity in the way genetics shapes the propensity for repeats to expand (Fig. 6A, copied below).

(iv) Repeat locus-specificity of genetic modifier effects and effect directions

The relative contributions of DNA repair genes to repeat instability varied across repeats, with *MSH2* variation having a greater influence on dinucleotide repeats and *MSH3* variation having a

larger effect on STRs with longer motifs. Moreover, *MSH3* variation exhibited surprising heterogeneity in the direction of its effects on instability of different repeats: whereas *MSH3* alleles carrying expression-decreasing variants or damaging coding variants associated with reduced somatic instability of almost all repeats, the opposite was true for somatic instability of the *TCF4* repeat. Similarly, common modifier variants at *GADD45A*, *MSH6*, *XPC*, and *MLH3* associated with opposite-direction effects on *TCF4* instability compared to instability of other repeats, revealing surprising complexity in the way genetics shapes repeat instability. These results are presented in the new Fig. 5A and Fig. 5C (copied below) and a new section of Results (pp. 6-7).

Genetic modifier locus	Variant type	Variant	AF (%)	Cons. effect dir.	Somatic expansion of DNA repeat in blood										Hastening of Huntington's disease[26]					
					AC		AAG		AGC		AAAG		AAACC		AATAG		Onset	TFC6	SDMT	TMS
					chr2 37.0	chr15 74.9	chr9 36.5	HTT [26]	TCF4	chr11 123.5	chr19 14.8	chr1 34.8	chr1 54.5	chr17 48.9						
GADD45A	Common	1:67554471:T:C	97.6	-	-0.8	-3.4	0.2	0.8	5.5	-0.3	-2.6	1.1	0.8	-1.5	1.5	0.8	-0.2	0.9		
PARP1	Common	1:226349834:C:T	15.3	+	0.9	2.2	-0.2	4.8	-1.9	-1.0	4.3	-0.2	0.7	3.2	0.0	-1.1	-0.5	-1.0		
MSH2	Missense	2:47416318:G:A	1.6	-	-10.5	-7.3	-6.4	-13.4	0.0	-6.5	-29.6	-6.5	-7.9	-10.4	2.3	1.4	0.7	2.2		
MSH2	Common	2:47476151:G:A	13.4	+	3.4	-1.4	0.3	2.4	-0.6	0.6	3.8	0.6	-0.7	1.6	-1.8	-0.9	-0.1	-0.2		
MSH6	eQTL (+)	2:47784947:C:T	18.4	+	4.0	3.3	-1.0	5.8	-4.8	0.3	11.3	0.0	-0.5	1.3	-0.2	-3.0	-2.8	-0.7		
PMS1	Missense	2:189795860:C:T	0.2	-	-1.9	-3.4	-1.5	NA	-1.6	-2.4	-10.8	-2.1	-2.9	-2.9	-2.8	-2.3	-2.2	-1.6		
PMS1	Common	2:189809109:C:T	19.7	+	2.7	2.3	0.8	1.4	0.0	3.7	6.9	0.6	0.7	4.1	6.8	7.2	6.6	7.0		
XPC	Low frequency	3:14139206:C:T	2.0	+	1.7	-1.9	1.9	1.1	-3.5	4.4	8.5	1.5	2.5	-0.2	-0.7	1.8	1.3	1.5		
XPC	Common	3:14235797:A:G	22.1	-	-1.0	1.2	-1.0	-0.1	4.3	-0.6	-8.5	-1.1	-1.7	-2.9	-1.7	-0.8	0.2	-1.2		
SMARCAD1	Common	4:94174939:A:AAC	47.0	+	3.0	2.0	1.8	2.1	-1.3	-0.1	9.4	0.7	0.5	1.1	-0.7	0.1	0.0	0.7		
MSH3	eQTL (+)	5:80632699:T:C	24.5	+	6.4	5.4	1.4	8.9	-10.3	8.7	56.7	8.7	4.9	22.0	8.6	15.8	14.9	11.5		
MSH3	Rare	5:80790685:A:G	0.5	-	-3.6	-3.8	-6.6	-6.5	0.3	-10.0	-25.8	-8.2	-10.0	-11.8	-6.3	-8.6	-7.0	-8.3		
MSH3	Missense	5:80813660:T:G	0.4	-	-1.1	-1.2	-3.4	0.4	4.5	-3.8	-17.6	-5.4	-6.9	-9.4	2.6	0.0	-0.9	1.0		
PMS2	Common	7:6003481:C:T	63.1	-	-4.7	-7.0	-3.7	-7.7	-5.4	-4.5	-16.1	-5.2	-3.0	-7.6	-2.0	-4.3	-3.9	-3.8		
PMS2	Low frequency	7:6006003:T:C	1.2	+	2.8	3.4	2.2	4.8	3.1	3.4	10.2	2.2	3.9	2.2	-1.0	-0.8	-0.5	-0.2		
NEIL2	Common	8:11763260:T:C	32.9	+	-0.2	0.9	2.2	3.7	1.1	1.5	5.1	1.3	1.8	3.0	0.2	0.6	0.4	0.3		
MLH3	sQTL+eQTL	14:75027345:C:T	45.9	-	-4.9	-2.8	-1.4	-6.1	3.0	-4.8	-36.5	-4.4	-6.3	-7.0	4.0	4.2	3.2	4.0		
FAN1	Common	15:30873651:C:T	55.7	-	-2.6	-4.8	-1.0	-13.2	-10.0	1.2	-1.2	-1.7	-2.2	-0.1	6.4	7.3	5.6	7.7		
FAN1	Missense	15:30910758:G:A	1.1	+	2.9	1.6	-0.4	8.8	6.5	-2.3	-0.1	1.5	0.5	0.8	12.5	15.2	11.1	12.6		
ATAD5	Common	17:30887878:C:G	27.3	+	-0.1	0.0	-0.5	7.7	6.9	1.8	0.7	0.7	0.8	1.5	0.4	-0.3	-1.0	-0.7		

(v) Potential blood biomarkers for target engagement of expansion-slowing therapeutics

Long alleles of an AAAG tetranucleotide repeat in *ADGRE2* that were carried by 49% of European-ancestry UKB participants expanded at an average rate of 0.4 repeats per decade, demonstrating that human genomes commonly contain repeat elements that expand as we age. This repeat and other common unstable repeats (highlighted in the new Fig. 5B, copied below) could potentially serve as biomarkers of target engagement for expansion-slowing therapies for repeat expansion disorders.

Part 3 (Figure 5): Germline TR-PheWAS

The authors test only four TRs (GLS, HTT, TCF4, DMPK) in the UK Biobank and All of Us, uncovering a single novel association between GLS TR germline expansion and chronic kidney/liver diseases. This part is less interesting since other studies (e.g., Nature Communications, 2024, PMID: 39627187; Cell Genomics, 2023, PMID: 38116119) have identified hundreds of new TR-disease/trait associations in these biobanks.

We have now extended the PheWAS analysis to 67,405 STRs expanded to ~150bp in at least five UKB participants, identifying 46 associations with disease phenotypes and 23 associations with quantitative phenotypes that were not explained by LD with nearby SNPs (Supplementary Data 7 and 8). We chose to highlight the novel association between expanded *GLS* repeats and kidney and liver diseases because of its uniquely large effect size on disease phenotypes (e.g., OR=14.0 [5.7-34.3] for stage 5 CKD; Fig. 7E). For comparison, the disease association highlighted by Manigbas et al. 2024 (*Nat Commun*, PMID: 39627187) is an association with a 1.1-fold range of hypertension risk, and Margoliash et al. 2023 (*Cell Genom*, PMID: 38116119) did not analyze disease phenotypes. As such, while we agree that these previous studies have indeed identified similar numbers of TR-phenotype associations (101 high-confidence TR-trait associations reported by Manigbas et al.; 119 candidate causal STR-trait associations reported by Margoliash et al.) and represent important prior work (cited as refs. 9 and 10), the *GLS* association reported here represents a different-in-kind result.

It is challenging to replicate or verify the authors' workflow using the provided short, less-detailed scripts. A more comprehensive, step-by-step pipeline would be needed. This should include:

WGS data mapping

Extraction of WGS reads covering long CAG repeats

Quantification of somatic instability in short CAG-repeat alleles

PCR stutter filtering

GWAS on somatic expansion of TCF4 repeat alleles

Phenotypic associations of long CAG repeats

A practical example, such as using 1000 Genomes Project data, might help illustrate the approach.

We agree that more comprehensive code and documentation were needed to support replication and reuse of our methods, and we appreciate this list of items to include and the suggestion to include an example. We have now greatly expanded our code release to include the following components provided in two GitHub repositories (<https://github.com/poruloh/extractLongSTRs> and <https://github.com/mhujoel/STRs>):

- 1) Standalone tool (extractLongSTRs) for efficiently extracting all reads with high repeat content (i.e., potential IRRs) from a WGS cram file generated using any standard read-mapper such as bwa or DRAGEN. The GitHub repository for this tool includes C++ code, precompiled binary executables, documentation, and an example.

- 2) Pipeline of bash and R scripts that post-processes extracted reads to identify IRRs that match known STR loci (from Zaiei Jam et al. 2023). We have included an example using publicly available 1000 Genomes Project data.
- 3) Pipeline for further processing IRR measurements into somatic expansion phenotypes and running GWAS and PheWAS on the UKB-RAP platform. This pipeline includes code that imputes inherited allele lengths (which are used to generate somatic expansion phenotypes) using phased SNP-haplotypes available on UKB-RAP.
- 4) Pipeline for assessing somatic mutability of short alleles at a given STR locus (using base quality information to filter potential PCR stutter errors) including an example using 1000 Genomes Project data.
- 5) Pipeline for estimating germline mutation rates of alleles at a given STR locus (using analysis of IBD segments) using phased SNP-haplotypes available on UKB-RAP. We have provided an example using DRAGEN STR genotypes available on UKB-RAP.
- 6) Pipeline for quantifying mean somatic expansion of unstable mid-length alleles of AAAG repeats at chr2:232.4Mb and chr19:14.8Mb, including an example using 1000 Genomes Project data.

As suggested, we have provided example output for all of these steps, using publicly available 1000 Genomes Project whenever possible (so that scripts can be directly run by any user) and using data available to all researchers with access to UKB-RAP for components that require biobank data to demonstrate.

Minor Concerns

The manuscript does not include a fine-mapping step to confirm whether the *GLS* expansion is causal rather than being in linkage disequilibrium (LD) with SNPs, indels, or structural variants.

This point is well taken. We have now confirmed this by performing fine-mapping on the *GLS* associations as well as other associations of STR expansions with quantitative traits (described in Methods; section 5.1, p. 39); only associations not explained by LD with nearby SNPs or indels are reported in the new Supplementary Data 7.

In Figure 5D, the authors use “ ≥ 1 IRR pair” to categorize samples as “Highly-expanded.” It would be informative to examine higher cutoffs (e.g., ≥ 3 , ≥ 5 , ≥ 10 IRR pairs) to determine whether there is a dose-dependent effect on expansion.

We investigated this question and found that while our power was somewhat limited, the effect of *GLS* allele length on liver and kidney biomarkers appeared to be more consistent with a threshold effect arising around ~ 100 -200 CAG repeats (and stabilizing thereafter) than a dose-dependent. We have now noted this in Results (p. 8) and illustrated the allelic series in Fig. 7C.

Figure 5E appears to include only stage 5 CKD. It would be helpful to address stages 1–4 or clarify why the authors focused solely on stage 5.

We had actually also analyzed chronic kidney disease of all stages (not just stage 5) and provided the association data in Fig. 5E (now Fig. 7E), but we see now that this was unclear in the original manuscript text. We have clarified the text accordingly (Results, p. 9).

Referee #3 (Remarks on code availability):

Please see Comments for Author

Referee #4:

This manuscript describes a study that characterizes somatic DNA repeat expansions in the blood, identifies germline genetic predictors of somatic expansions in TCF4, and links carrier status of somatic expansions to clinical phenotypes and traits. The paper is generally well written and clear. The methodology of measuring somatic repeat expansions is innovative and tested thoroughly through multiple different analyses. There are novel findings in the results, including genetic loci associated with somatic repeat expansion in TCF4 and links between somatic expansion in GLS and kidney and liver disease.

We appreciate these positive comments about our work, and we have implemented the helpful suggestions below of ways to improve the manuscript.

Introduction

The study focuses on somatic expansions and the introduction should clearly provide the setting for why the authors chose to address this specific topic.

The first paragraph is an extensive list of findings from the literature on short tandem repeats. I would suggest that the authors consider restructuring this into two paragraphs, the first devoted to STRs and germline changes to STRs and the second focused on somatic changes, which is the topic of the current manuscript. The section on somatic changes can be expanded a bit to set up the study at hand. The current second paragraph can then be made the third paragraph explaining how sequence data from large biobanks enable an investigation into somatic expansions that would not otherwise be possible on this scale.

We have taken this suggestion and divided the first paragraph into two, focusing the first paragraph on STRs and germline mutations of STRs and the second on somatic mutability of STRs. We agree that this reorganization improved the clarity of the introduction (p. 2).

Results

CAG trinucleotide repeat expansions in UK Biobank:

The text here is clear as to how the authors narrowed their focus to a specific set of CAG repeats in the UK Biobank samples. Either here or in the introduction, the authors should clarify why they chose to focus on trinucleotide repeats, as opposed to other (or additional) repeat lengths. They should also clarify why they focused on CAG trinucleotide repeats versus other types of repeats (CTG repeats are also mentioned in the first paragraph).

We have now clarified at the beginning of Results (p. 3) that we began by analyzing CAG repeats because they are of particular interest for their role in progressive, neurodegenerative repeat expansion disorders and because they could be efficiently analyzed in biobank sequencing data. As such, we performed a deep analysis of CAG repeats, focusing in particular on the pathogenic *TCF4* repeat.

At the suggestion of the other reviewers, we have now broadened the scope of our study to extend our key analyses of somatic instability (and our STR-phenotype association analyses) to all 2-6bp repeat motifs. These analyses uncovered several exciting new results that we have now described in two new sections of Results (pp. 6-8) and two new figures (Fig. 5 and Fig. 6).

Regarding the mention of CTG repeats in the first paragraph, we had intended to simply indicate that our analysis included CAG repeats on either strand of the GRCh38 human reference genome, such that we considered both CAG and CTG repeat sequences in GRCh38. To avoid confusion, we have removed the mention of CTG in the main text and provided a proper explanation in Methods (section 1.1, p. 3).

Germline instability of common CAG-repeat alleles:

This is a very interesting section and well written on its own, but it needs a bit more of a transition from the first section. As it is, it reads almost like it was plucked from a different paper. One thing that might help here is to note that goal of characterizing changes in germline repeat length is to provide context for the core question of the paper, which is what leads to somatic expansion of CAG repeats in blood?

This is a good point, and as suggested, we have now changed the transition to this section (Results, p. 3) to explain that estimating germline mutation rates of repeats provides context for the subsequent analyses of somatic mutability.

Somatic expansion of common CAG-repeat alleles in blood:

There is an important question about the somatic expansions identified in this study, and that is whether these are causal changes or a product of some blood specific process (clonal expansion). While it is not possible to compare these findings to other tissues (at this scale), it would be valuable to show that the somatic CAG expansions are not driven by other somatic alterations of the blood. The authors should test whether any of the somatic expansions are associated with telomere length, CHIP, mCA, mLOY, and mLOX. Prior studies have characterized these sequence derived phenotypes in the UKB and there are available GWAS summary statistics for them as well.

We agree that it is important to verify that the somatic expansion phenotypes that we analyzed are not driven by other somatic alterations of the blood. We have now have tested all 21 such phenotypes (four for somatic expansion of short CAG repeat alleles in *GLS*, *ATNI*, *TCF4*, and *DMPK*; 17 from our new IRR-based analyses inclusive of non-CAG repeats) for association with telomere length, mCA, mLOY, and mLOX. (CHIP calls in UKB are not yet available; they are slated for the v20 data release in late 2025.) No association was Bonferroni-significant (minimum p-value 0.002 for $21 \times 4 = 84$ tests; Bonferroni-adjusted p-value of 0.17). We have noted this result in the legend of Supplementary Fig. 5 and described the analyses in Methods (section 4.3, p. 29).

Inherited genetic modifiers of somatic TCF4 DNA-repeat expansion in blood:

The authors perform a GWAS of TCF4 somatic expansion, but not ExWAS. The data is available for ExWAS. The strength of the GWAS associations, in particularly MSH3, suggests

that there may be sufficient power to conduct an ExWAS.

This was a great suggestion that we applied across all of our new somatic expansion phenotypes (including *TCF4* and 16 other non-CAG repeats; Fig. 5A), revealing strong effects of rare coding variants in genes not previously linked to repeat expansion. Specifically, burden tests identified strong AAAG repeat expansion-accelerating effects of pLoF variants in two genes (*XPC*, involved in nucleotide excision repair, and *NEIL2*, involved in base excision repair; Supplementary Table 7), suggesting that deficiencies in nucleotide excision repair and base excision repair increase instability of repeated DNA (Fig. 6C, copied below on the right).

Among single-variant associations with somatic expansion phenotypes, we also observed two associations involving low-frequency missense variants (MAF 0.1%–1%). One was the *MSH3* L911W variant (chr5:80813660:T>G) that we had observed to be associated with increased *TCF4* repeat expansion; this variant now also associated with decreased expansion of AAAG repeats in our new analyses (Supplementary Data 6), consistent with the opposing effects of *MSH3* variation on *TCF4* repeat expansion compared to other repeats (Fig. 5C). The other was a missense variant in *PMS1* (chr2:189795860:C>T; *PMS1* T75I) that associated with decreased expansion of AAAG repeats (Supplementary Data 5). This variant appears to strongly delay onset of Huntington’s disease (by 5.2 years (s.e. 1.9 years); $p=0.005$ in summary statistics from GeM-HD). We have now noted this new result in the main text (Results, p. 8).

Repeat expansions in GLS associate with kidney and liver diseases:

Given the presentation of GLS somatic expansion associations with clinical outcomes, it is worth noting in the main text why the GWAS was restricted to *TCF4*.

We have now extended the GWAS analyses to a total of 17 repeats at which we observed evidence of somatic instability based on associations of repeat length with age or with haplotypes of mismatch repair genes (*MSH2* and *MSH3*; Fig. 5A) and described these analyses in Results (pp. 6-7) and Supplementary Fig. 10). The list of repeats included in GWAS did not contain *GLS* because the highly expanded *GLS* alleles that associated with clinical outcomes were too rare to create a somatic expansion phenotype useful for GWAS ($n=98$ carriers in UKB).

Discussion

While the authors are right to emphasize the importance of large datasets and new computational approaches, the novel associations of both germline predictors of somatic expansions and the clinical consequences of those somatic expansions should be included in this first paragraph.

We agree and have modified this paragraph to highlight the novel findings about genetic modifiers and clinical consequences of repeat instability (Discussion, p. 10).

The use of the term “tissue-specific” in the second paragraph seems a bit presumptive given the phenotypes listed—two age of onset phenotypes and one in blood, which is the same tissue being studied here. If the comparison was to *TCF4* somatic expansions in brain that would fit tissue specificity better. The issue here is the extent to which somatic changes observed in blood reflect similar effects in target tissues or whether they exert an influence indirectly via changes to the blood itself (immune system effects for example). This seems like an open question with some evidence from the current study for the same loci/genes but potentially different alleles being important in different phenotypes.

This point is well taken, and we have modified this paragraph to focus on the repeat-specific differences we observed in the effect of genetic modifiers on expansion of the *TCF4* repeat versus other repeats (Discussion, p. 10).

The discussion could use a final paragraph highlighting that there are novel associations between somatic expansions in the blood and clinical outcomes for multiple genes/loci. A second point for this final paragraph is the evidence identified within this study that disruption of expanded repeats appears to limit the impact of somatic expansions on disease outcomes. Both of these points are relevant to developing new treatments as the former identified potential novel causes of disease and the latter could point to an approach to treatment of somatic expansion associated disorders.

We appreciate this suggestion to discuss the potential implications of these results for development of new treatments. We decided to focus this additional paragraph (p. 10) on the potential for some of the unstable repeats that we have identified to serve as biomarkers of target engagement for expansion-slowing therapies for repeat expansion disorders. Regarding the potential for therapeutic introduction of repeat interruptions as an approach to treating somatic repeat expansion disorders, a new study has just been published using base editing to demonstrate this approach in a mouse model of Huntington’s disease (Matuszek et al. 2025, *Nat Genet*; PMID: 40419681), so we have added a citation to this study (in Results, p. 4; ref. 44).

Supplement:

There are a number of analyses that would benefit from a finemapping and/or credible set analysis. If it is not possible to do this in the All of Us setting, doing so in UKB would greatly benefit the interpretation of the results.

We agree and have now performed fine-mapping and credible set analyses (in UKB) as suggested. Specifically, we implemented a fine-mapping filter to eliminate associations between long STR alleles and quantitative traits that were explainable by LD with nearby SNPs or indels (Methods, section 5.1, p. 39; Supplementary Data 7), and we fine-mapped GWAS loci for somatic expansion phenotypes (Methods, section 4.8, pp. 35-36) and reported the credible sets in Supplementary Data 6.

Page 21. Mean PC value. For consideration—median might make more sense when the PC loadings are not normally distributed. (minor point)

Yes, using median PC values rather than mean PC values to define ancestry cluster centers would be a more robust approach in general. We would do this if we were starting the study now, but given that our existing genetic ancestry assignments showed good concordance with self-reported ethnicity, we opted not to make this change as it would require rerunning nearly every analysis that we performed.

Page 25 4.1 typo: resiudalized (minor point)

Thanks; we have corrected this typo.

Supplementary Figure 9: I find this figure a bit hard to interpret. The patterns are sort of the same but there are some differences. It's not clear how much these patterns are driven by LD with lead SNPs. It would be helpful to finemap this locus and show where there is similarity and where there are differences.

Second point—I am not sure that I agree with the last two parenthetical statements. Germline allele length and somatic expansion are correlated phenotypes, as they authors show earlier in the paper. If they are certain that the germline and somatic effects they are reporting in this figure are independent, they should expand on that point to support those claims.

These points are well taken, and we have therefore removed this supplementary figure given its limited interpretability. We agree that it is unclear whether associations of SNPs at the *TCF4* locus with *TCF4* somatic repeat expansion reflect genuine *cis*-effects or imperfect control for inherited allele length, and we have now updated the legend of Fig. 4A to note this.

Supplementary Figure 10A: Same as for Figure S9, it would be easier to interpret with finemapping and plotting LD with lead signals. It's not clear how rare rarer variants are.

We agree and have also removed this supplementary figure, as our new GWAS analyses of somatic AAAG repeat expansions were much better-powered, and we have now fine-mapped the much stronger associations at these loci and provided fine-mapped credible sets (Supplementary Data 6).

Supplementary Table 3 should also provide effect estimates and direction of effect for both UKB and All of Us.

We have added the effect directions for UKB and *All of Us* to this table (now renumbered to Supplementary Table 4; p. 58).

Supplementary Tables 4 and 5: Provide more detail on the haplotype nomenclature in the Modifier columns.

We have now explained the haplotype nomenclature (defined by the GeM-HD Consortium) in the legends of these supplementary tables (now Supplementary Tables 5 and 6; pp. 59-60). Briefly, haplotypes are named according to chromosome number, then order of locus discovery on a given chromosome (i.e., A, B, ...), and then a sequential number for each modifier haplotype at a given locus (i.e., M1, M2, M3, ...).

Supplementary Tables 6-8: Fine to keep these with the highlighted findings, but the full set of tested phenotypes and their association results should be provided in a supplemental file so that readers can also see whether their phenotype was tested and not associated.

As these files are bulky (~4GB total), we have uploaded the full set of PheWAS association data for quantitative and disease phenotypes to https://data.broadinstitute.org/lohlab/UKB_STR_expansion_sumstats/ and will deposit them on Zenodo upon completion of the review process. We have updated the Data Availability section accordingly (p. 12).

Referee #4 (Remarks on code availability):

"Code will be uploaded to GitHub upon publication (<https://github.com/mhujoel>)"

At the suggestion of the other reviewers, we have greatly expanded the code release to include a GitHub repository providing a standalone tool for efficient extraction of IRRs from WGS cram files (<https://github.com/poruloh/extractLongSTRs>) as well as a GitHub repository describing all other computational pipelines used for analysis (<https://github.com/mhujoel/STRs>).

Referee #1/2:

We would like to commend the authors for thoroughly answering and responding to all the initial review questions and comments.

In this revised version of the manuscript, the cohort size has increased substantially by ~200,000 more samples. Addressing the initial concern of limiting the study to only CAG motif repeat loci, the authors expanded the somatic instability analysis to include non-CAG repeat motifs with a total of ~350,000 short tandem repeats. Of these, they identified 154 STRs with common long alleles for further analysis and identified 17 of these to be somatically unstable. The results from the expanded analysis showed some interesting results with insights into the contributing factors of tandem repeat somatic instability.

We appreciate the authors' efforts in sharing their code with such thorough documentation. The inclusion of all post-processing steps along with example input files is especially helpful.

We have a few comments and questions remaining, mostly regarding the new methods implemented for the revision and the discrepancy of analysis between CAG and non-CAG loci.

We appreciate the careful review and the helpful feedback below, which we have incorporated into the new revision as detailed below.

Major:

The authors introduced a new, more extensive analysis of repeat instability in the revision. We have some concerns about the new approach to identifying unstable repeat loci. However, it is unclear if this is an issue of clarity or of the method itself. To summarise our understanding of the procedure used to identify somatically unstable non-CAG STR loci: the loci were classified as unstable if their instability was associated with variants in *MSH2* and *MSH3*, which have been linked to instability at the *TCF4* locus (by both previous studies and the current study). The authors then used these loci to perform a GWAS, which in turn revealed associations between repeat instability and variants in DNA repair genes, including *MSH2* and *MSH3*, among others. This appears to create a circular logic, where the analysis is biased towards identifying the same GWAS loci that were used to select the STR in the first place. If this understanding of the method is incorrect, could the authors please edit the methods to improve clarity? If our understanding is correct, the authors should justify/defend the use of the method prominently in the manuscript. For example, by clearly stating this limitation in the relevant part of the results section, in addition to where it is currently described in the supplementary methods.

This is a good question that we realize was unclear because of the abbreviated description of the approach (and the analyses that led to it) in the main text.

The logic behind the approach was that we initially identified somatically unstable STR loci as those for which repeat length (measured by IRR count) associated significantly with age. This analysis (which looked only at age, not at association with *MSH2* or *MSH3* variants) identified 10 somatically unstable STRs (p-value for age < 0.00016; Fig. 5A). When we ran GWAS on

somatic expansion of these 10 STR loci, we observed that eight of the 10 STRs had at least one GWAS hit, and for all eight STRs, the lead variant in the GWAS was either at *MSH2* or *MSH3* (Supplementary Fig. 10 and Supplementary Data 3). Moreover, for half of the STRs, the somatic expansion phenotype associated more strongly with the lead GWAS variant than it did with age. This suggested the possibility of identifying a few additional somatically unstable STRs by also ascertaining unstable repeats based on their association with genetic variants at mismatch repair genes.

To minimize hypothesis testing burden and computational cost, we focused this additional identification strategy on *MSH2* and *MSH3* haplotypes (based on the above observation that for all of the somatically unstable STRs identified from the age association analysis, the lead GWAS locus was either *MSH2* or *MSH3*). We further observed that at *MSH2*, the G322D missense variant chr2:47416318:G>A always had the strongest or near-strongest signal (>90% the strength of the lead variant), and at *MSH3*, the common haplotype chr5:80638411:G>T was always a good representative of the locus (lead variant for three STRs; always $\geq 50\%$ the strength of the lead variant). We therefore tested all of the 154 STRs for association with these two variants, identifying an additional seven STRs with evidence of somatic instability (for a total of 17 somatically unstable STRs, including the 10 initially identified based on association with age).

While we agree that this approach could in theory bias our ascertainment of unstable STRs to those affected by *MSH2* and *MSH3* variants, this bias is at most modest (given that 10 of the 17 unstable STRs were identified based on association with age alone) and probably minimal (given that *MSH2* or *MSH3* was the lead GWAS locus for all of the age-identified unstable STRs).

We have revised the exposition in the main manuscript (p. 6) to clarify the approach and its rationale, with a reference to Methods, where we have provided a full discussion (section 4.3.1: “Pilot analyses motivating using association with *MSH2* and *MSH3* haplotypes to find unstable STRs”).

To our understanding, the instability measure calculated for CAG loci has not been applied to mid-length alleles of non-CAG loci. If this is the case, could the authors clarify why the same method was not used for non-CAG loci?

Correct; we did not end up applying the approach that we had developed to estimate somatic instability of CAG loci (Fig. 2D and Supplementary Figs. 5 and 6) to analyze mid-length alleles of non-CAG loci (specifically, AAAG repeats) in our revised manuscript. The reason was that the CAG and AAAG repeats turned out to have very different relative signal-to-noise ratios of somatic expansion vs. PCR stutter:

- For CAG repeats, we had observed that PCR stutter errors comprised a considerable fraction of apparent repeat expansions, which were infrequently observed (Supplementary Fig. 4B). As such, to accurately quantify instability of CAG repeats, we needed to stringently filter for PCR stutter error (by analyzing base qualities within reads), considerably reducing the pool of reads available for analysis in order to minimize bias in our estimates of instability.

- In contrast, the mid-length AAAG alleles that were a focus of our revision were highly mutable, such that reads containing somatic expansions were usually observable in carriers of alleles with 22+ repeats, and few of these reads could have originated from PCR stutter based on the observed relationship with age (Fig. 6D and Supplementary Fig. 12A,B). Additionally, a main focus of our analyses of mid-length AAAG repeat instability was to identify germline drivers (by generating a powerful GWAS phenotype) rather than to compute an unbiased estimate of mutation rate. This motivated retaining as many reads as possible for analysis rather than attempting to filter for PCR stutter.

We have clarified this reasoning in Methods (section 4.6.3: “Differences in approach compared to spanning-read analyses of CAG repeats”).

Specifically for the two AAAG loci, the authors generated a genetic liability for somatic expansion based on age, inherited allele length, and mean somatic expansion. Could the authors clarify why a similar analysis was not carried out for all 17 non-CAG unstable loci?

We only performed this analysis on the two highly unstable AAAG repeats because we only generated the mean somatic expansion phenotype for these two STRs of highest interest (as their mid-length alleles were both common and highly unstable). We did not extend this analysis to the other STRs because estimating mean somatic expansion of mid-length alleles was a complex task—requiring locus-specific optimizations—for four main reasons:

1. The most informative mid-length alleles of an STR are the longest, most mutable alleles, but these alleles also have the fewest spanning reads (due to their length). As such, the precise criteria used to identify spanning reads (e.g., thresholds on numbers of flanking bases and mismatches allowed) can substantially impact performance, and the parameter values that maximize detection sensitivity while controlling false positives are locus-specific.
2. The germline allele from which a putatively-expanded allele originates can be difficult to determine. In the simplest scenario, an individual is heterozygous for two distinct germline alleles, each supported by several spanning reads. However, in instances in which only one germline allele has support from multiple spanning reads, the individual is likely to either be homozygous for a single germline allele, heterozygous for a mid-length allele that has expanded to many different lengths, or heterozygous for a long allele. These possibilities can usually be distinguished based on counts of spanning reads, flanking reads, and IRRs, but the optimal way to utilize this information depends on the allele frequency distribution, level of instability, and error profile of a specific locus.
3. The sequence immediately flanking a repeat can harbor common variants that need to be taken into account when identifying spanning reads. This did not affect the two highly unstable AAAG repeats that we analyzed, but we noticed common variation in the flanks of a few of the other 15 unstable STRs that would need to be modeled.
4. The extent to which PCR stutter error contributes to observed spanning reads supporting putative expansions is locus-specific, such that filtering may or may not be required.

We have added this explanation to Methods (section 4.6.4: “Complexities of assessing instability of mid-length alleles of other STRs”).

Minor:

Figure 3C. You may wish to confirm that this representation of sample-level data is sufficiently anonymized e.g. see <https://www.researchallofus.org/faq/data-user-code-of-conduct/>. If you haven't already, you can ask the Resource Access Board (RAB) by emailing aouresourceaccess@od.nih.gov.

Yes, this representation of individual-level data required an exception from the AoU RAB. We had already obtained this exception (as indicated in the figure legend), but we appreciate the pointer.

Figure 5C. Could you please define these acronyms in the legend: TFC6, SDMT, TMS

We have now defined the acronyms in the legend.

Referee #1/2 (Remarks on code availability):

We appreciate the authors' efforts in sharing their code with such thorough documentation. The inclusion of all post-processing steps along with example input files is especially helpful.

We are grateful for the suggestion to improve the reusability of the code and are glad to hear that the additional documentation and examples are helpful.

Referee #3:

The authors have addressed all my concerns with detailed response and changes to the manuscript

Referee #3 (Remarks on code availability):

NA

We appreciate the reviewer's useful feedback that helped make the revised manuscript much stronger.

Referee #4:

The authors have provided a manuscript with greatly expanded scope and impact. Changes include an expansion of the number of repeats analyzed and additional association analyses. There remain a few additional issues to address.

1. “The vast majority of CAG repeat expansions in UKB occurred at only a few loci: 18 autosomal CAG repeat sequences in the human genome were expanded to ≥ 45 repeat units in at least five UKB participants (Fig. 1B and Supplementary Table 1).”

There is a brief reference to autosomal repeat sequences in the main text, as well as several notes about autosomal repeats in the supplement, but the rationale for focusing on only autosomal loci is not provided. Given that there is a well-studied, disease-associated GGC repeat in the first exon of the androgen receptor (see, for example, <https://academic.oup.com/molehr/article/9/7/375/1088087> and <https://www.sciencedirect.com/science/article/pii/S0002929707609100>), which is located on the X chromosome, this seems like a lost opportunity to include repeats with germline variation in length that are correlated with clinical phenotypes.

It would be worth calling repeats on the X chromosome, sex-stratified, and including them in the PheWAS, also sex-stratified, if there are any that reach criteria for inclusion.

We agree that repeats on the X chromosome are also of interest, and the existence of several known X-linked repeat expansion disorders motivates extending the PheWAS to include X chromosome repeat expansions. We had previously excluded X chromosome repeats because the database of STRs that we used to ascertain repeat expansions did not include STRs on the X chromosome. However, we have now circumvented this difficulty by implementing a reference-free approach to test X chromosome repeat expansions for associations with phenotypes.

In more detail, we generated bin-level genotypes indicating, for each 1kb bin on chromosome X and for each possible repeat unit size (2bp, 3bp, 4bp, 5bp, 6bp), whether or not an individual had a highly repetitive read with the corresponding motif size whose mate aligned to the 1kb bin or one of its two flanking 1kb bins (building off the reference-free approach described in Methods, section 4.2.1). We identified 108,831 such bin+motif-size pairs on chromosome X for which at least 10 UKB participants were carriers of potential repeat expansions (i.e., had at least one highly repetitive read whose mate aligned within the 3kb window corresponding to the bin). We then conducted sex-stratified PheWAS, testing whether expansion status putatively encoded by any of these 108,831 measurements associated with any quantitative or disease traits.

These PheWAS analyses only identified disease associations involving two clusters of bin measurements (at chrX:154.341–154.343Mb and chrX:154.389–154.392Mb) that turned out to both reflect mis-mapping of reads derived from repeat expansions of the *C9orf72* CCCC GG repeat on chromosome 9. Most individuals genotyped to be “carriers” of these bin-level measurements had expanded *C9orf72* CCCC GG repeats (>80%), and the associated diseases were the same neurodegenerative and dementia-related diseases that we had observed to associate with *C9orf72* expansions (Supplementary Data 8).

Similarly, the PheWAS of quantitative traits also identified only associations that appeared to reflect mis-mapping of reads derived from expanded autosomal STRs to loci on the X chromosome:

- associations of bins capturing mis-mapped *C9orf72* CCCC GG repeat expansions with reduced BMI and monocyte counts
- associations of bins capturing mis-mapped *CNBP* TCTG repeat expansions (actually located on chromosome 3, and pathogenic for myotonic dystrophy 2) with reduced creatinine
- associations of bins in the Xq telomere (likely capturing reads derived from telomere repeats on many chromosomes) with increased telomere length
- associations of bins in the Xp subtelomeric region (containing only a 26bp VNTR but not STR, and thus likely reflecting mis-mapping) with decreased neutrophil and white blood cell counts.

While these results primarily demonstrate the challenges of short-read-based analyses of STRs and the importance of using a reference panel for such analyses (and did not identify any real associations of expanded X chromosome repeats with diseases or quantitative traits), we have included them in the supplement (section 5.5).

2. “Comparing genetic modifiers of somatic expansion of the TCF4 and HTT CAG repeats in blood revealed surprising heterogeneity (Fig. 4A,B). Common haplotypes at PMS2, FAN1, and ATAD5 associated with broadly concordant effects on TCF4 and HTT repeat expansion in blood, whereas at MSH3, common haplotypes that decreased expansion of the TCF4 repeat tended to increase expansion of the HTT repeat in blood (Fig. 4B). Moreover, the strongest modifier of HTT expansion in blood—a haplotype containing a missense variant in MSH2 also implicated in germline STR mutation¹⁴—appeared not to affect TCF4 expansion ($p=0.96$; Fig. 4A and Supplementary Table 5).”

It’s unclear to me why the heterogeneity would be considered surprising—it also seems there is more consistency of effects than there is heterogeneity.

These are good points. We have reworded this text to note both that we observed both consistency and heterogeneity of effects, and we have removed the word “surprising”.

3. Figure 4

The X chromosome results are missing for the somatic expansion of HTT Manhattan plot.

These data are missing because the GWAS data that we downloaded for somatic expansion of *HTT* (from the GeM-HD Consortium’s recent paper in *Nature Genetics*) did not include variants on the X chromosome (see Fig. 3a from that paper, copied below):

4. Figure 5

There is a lot of information here and I think the presentation can be improved with some minor modifications.

Panel A: The middle columns present p-values and color code them according to significance. This presents the same (or similar) information twice. I think it would be more informative to present the effect sizes and directions and use the color coding to indicate significance level (nominal, genome-wide significance, or neither).

We appreciate the suggestion and attempted to implement it (see below), but we realized that the effect sizes were not very informative because we were unable to compute effect sizes on a consistent, interpretable scale. We were unable to measure effect sizes on the scale of repeat length (because we could only indirectly estimate lengths of expanded repeats >150bp by counting IRRs). Additionally, STRs with longer mean expanded allele lengths would be expected to expand faster than those with shorter alleles, further complicating comparison of effect sizes. In light of these complexities, we estimated effect sizes in units of standard deviations of the somatic expansion phenotypes, but these were difficult to compare given the highly variable amounts of noise in the instability measurements for different STRs.

Locus (Mb)	Motif	Gene	Expression in blood (percentile)	Long allele carriers (EUR)		Effect size (units of s.d.)			Relative contributions of genetic modifiers of repeat expansion
				n	freq.	MSH2 assoc	MSH3 assoc	Age assoc	
chr2:37	AC	STRN prom.	87.7	126089	30.0%	-0.094	-0.019	0.003	
chr15:74.9	AC	MPI	88	29365	7.0%	-0.066	-0.0083	0.0038	
chr16:4.8	AC	GLYR1	96.4	10292	2.4%	-0.031	-0.0065	0.0016	
chr2:62.7	AG			30289	7.2%	-0.056	-0.0024	0.00076	
chr4:28.7	AAG			57827	13.8%	-0.044	0.0023	0.00025	
chr9:36.5	AAG			134142	31.9%	-0.058	-0.00032	0.00018	
chr13:102.2	AAG	FGF14	<58	118085	28.1%	-0.043	-0.0061	-0.00014	
chr18:55.6	AGC	TCF4	75.2	36158	8.6%	0.00041	0.014	0.0027	
chr2:232.4	AAAG			6617	1.6%	-0.015	-0.014	0.0013	
chr11:123.5	AAAG	GRAMD1B	79.2	112150	26.7%	-0.058	-0.021	0.001	
chr12:7.1	AAAG	CLSTN3	95.6	6781	1.6%	-0.013	-0.0095	-0.00033	
chr17:74.5	AAAG	CD300A	99.4	22082	5.3%	-0.017	-0.0096	0.0011	
chr17:57.5	AAAG	MSI2	86	3279	0.8%	-0.035	-0.014	0.00025	
chr19:14.8	AAAG	ADGRE2	97.4	204138	48.5%	-0.27	-0.14	0.0025	
chr1:34.8	AAACC			398708	94.8%	-0.059	-0.016	0.00053	
chr1:54.5	AAACC			241331	57.4%	-0.071	-0.012	8.5e-05	
chr17:48.9	AATAG	CALCOCO2	96.2	250748	59.6%	-0.091	-0.048	0.0016	

Given these difficulties, we decided to retain the p-values presented in the previous version of the figure (as these p-values give a clear sense of the striking amounts of GWAS signal at *MSH2* and *MSH3* as well as the relative strengths of these associations compared to age). The effect sizes are available in Supplementary Data 2.

It's also unclear why certain loci do not have relative contribution plots. A note in the legend can address this.

We agree that more detail was needed and have provided clarification in the legend of Fig. 5A with further information in Methods (section 4.5).

Panel B: How were these four expansions selected for presentation when there were many more in the table in panel A? A note in the legend can address this.

We selected the two STRs with the strongest age association and the two with the strongest genetic associations. We have added this information to the legend of Fig. 5B as suggested.

Panel C: What do the signs (+/-) and scale of the effects mean here? Negative means shorter allele? Z score represents what measured units (a link to a supplemental file or methods description that gives these conversions or explains the scaling would be useful in the legend). What do the effect sizes shown mean for Huntington's disease (scale)?

Thank you for raising these questions, which we agree needed clarification. Regarding the effect signs, yes, a negative effect direction indicates that the alternate allele associated with less somatic expansion. We have edited the legend of Fig. 5C to explain this.

Regarding effect scales, the numbers in the table are actually z statistics from the association tests (i.e., $\beta / \text{se}(\beta)$)—not z-scores, as we had incorrectly written—both for somatic expansion phenotypes as well as for HD clinical phenotypes. As such, they do not provide information regarding effect size, only effect significance and direction. We chose to emphasize effect direction and presence/absence of an effect (significance) given the difficulties of putting effect sizes for different phenotypes on a consistent scale (see above response regarding Fig. 5A). We have corrected and clarified this in the legend of Fig. 5C.

Why not make the colors consistent for positive and negative across the table. In general, the presentation of this data could be improved.

We had considered this as well and agree that coloring effects according to their sign consistently across the table would be a more natural choice. However, this turns out to make the table much more difficult to visually interpret (see the version below, in which we have colored positive effects green and negative effects red). Coloring effects according to whether they agree or disagree with the consensus effect direction of a genetic modifier more clearly brings out the main results of these analyses: general consistency in effect directions, apart from the CAG repeat expansion in *TCF4*, and dissimilarity between effects on STR expansions in blood versus effects on *HTT* expansion in brain (using HD phenotypes as a proxy).

		Somatic expansion of DNA repeat in blood											Hastening of Huntington's disease[26]							
Genetic modifier locus	Variant type	Variant	Cons. effect dir.	AF (%)	AC		AAG		AGC		AAAG		AAACC		AATAG		Onset	TFC6	SDMT	TMS
					chr2 37.0	chr15 74.9	chr9 36.5	HTT [26]	TCF4	chr11 123.5	chr19 14.8	chr1 34.8	chr1 54.5	chr17 48.9						
GADD45A	Common	1:67554471:T:C	97.6	-	-0.8	-3.4	0.2	0.8	5.5	-0.3	-2.6	1.1	0.8	-1.5		1.5	0.8	-0.2	0.9	
PARP1	Common	1:226349834:C:T	15.3	+	0.9	2.2	-0.2	4.8	-1.9	-1.0	4.3	-0.2	0.7	3.2		0.0	-1.1	-0.5	-1.0	
MSH2	Missense	2:47416318:G:A	1.6	-	-10.5	-7.3	-6.4	-13.4	0.0	-6.5	-29.6	-6.5	-7.9	-10.4		2.3	1.4	0.7	2.2	
MSH2	Common	2:47476151:G:A	13.4	+	3.4	-1.4	0.3	2.4	-0.6	0.6	3.8	0.6	-0.7	1.6		-1.8	-0.9	-0.1	-0.2	
MSH6	eQTL (+)	2:47784947:C:T	18.4	+	4.0	3.3	-1.0	5.8	-4.8	0.3	11.3	0.0	-0.5	1.3		-0.2	-3.0	-2.8	-0.7	
PMS1	Missense	2:189795860:C:T	0.2	-	-1.9	-3.4	-1.5	NA	-1.6	-2.4	-10.8	-2.1	-2.9	-2.9		-2.8	-2.3	-2.2	-1.6	
PMS1	Common	2:189809109:C:T	19.7	+	2.7	2.3	0.8	1.4	0.0	3.7	6.9	0.6	0.7	4.1		6.8	7.2	6.6	7.0	
XPC	Low frequency	3:14139206:C:T	2.0	+	1.7	-1.9	1.9	1.1	-3.5	4.4	8.5	1.5	2.5	-0.2		-0.7	1.8	1.3	1.5	
XPC	Common	3:14235797:A:G	22.1	-	-1.0	1.2	-1.0	-0.1	4.3	-0.6	-8.5	-1.1	-1.7	-2.9		-1.7	-0.8	0.2	-1.2	
SMARCA1	Common	4:94174939:A:AAC	47.0	+	3.0	2.0	1.8	2.1	-1.3	-0.1	9.4	0.7	0.5	1.1		-0.7	0.1	0.0	0.7	
MSH3	eQTL (+)	5:80632699:T:C	24.5	+	6.4	5.4	1.4	8.9	-10.3	8.7	56.7	8.7	4.9	22.0		8.6	15.8	14.9	11.5	
MSH3	Rare	5:80790685:A:G	0.5	-	-3.6	-3.8	-6.6	-6.5	0.3	-10.0	-25.8	-8.2	-10.0	-11.8		-6.3	-8.6	-7.0	-8.3	
MSH3	Missense	5:80813660:T:G	0.4	-	-1.1	-1.2	-3.4	0.4	4.5	-3.8	-17.6	-5.4	-6.9	-9.4		2.6	0.0	-0.9	1.0	
PMS2	Common	7:6003481:C:T	63.1	-	-4.7	-7.0	-3.7	-7.7	-5.4	-4.5	-16.1	-5.2	-3.0	-7.6		-2.0	-4.3	-3.9	-3.8	
PMS2	Low frequency	7:6006003:T:C	1.2	+	2.8	3.4	2.2	4.8	3.1	3.4	10.2	2.2	3.9	2.2		-1.0	-0.8	-0.5	-0.2	
NEIL2	Common	8:11763260:T:C	32.9	+	-0.2	0.9	2.2	3.7	1.1	1.5	5.1	1.3	1.8	3.0		0.2	0.6	0.4	0.3	
MLH3	sQTL+eQTL	14:75027345:C:T	45.9	-	-4.9	-2.8	-1.4	-6.1	3.0	-4.8	-36.5	-4.4	-6.3	-7.0		4.0	4.2	3.2	4.0	
FAN1	Common	15:30873651:C:T	55.7	-	-2.6	-4.8	-1.0	-13.2	-10.0	1.2	-1.2	-1.7	-2.2	-0.1		6.4	7.3	5.6	7.7	
FAN1	Missense	15:30910758:G:A	1.1	+	2.9	1.6	-0.4	8.8	6.5	-2.3	-0.1	1.5	0.5	0.8		12.5	15.2	11.1	12.6	
ATAD5	Common	17:30887878:C:G	27.3	+	-0.1	0.0	-0.5	7.7	6.9	1.8	0.7	0.7	0.8	1.5		0.4	-0.3	-1.0	-0.7	

5. “Phenome-wide association analyses of 67,405 repeat loci expanded in at least five UKB participants identified seven repeat loci involved in 23 likely-causal associations with quantitative phenotypes and 46 associations with disease phenotypes (Supplementary Data 7 and 8).”

How many binary and quantitative phenotypes were tested?

Thank you for pointing out that this information was not included in the main text. We have now reported the numbers of traits tested (6,483 disease phenotypes and 57 heritable quantitative traits) in Results (p. 8).

Referee #4 (Remarks on code availability):

Detailed readme, code, and example output provided.